# Tracking single hiPSC-derived cardiomyocyte contractile function using CONTRAX an efficient pipeline for traction force measurement

Gaspard Pardon [1,2,3,4,10], Alison S. Vander Roest [1,3,5,11], Orlando Chirikian[6], Foster Birnbaum [2], Henry Lewis[1], Erica A. Castillo[1,4], Robin Wilson[1], Aleksandra K. Denisin [1], Cheavar A. Blair[1,4,7], Colin Holbrook[2], Kassie Koleckar[2], Alex C. Y. Chang [2,3,8,9], Helen M. Blau [2,3] & Beth L. Pruitt [1,3,4,6] ✉

Cardiomyocytes derived from human induced pluripotent stem cells (hiPSC-CMs) are powerful in vitro models to study the mechanisms underlying cardiomyopathies and cardiotoxicity. Quantification of the contractile function in single hiPSC-CMs at high-throughput and over time is essential to disentangle how cellular mechanisms affect heart function. Here, we present CONTRAX, an open-access, versatile, and streamlined pipeline for quantitative tracking of the contractile dynamics of single hiPSC-CMs over time. Three software modules enable: parameter-based identification of single hiPSC-CMs; automated video acquisition of >200 cells/hour; and contractility measurements via traction force microscopy. We analyze >4,500 hiPSC-CMs over time in the same cells under orthogonal conditions of culture media and substrate stiffnesses; +/− drug treatment; +/− cardiac mutations. Using undirected clustering, we reveal converging maturation patterns, quantifiable drug response to Mavacamten and significant deficiencies in hiPSC-CMs with disease mutations. CONTRAX empowers researchers with a potent quantitative approach to develop cardiac therapies.

Cardiomyocytes (CMs) differentiated from human induced pluripotent stem cells (hiPSCs) have emerged as a powerful in vitro model for the study of cardiomyopathies, cardiotoxicity, and in drug development[1,2]. However, the most important function of these cardiomyocytes—their ability to contract—remains cumbersome to quantify at high throughput and over time in longitudinal assays[1].

Current assays of cardiomyocyte contractile function are often mostly qualitative, lack field-wide standards, and generally fail to offer

[1]Departments of Mechanical Engineering and of Bioengineering, Stanford University, School of Engineering and School of Medicine, Stanford, CA, USA. [2]Baxter Laboratory for Stem Cell Biology, Department of Microbiology and Immunology, Stanford University School of Medicine, Stanford, CA, USA. [3]Stanford Cardiovascular Institute, Stanford University School of Medicine, Stanford, CA, USA. [4]Departments of Bioengineering and Mechanical Engineering, University of California, Santa Barbara, CA, USA. [5]Department of Pediatrics (Cardiology), Stanford University School of Medicine, Stanford, CA, USA. [6]Biomolecular Science and Engineering Program, University of California, Santa Barbara, CA, USA. [7]Department of Physiology, University of Kentucky, Lexington, KY, USA. [8]Shanghai Institute of Precision Medicine and Department of Cardiology, Ninth People's Hospital, Shanghai Jiao Tong University School of Medicine, Shanghai 200125, China. [9]Division of Cardiovascular Medicine, Stanford University School of Medicine, Stanford, CA, USA. [10]Present address: School of Life Sciences, EPFL École Polytechnique Fédérale de Lausanne, Lausanne, Switzerland. [11]Present address: Department of Biomedical Engineering, Michigan Engineering, University of Michigan Ann Arbor, MI, USA. ✉e-mail: blp@ucsb.edu

the scalability and throughput necessary for larger-scale studies[1,2]. For example, high-speed video microscopy of CM monolayer contraction is relatively uncomplicated and scalable, but it only provides semi-quantitative measurements of contractile frequency and speed[2–5]. Recent advances have enabled quantifying the total contractile force. Multicellular constructs, engineered heart tissue or heart-on-chip systems advantageously reveal the effect of cell-cell interactions, which have been shown to promote hiPSC-CM maturation[5–11], nevertheless these data often remain difficult to standardize across platforms and studies. Further, substrate stiffness, culture medium and maturation time have been shown to affect cardiomyocyte contractile function at the single cell and subcellular level by driving differences in cell morphology, intracellular protein structures, calcium transients, and sarcomere assembly and dynamics[12–15]. Single cell traction force microscopy (TFM) complements the above approaches by offering quantitative and standardizable measurements of contractile force at the single-cell level with the potential to resolve the direct impact of subcellular defects, to reveal cell-to-cell variations, and to characterize population heterogeneity[3,16–26]. TFM is a conceptually straightforward method that measures the traction stress exerted by a cell on an elastic substrate, often a hydrogel. TFM tracks the hydrogel deformation using fiducial markers, generally embedded fluorescent microspheres[16,17,19,20,27,28]. The measured deformation is then used to compute the traction stress. Additionally, the stiffness of the hydrogels can be tuned to mimic healthy physiologic or pathologic tissue mechanics[12–14,21,22,27]. Such a single cell measurement approach also allows for performing other types of live assays at the single-cell level, such as sarcomere dynamics or electrophysiology patch clamp measurements.

Despite recent progress[19,20], TFM data acquisition and downstream analysis remain time-consuming processes requiring a significant level of expertise. The acquisition of video recordings requires several hours per experiment; most of that time spent manually selecting cells of interest; and data processing with currently available TFM computational packages[19,29–31], or with the ImageJ Traction Force Microscopy plugin[28], is demanding in computing power and on user involvement. These challenges are compounded by the size of the datasets; measuring dynamic contractions in CMs requires seconds-long, high-framerate videos (at least 5-10 s and 150-400 frames). Consequently, the currently achievable throughput is low, limiting the ability to measure enough single cells and to track changes in the same individual cells over time. These limitations have left the potential of such single-cell resolution measurements largely untapped.

Here, to address these bottlenecks, we developed CONTRAX, a pipeline that provides an accessible, comprehensive, and streamlined TFM assay workflow. CONTRAX enables quantitative *tracking* of the *contractile* dynamics of thousands of single hiPSC-CMs over time at increased throughput (Fig. 1). In total, we analyzed >4,500 hiPSC-CMs at single-cell resolution to demonstrate the ability of CONTRAX to characterize the functional heterogeneity and contractile maturation in hiPSC-CM populations and sub-populations, to assess the effect of drugs on hiPSC-CMs contractility, and to study the impact of the disease-causing mutation on the functional biology of hiPSC-CMs.

First, to characterize the contractile phenotype in hiPSC-CMs populations and to demonstrate the power of CONTRAX in tracking a large number of single cells over time, we performed a Longitudinal experiment (Figs. 2–7). We studied the impact of substrate stiffness and culture media over three-time points (d20, d30, and d40−days post differentiation) across 2 × 2 orthogonal experimental conditions (two substrate stiffnesses: 10 kPa (healthy myocardium) and 35 kPa (fibrotic myocardium), and two media compositions: M16$_{lac}$: lactate-rich and M16$_{glu}$: glucose-rich).

Second, to demonstrate the ability to analyze the same cells before and after a drug treatment, we performed a Drug experiment

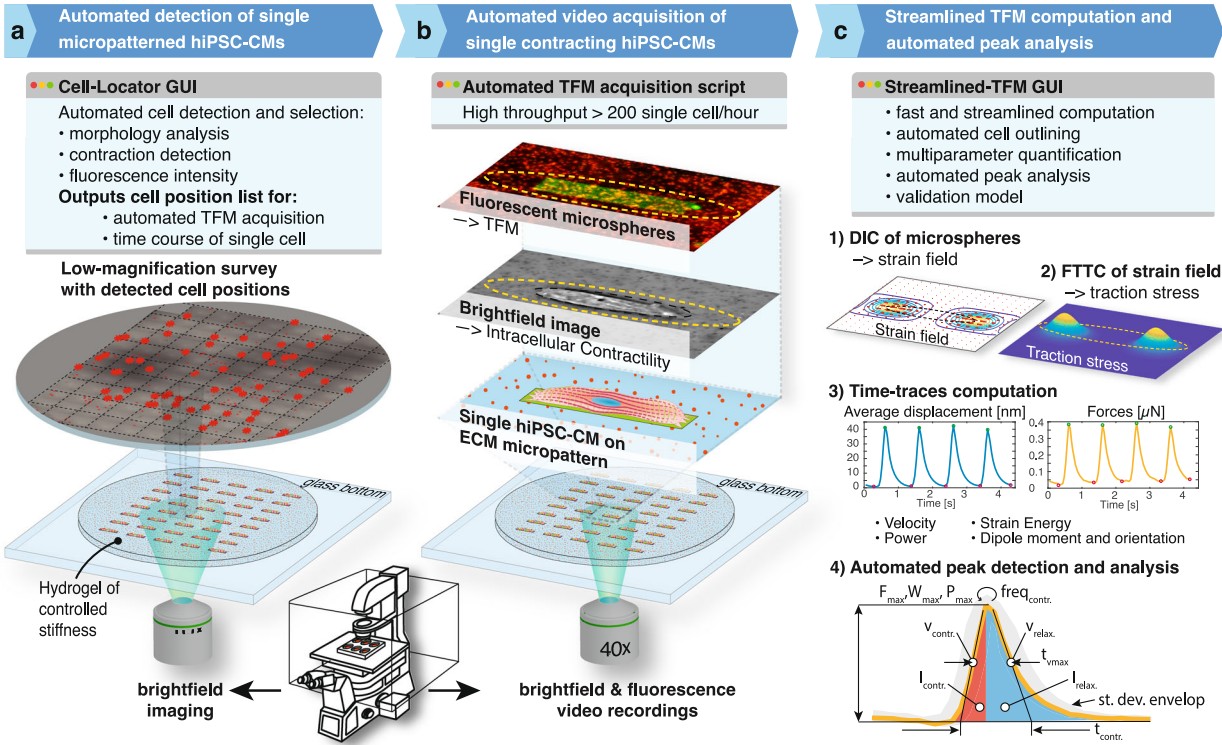

**Fig. 1 | The CONTRAX workflow for traction force microscopy (TFM) is comprised of three main steps and software modules that streamline and automate the workflow, thereby increasing the throughput of the assay.** Each module can be used independently or within the workflow. The modules are open-source and available online through our GitHub repository[77]. More details regarding the definition and calculation for the symbols of panel 4) are provided in Supplementary Information. Abbreviation: DIC: Digital Image Correlation; FTTC: Fourier Transform Traction Cytometry; ECM: ExtraCellular Matrix.

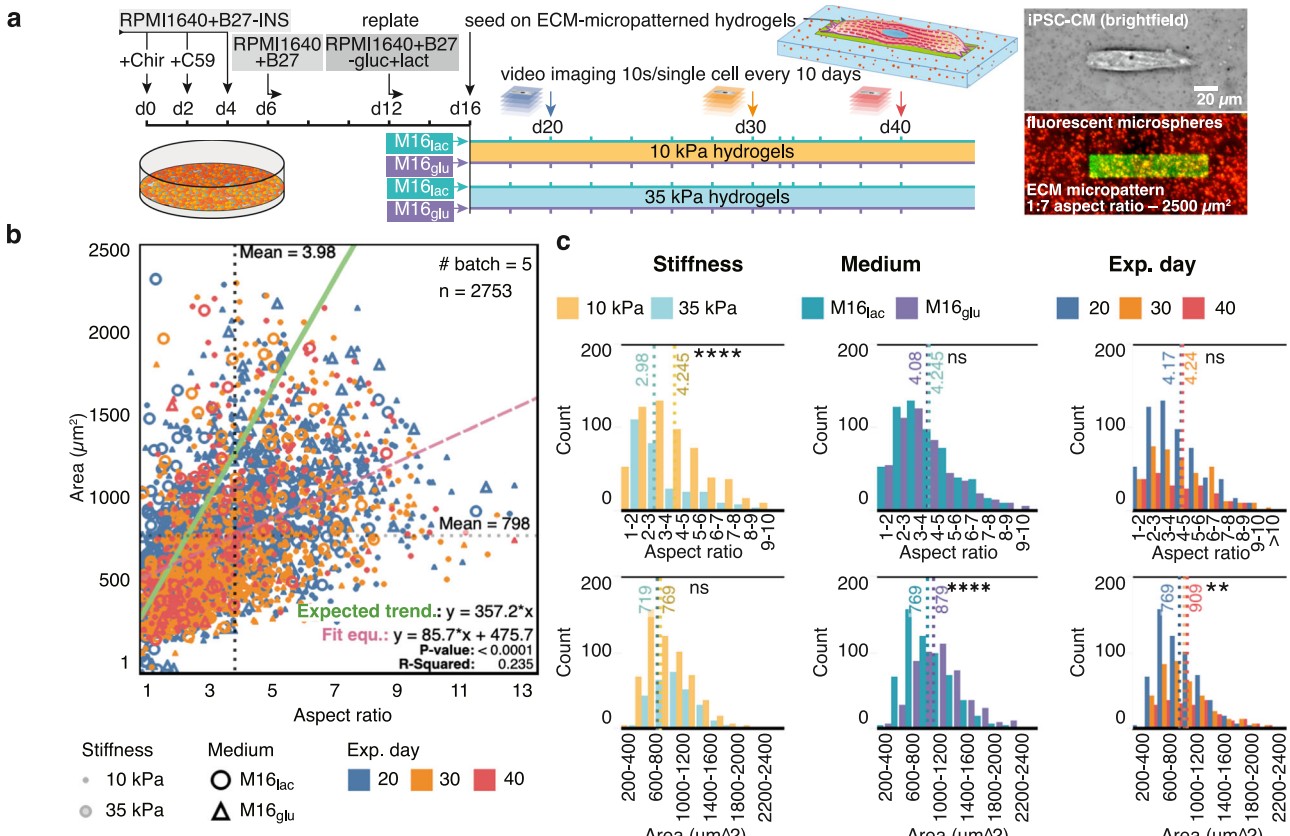

**Fig. 2 | Characterization of cell morphology reveals heterogeneities in populations of hiPSC-CMs. a** Left: Schematic of hiPSC-CM culture, differentiation, micropatterning and experimental groups. Right: brightfield image of a single micropatterned hiPSC-CM and corresponding fluorescence image; the ECM micropattern (GFP-labelled gelatin + Matrigel) printed on polyacrylamide hydrogel with embedded red-fluorescent. Brightfield images were systematically recorded for each single measured hiPSC-CM. The corresponding protein micropattern were imaged in > 5 independent experiments and in other studies in our lab, but was not systematically imaged by fluorescence microscopy for each hiPSC-CM. Abbreviation: M16$_{lac}$: lactate-rich and M16$_{glu}$: glucose-rich) media. **b** Single hiPSC-CMs adopt an elongated morphology when adhered onto the ECM micropatterns (7:1 aspect ratio). Green line: theoretical trend for the area of a cell filling the whole width of the micropatterned while elongating along the micropattern (Eq. 1). Red dashed line,

linear regression; dotted lines: median values for spread area and aspect ratio. Sample number n is provided in the figure panel and were analyzed from five differentiation batches in independent experiments. Linear regression model was tested against the null hypothesis using an F test, with p values < 0.001 against the null hypothesis of zero-slopes for the models and a p-value of 0.0599 when comparing the slopes. **c** Stiffness, maturation medium, and experimental day influence the distributions of spread area and cellular aspect ratio. Normality versus lognormality tests were performed using multiple tests including D'Agostino-Pearson omnibus, Anderson-Darling, Shapiro-Wilk, and Kolmogorov-Smirnov tests for normality with an alpha coefficient of 0.05, indicating a lognormal distribution. Unpaired t-test using parametric test were performed on the log value of the data using a two-tailed $P$ value at confidence level of 95%. **d** and **e** see Figure S1 In all panels, *$p < 0.05$, **$p < 0.005$, ***$p < 0.001$, ****$p < 0.0001$. ns, not significant.

(Fig. 8). We analyzed the response of single hiPSC-CMs to Mavacamten, a cardiac myosin inhibitor recently FDA-approved to treat obstructive hypertrophic cardiomyopathy in the United States. We measured the morphologic and contractile phenotypes of cells before and after one-hour treatment with a physiologic dose of Mavacamten.

Third, to demonstrate the ability to observe distributions of morphological and contractile phenotypes across populations, we performed a disease modelling experiment (Fig. 9). We studied the impact of disease mutations using multiple cell lines with or without mutations in the dystrophin gene leading to Duchenne Muscular Dystrophy (DMD)-linked dilated cardiomyopathy. For controls, we performed measurements on multiple lines, including matched isogenic cell lines with CRISPR-induced or CRISPR-corrected mutations.

## Results

### CONTRAX workflow and performance

The CONTRAX workflow comprises three stand-alone yet interlocking software modules with graphical user interfaces (GUIs) (Fig. 1). First, the *Cell-Locator* module (Fig. 1a) automates the detection and localization of single cells. Second, the *Automated TFM acquisition* module (Fig. 1b) loops through the position list generated by the *Cell-Locator*

and automates the recording of high-resolution TFM videos. Third, the *Streamlined TFM* module (Fig. 1c) quantifies the contractile phenotype of each recorded cell.

The *Cell-Locator* module (Fig. 1a) uses a low-magnification (10x) survey of the cells on the hydrogel and generates a position list of cells, with the option of using user-defined criteria, e.g., cell area, elongation, orientation, fluorescence intensity, or contractility thresholds, and to filter for cells matching these criteria. Additionally, fluorescently labelled ECM proteins could be used to monitor the quality of each micropatterns, as shown in Fig. 2a. This automated, targeted and reproducible selection, which can be applied as gates in a similar way to what is done in Flow Cytometry and Fluorescence-Activated Cell Sorting (FACS) sorting. Automated selection alleviates user-induced cell selection bias across experiments and provides population-wide cell-morphology metrics. Further, this module increases the robustness and throughput of the assay by limited and targeted acquisition of TFM videos of strictly relevant cells. On average, 10–50% of patterns are occupied by cells, and the Cell-Locator module identified >200 relevant cells per patterned gel substrates out of a total of 1600 patterns regions per gel substrates, increasing both the speed and throughput of acquisition compared to an undirected acquisition across every

possible patterns. The generated position list also enables returning to the same single cells, thereby enabling single cell tracking in long-itudinal studies.

The *Automated TFM acquisition* module (Fig. 1b) loops over the cell position lists generated by the *Cell-Locator*, moving the micro-scope stage to the selected cell locations, performing an optional autofocus step, and acquiring video recordings according to user-defined imaging parameters at a magnification of 40X (0.275 micron /pixel). The module was developed as a script for *Micro-Manager*, a widely used open-source microscopy software, and as a GUI macro for *Zen Blue*, the proprietary Zeiss microscopy software.

The *Streamlined TFM* module (Fig. 1c) computes the traction stress from the acquired TFM videos using Digital Image Correlation (DIC) to track fluorescent fiducial markers (i.e., fluorescent micro-spheres) and measure the deformation of the hydrogel (Fig. 1c1). Fourier Transform Traction Cytometry (FTTC) back-calculates the traction stress from the hydrogel deformation (Fig. 1c2) and its trace over time (Fig. 1c.3)[16,17]. This module innovates on prior work[19] by using parallel computing to speed up calculations, introducing a streamlined interface to reduce the burden on the users without compromising on the control of the analysis parameters, and adding custom-built algorithms. These algorithms provide: automated outline detection, and contraction peaks detection to automate the quantification of multiple contractile parameters (Fig. 1c4): the maximum delta between the most relaxed and contracted states of the cardiomyocytes over several cycles of contractions of the force $F_{max}$, work $W_{max}$, power $P_{max}$, strain energy $E_{max}$, as well as the frequency $f_{contr}$, the contraction and relaxation velocities $v_{contr}$ and $v_{rel}$, the contraction and relaxation impulse $I_{contr}$ and $I_{rel}$[32,33], the contraction duration at maximal velocity, $t_{vmax}$, and the overall duration of the contraction, $t_{contr}$. (see Supple-mentary information for details of calculations) The TFM computation was validated with a validation model and standardization tool (see Supplementary information)[34], enabling the optimization of analysis parameter and the results comparable across labs and studies.

We utilized CONTRAX to analyze > 4500 cells with enhanced performance in terms of workflow parallelization and efficiency and throughput with minimized user intervention. The *Cell-Locator* mod-ule could generate a position list for hundreds to thousands of single hiPSC-CMs of interest in only a few minutes. Thereafter, the *Automated TFM acquisition* module enabled a TFM video acquisition throughput of > 200 cells/hour, an order of magnitude higher than conventional manual acquisition, without compromising spatial resolution or accuracy[20], and with significantly-reduced user-burden. Finally, the *Streamlined TFM* module shortened data analysis of this large dataset by > 20x compared to previous tools (on a regular MacBook Pro lap-top), and running this workflow using cloud computing infrastructure can tremendously accelerate the analysis[19]. Automated cell outline detection is accomplished using a single bright field image, reducing the acquisition time and amount of user mouse clicks by at least 5x during video analysis preprocessing. Using a bright field image alle-viates the need for a fluorescent membrane label or dye, thereby preserving a fluorescent channel for additional labeling observations. Further, by exploiting parallel computing, data analysis was shortened to less than 150 s per typical TFM video (using a 12-core 2.7 GHz pro-cessor workstation).

## Longitudinal study: impact of substrate stiffness and culture media

We performed a large longitudinal experiment comparing control cells contractile behavior on different stiffnesses and with different media over a 20-day period (Figs. 2–7). At day 15-17 after differentiation[15,21,35,36] from wild-type WTC hiPSCs[36–40], hiPSC-CMs were seeded on poly-acrylamide hydrogel substrates. The hydrogels had a targeted stiffness of 10 kPa or 35 kPa and were micropatterned by microcontact printing of Matrigel extracellular matrix (ECM) proteins that constrain the cell

morphology[12,41–43]. After seeding, micropatterned hiPSC-CMs were left to recover for 2–5 days, and subsequently imaged and analyzed using CONTRAX. Details of the experimental workflow and of the fabrication procedures for the hydrogel substrates are reported in the Methods section and in the online Supplementary Methods in Supplementary information. We further note that micropatterns are not strictly required for TFM measurements with CONTRAX, as long as the inter-spacing of cells is sufficient to mitigate unwanted cell-cell dis-turbances. However, providing physiologically-relevant elongated shape with patterns was shown to enhance contractile maturity of hiPSC-CMs, and further contributes to standardizing the measurement of contractile force against cell-to-cell variability in spread area, shape and size[19].

## Micropatterning controls cell shape
Native adult CMs display well-aligned myofibrils and elongated shapes, features that are often not spontaneously developed in largely immature hiPSC-CMs. However, previous reports showed that hiPSC-CMs can adopt more physiological elongated shapes and enhanced contractile maturity when seeded on physiological micropatterned substrates[12,42]. Changes in cell morphology and contractile strength and velocity have also been correlated with changes in gene expres-sion associated with more mature cardiomyocytes[8]. Therefore, we seeded our hiPSC-CMs on rectangular micropatterns of aspect ratio 7:1, shown to elicit maximal alignment and contractile efficiency (Fig. 2a)[12,44]. We deliberately chose a micropattern area of ~2500 μm² (~1500 μm² for the Mavacamten drug study) larger than the reported average spread area of unpatterned d30 hiPSC-CMs[25], to afford us the ability to observe a range of cell morphology while maintaining the necessary spatial constraints to induce cell alignment.

Using the *Cell-Locator* and the *Streamlined TFM* modules (Fig. 1a & b), we measured the morphology of our micropatterned hiPSC-CMs as a function of substrate stiffness, culture medium and time (Fig. 2b, c and Figure S1.a). Our micropatterned hiPSC-CMs suc-cessfully adopted elongated shapes (Fig. 2b) and were constrained within the micropattern, with a measured mean cell aspect ratio of 3.98 ± 2.02 (median = 3.58) and a mean spread area of 798 ± 383 (std) μm² (median = 733 μm²) across all groups at d20. If all cells filled the whole width of the micropattern as they elongate, the cell spread area would be expected to increase with aspect ratio following a linear trend defined by: $A = L*W = r*W^2$ (1), where $A$ = micropattern area (μm²), $L$ = length (μm), $r$ = length:width aspect ratio, and $W$ = width (μm) (for $A$ = 2500 μm², $W$ = 18.9 μm, and $r$ = 7:1) (solid green line in Fig. 2b). In contrast, our measurements showed a relatively large variance and a flatter slope (red dashed linear regression in Fig. 2b) compared to (1). Nevertheless, as expected from the dimensional constraints imposed by the micropatterns, cells with an aspect ratio exceeding that of the micropattern ( > 7:1) had a smaller spread area than the maximum possible 2500 μm² on the micro-pattern (Figure S1a, b) due to a reduced cell width. Overall, this confirms that the cells were adequately constrained on the micro-patterned and the large variance is mainly a result of potential defects in the micropatterns and, mainly, of heterogeneity in cell size.

## Stiffness, culture medium and maturation time affect cell morphology
The morphology of unpatterned CMs has been observed to change with increased substrate stiffness in vitro, with larger cells found on stiffness matching that of stiffer fibrotic tissues[14]. On our micropattern, a substrate stiffness of 35 kPa resulted in wider and shorter hiPSC-CMs compared to the control (M16$_{lac}$ on 10 kPa at d20), with similar spread area but smaller aspect ratio (Fig. 2c & Table 1a).

CMs also experience a switch in their metabolism from glycolysis to mitochondrial oxidative metabolism during early development and

**Table 1 | Summary of measurements across experimental groups for Figs. 2, 3 and 4**

**a**

| Experimental group | | Spread area (mean ± std (n)) ($\mu m^2$) | Aspect ratio (mean ± std (n)) | p-value (bold = significant) | Linear regression area vs aspect ratio |
|---|---|---|---|---|---|
| Substrate stiffness $M16_{lac}$ | 10 kPa | 769 ± 353 (590) | 4.25 ± 1.92 (590) | Spread area: $p = 0.0687$<br>Aspect ratio: $p < 0.0001$ | $m = 84.11$<br>$p = 0.0599$ |
| | 35 kPa | 719 ± 314 (282) | 2.98 ± 1.81 (282) | | |
| Media 10 kPa | $M16_{lac}$ | 769 ± 353 (590) | 4.25 ± 1.92 (590) | Spread area: $p < 0.0001$<br>Aspect ratio: $p = 0.0757$ | $m = 84.58$<br>$p = 0.1986$ |
| | $M16_{glu}$ | 879 ± 380 (498) | 4.08 ± 1.99 (498) | | |
| Time course: $M16_{lac}$ 10 kPa | d20 | 769 ± 353 (590) | 4.25 ± 1.92 (590) | Spread area (20 vs. 30; 20 vs. 40; 30 vs. 40) $p = (0.209; 0.0018; 0.140)$ Aspect ratio (20 vs. 30; 20 vs. 40; 30 vs. 40) $p = (0.7794; 0.4587; 0.2311)$ | $m = 91.04$<br>$p = 0.1044$ |
| | d30 | 830 ± 421 (376) | 4.37 ± 2.05 (376) | | |
| | d40 | 909 ± 460 (220) | 4.23 ± 2.22 (220) | | |

**b**

| Experimental group | | Contractile Force: geometric mean [95% conf. interval] (n) (nN)* | p-value (bold = significant) | Slope of linear regression vs Aspect ratio (mean ± SE) |
|---|---|---|---|---|
| $M16_{lac}$ | 10 kPa | 13.88 (12.42–15.52) (590) | $p < 0.0001$ | 3.934 ± 0.776 |
| | 35 kPa | 23.34 (19.54–27.93) (282) | | 10.71 ± 1.698 |
| 10 kPa | $M16_{lac}$ | 14.16 (12.71–15.78) (590) | $p = 0.0315$ | 3.934 ± 0.776 |
| | $M16_{glu}$ | 17.13 (14.93–19.68) (498) | | 4.701 ± 1.354 |

**c**

| Experimental group | | Power law regression fit p-value, $R^2$ | Log-log slopes & SE | p-value (bold = significant) |
|---|---|---|---|---|
| $M16_{lac}$ | 10 kPa | Force: <0.0001, 0.474 (F = 494.0, DFn = 1, DFd = 549) Power: <0.0001, 0.764 (F = 168.7, DFn = 1, DFd = 284) | 0.687 ± 0.0309<br>1.558 ± 0.0365 | $p = 0.0029$ (F = 9.924, DFn = 1, DFd = 833) |
| | 35 kPa | Force: <0.0001, 0.373 (F = 1819, DFn = 1, DFd = 560) Power: <0.0001, 0.649 (F = 523.7, DFn = 1, DFd = 283) | 0.892 ± 0.069<br>1.796 ± 0.0785 | $p = 0.0032$ (F = 9.757, DFn = 1, DFd = 843) |
| 10 kPa | $M16_{lac}$ | Force: <0.0001, 0.474 (F = 494.0, DFn = 1, DFd = 549) Power: <0.0001, 0.764 (F = 420.9, DFn = 1, DFd = 497) | 0.687 ± 0.0309<br>1.558 ± 0.0365 | $p = 0.0317$ (F = 4.625, DFn = 1, DFd = 1046) |
| | $M16_{glu}$ | Force: <0.0001, 0.459 (F = 1819, DFn = 1, DFd = 560) Power: <0.0001, 0.698 (F = 1137, DFn = 1, DFd = 492) | 0.792 ± 0.0386<br>1.677 ± 0.0497 | $p = 0.0512$ (F = 3.810, DFn = 1, DFd = 1052) |

**a** Cell morphology measurements: mean values show a statistically significant decrease in aspect ratio on stiffer 35 kPa substrates and larger cell area in, $M16_{glu}$. The standard deviations show a large variance across each group. Data distribution was tested for normality and lognormality, and comparisons were made accordingly using a two-tailed unpaired t-test with confidence interval of 95% for the effect of stiffness and media, and using ordinary ANOVA test corrected for multiple comparison using the Tukey's method and reporting multiplicity adjusted p-values using a confidence interval of 95%. **b** Contractile force measurements: The geometric mean of the contractile force (lognormal distribution) shows stronger force production on 35 kPa substrates and in $M16_{glu}$ medium versus control conditions of 10 kPa substrate stiffness, $M16_{lac}$ culture medium, and experimental day 20 (d20 post differentiation). In the last column, we report the slope of the linear regression and the standard error of estimate for comparison of the linear model. Data distribution was tested for normality and lognormality, and comparisons were made accordingly using two-tailed unpaired t-test with confidence interval of 95% for the effect of stiffness and media. **c** The slope of log the contraction force and power versus the log of the contractile velocity is higher on 35 kPa and in $M16_{glu}$ medium versus control condition. The log-log regression models were compared using extra sum-of-squares F tests using a 95% confidence interval against the null hypothesis of zero-slopes and against each other.

\* $m$ = slope of the linear regression.

*(lognormal distribution).

reverting to glycolysis under hypertrophic stress[36,45,46]. Accordingly, for d20 on 10 kPa substrates, we observed that the glucose-rich $M16_{glu}$ medium resulted in larger cell area than the lactate-rich $M16_{lac}$ medium, with no difference in cell aspect ratio. Over time, however, the $M16_{lac}$ medium led to an increase in the spread area in our $M16_{lac}$ control cells, surpassing that of the $M16_{glu}$ cells by d30, notably on stiffer 35 kPa substrates (Fig. 2b, Figure S1.a-b & Table 1a). This trend correlates with an increased mitochondrial respiration in $M16_{lac}$ cells compared to $M16_{glu}$ cells at d30, as measured by Seahorse metabolic assay (Figure S2).

**Stiffer substrates increase contractile strength**

An increased substrate stiffness was shown to induce CMs to contract more strongly[14,32]. Accordingly, on 35 kPa substrates, our control in $M16_{lac}$ at d20 showed increased contractile force compared to control 10 kPa substrates (Fig. 3a, b & Table 1b). Further, the contractile force increases with aspect ratio (or cell area) in a significantly more pronounced manner on stiffer 35 kPa substrates compared to control 10 kPa substrates (Fig. 3b and Figure S1a, b). Interestingly, both contractile force and contractile power presented the signature of a power-law scaling with contraction velocity (Fig. 3c & Table 1c; see Figure S3 and Table S1 & S2 for comparison of fit models), meaning that the contractile force or power scaled exponentially with the

contractile velocity. Here again, the slope of the power law is steeper on stiffer 35 kPa substrates. The effect of increased substrates stiffness on the contractile output is summarized in the spider plot in Fig. 3d, where the measured parameters normalized to 10 kPa substrates in $M16_{lac}$ media reveal a robust and significant changes across all metrics.

**Glucose-rich culture medium promotes early contractile maturation**

We questioned whether the type of metabolic substrate affects the maturation of the contractile function in hiPSC-CMs. At d20, similar to its effect on cell morphology, the $M16_{glu}$ medium resulted in a higher contractile force than in control $M16_{lac}$ cells (Fig. 4a, b & Table 1B). However, the two different culture media did not significantly affect the slope between contractile force and aspect ratio (Fig. 4b), or area (Figure S1). The media did not either meaningfully affect the observed power-law relationship between contractile force or power with the contraction velocity (Fig. 4c & Table 1c). The effect of these two substrates media on the contractile output is summarized in the spider plot in Fig. 4d, where the measured parameters normalized to $M16_{lac}$ medium on 10 kPa substrates reveal no significant change in terms of cell morphology, but a significant increase in terms of contractile strength and dynamics, e.g., force, velocity, in $M16_{glu}$ medium.

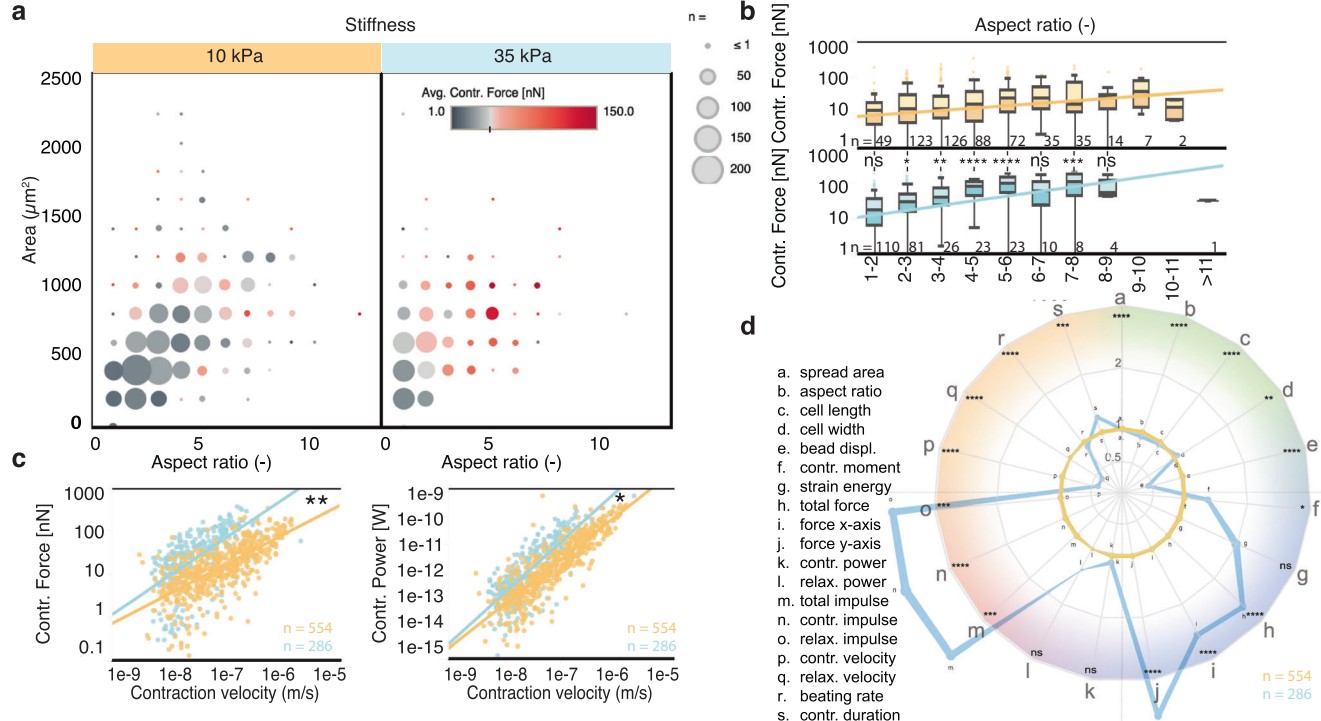

**Fig. 3 | Contraction force, velocity, and power are impacted by substrate stiffness at day 20 in M16$_{lac}$ medium. a** Balloon plot of the contractile force as a function of cell aspect ratio and area (grouped by range) shows a generally higher force on 35 kPa substrate and fewer cells of large spread area. **b** Force production increases with aspect ratio until reaching a maximum around aspect ratio 7:1, beyond which force seems to taper off. Sample number n is provided in the figure panel. Boxplots show median value and the upper and lower quartile (25th and 75th percentile respectively), and the whiskers indicate data within 1.5 times the inter-quartile range. Differences in force between aspect ratio were tested for significance using ordinary two-way ANOVA using Šidák's multiple comparisons test with a single pooled variance to adjust the p-value for multiple comparisons. Linear regression models were tested against the null hypothesis using an F test using a 95% confidence interval, with p-values 0.0002 ($F = 14.36$, DFn = 1, DFd = 540) and <0.0001 ($F = 73.04$, DFn = 1, DFd = 283) respectively against the null hypothesis of zero-slopes and a p-value of <0.0001 ($F = 46.39$, DFn = 1, DFd = 823) when comparing the slopes. **c** Contraction force and contraction power follow a, potentially allometric, power law (linear in this log-log plot) with contraction velocity that significantly depends on substrate stiffness. Sample number n is provided in the

figure panel. The log-log regression models were compared using extra sum-of-squares F tests using a 95% confidence interval, with p values < 0.0001 ($F = 494.0$, DFn = 1, DFd = 549) and <0.0001 ($F = 168.7$, DFn = 1, DFd = 284) respectively against the null hypothesis of zero-slopes and a p-value of 0.0029 ($F = 9.924$, DFn = 1, DFd = 833) when comparing the slopes of force vs velocity; and p-values < 0.0001 ($F = 1819$, DFn = 1, DFd = 560) and <0.001 ($F = 523.7$, DFn = 1, DFd = 283) respectively against the null hypothesis of zero-slopes and a p-value of 0.0032 ($F = 9.757$, DFn = 1, DFd = 843) when comparing the slopes of power vs velocity. **d** Spider plot providing a summary of the comparison of all measured parameters at day 20, normalized to 10 kPa in M16$_{lac}$ medium. The color shading groups parameters related to: green: cell morphology, blue: contractile stress, red: contractile dynamics, orange: temporality (see Figure S10). Line thickness is proportional to the standard deviation. Individual pairs of parameters were tested for significant differences using two-tailed unpaired parametric t-test with a confidence interval of 95%. Statistics show: 35 kPa vs. 10 kPa in M16$_{lac}$ medium at day 20. Sample number n is provided in the figure panel. In all panels, $*p < 0.05$, $**p < 0.005$, $***p < 0.001$, $****p < 0.0001$. ns, not significant.

## Lactate-rich medium promotes contractility and survival over time

To characterize the maturation of the contractile function over time, we repeatedly measured the same single cells at d20, d30 and d40 (Fig. 5 and Figures S5-S8). Interestingly, the contractile performance of the M16$_{glu}$ cells tend to decline over time, while that of the control M16$_{lac}$ cells tend to improve (Fig. 5a, b). The survival of the control M16$_{lac}$ cells was also observed to be better compared to M16$_{glu}$. Over time, the contractile performance was affected by the substrate stiffness, with a plateau (M16$_{lac}$) or decrease (M16$_{glu}$) of most contractile and morphology parameters on stiffer 35 kPa substrate compared to control 10 kPa substrates (Fig. 5b & Figures S4 & S5). This worsening of the contractile function on stiffer 35 kPa substrates was also accompanied by faster cell loss or death, which could be due to the consistently higher contractile stress experienced on these stiffer substrates. In line with a previously reported progressive loss of rhythmic beating on a rigid substrate[14], over time stiffer 35 kPa substrates led to an increase in the mean beating frequency, despite the externally imposed 1 Hz electrical pacing stimulation. (Figures S4 & S5) In addition, both the total contraction duration and contraction

impulse (contractile force integrated over contraction duration) decreased over the time course experiment (except for the control cells on 10 kPa in M16$_{lac}$ medium). The effect of maturation time on the contractile output is summarized in the spider plot in Fig. 5c (and Table S3), where the measured parameters normalized to M16$_{lac}$ medium on 10 kPa substrates clearly reveal how the various contractile parameter evolve over time under each condition. We note that appropriate controls and experimental repeats are important to make the experiments independent of experimental condition.

## Multidimensional phenotyping of contractile maturation trajectory

We leveraged CONTRAX's throughput to characterize distinctive multi-parameter contractile profiles of 2,753 single cells using undirected clustering. We used X-shift clustering and identified eight clusters using K-nearest-neighbor density estimation with the following input parameters: average contraction displacement, total contractile force and strain energy, contraction and relaxation velocities and power, contraction duration, beating frequency, cell spread area and aspect ratio (Fig. 6a–c)[47–49]. Clustering most strongly followed from

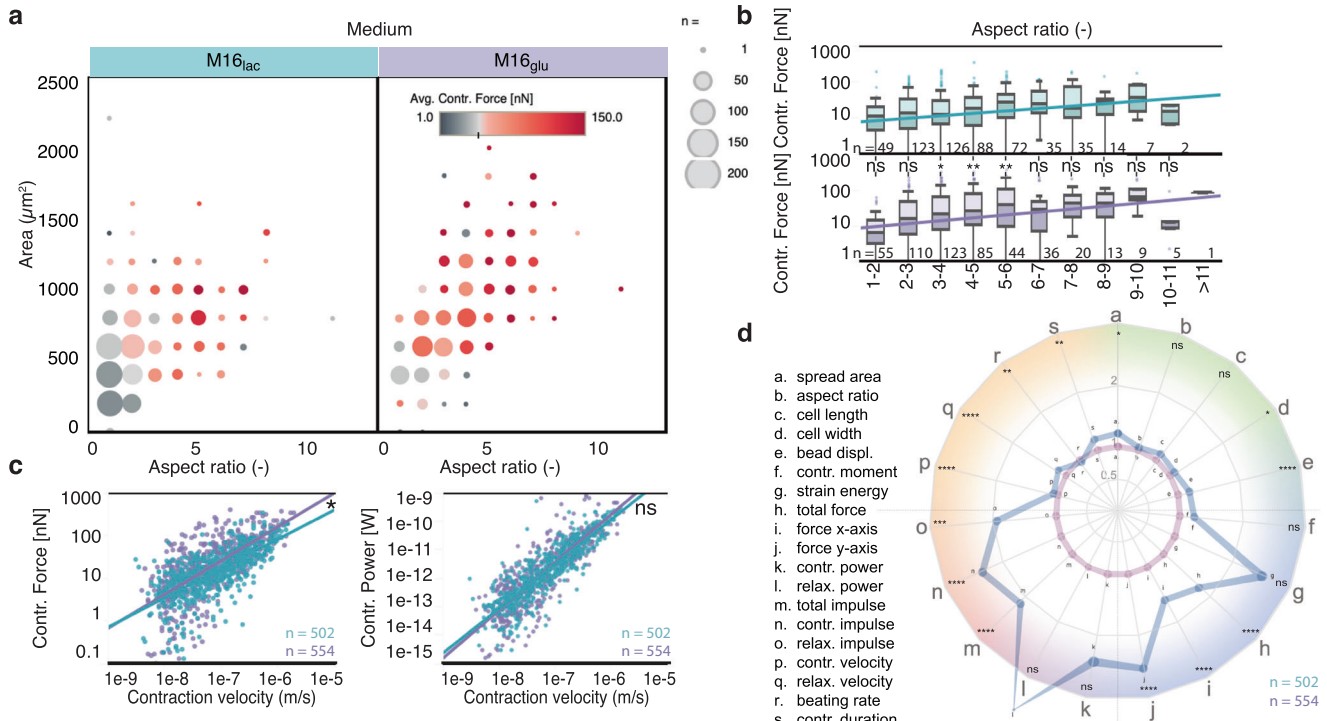

**Fig. 4 | Contraction force, velocity, and power are equally dependent on area and aspect ratio for both medium compositions at day 20 on 10 kPa substrates.** **a** Balloon plot of the contractile force as a function of cell aspect ratio and area (grouped by range) shows more elongated and larger cell in M16glu than in M16lac. **b** Force is equally dependent on aspect ratio in both media, except in for aspect ratios of 3–6. Sample number *n* is provided in the figure panel. Boxplots show median value and the upper and lower quartile (25th and 75th percentile respectively), and the whiskers indicate data within 1.5 times the interquartile range. Differences in force between aspect ratio were tested for significance using ordinary two-way ANOVA using Sidak's multiple comparisons test with a single pooled variance to adjust the *p* value for multiple comparisons. The linear regression model was tested against the null hypothesis using an F test using a 95% confidence interval, with *p* values 0.0009 (F = 11.22, DFn = 1, DFd = 498) and <0.0001 (F = 16.54, DFn = 1, DFd = 551), respectively, against the null hypothesis of zero-slopes and a *p* value of <0.3374 (F = 0.9210, DFn = 1, DFd = 1049) when comparing the slopes. **c** Contraction force and power follow a power law (linear in log-log plot) with contraction velocity. The linear regression for the force shows a weak dependence on medium composition. Sample number n is provided in the figure

panel. The log-log regression models were compared using extra sum-of-squares F tests using a 95% confidence interval, with *p* values < 0.0001 (F = 494.0, DFn = 1, DFd = 549) and <0.0001 (F = 420.9, DFn = 1, DFd = 497), respectively against the null hypothesis of zero-slopes and a p-value of 0.0317 (F = 4.625, DFn = 1, DFd = 1046) when comparing the slopes of force vs velocity; and *p* values < 0.0001 (F = 1819, DFn = 1, DFd = 560) and <0.0001 (F = 1137, DFn = 1, DFd = 492) respectively against the null hypothesis of zero-slopes and a *p* value of 0.0512 (F = 3.810, DFn = 1, DFd = 1052) when comparing the slopes of power vs velocity. **d** Spider plot providing a summary of the comparison of all measured parameters at day 20, normalized to 10 kPa in M16lac medium. The color shading groups parameters related to: green: cell morphology, blue: contractile stress, red: contractile dynamics, orange: temporality (see Figure S10). Line thickness is proportional to the standard deviation. Individual pairs of parameters were tested for significant differences using a two-tailed unpaired parametric t-test with confidence interval of 95%. Statistics show: M16gluc vs. M16lac on day 20 and 10 kPa substrates. Sample number n is provided in the figure panel. For all panels, *p < 0.05, **p < 0.005, ***p < 0.001, ****p < 0.0001. ns, not significant.

---

changes in three parameters: contractile force, cell spread area and beating frequency, revealing these parameters as key differentiator of contractile maturation.

We distinguished distinct contractile phenotypes using undirected clustering. Of the resulting clusters (Fig. 6a), clusters 1, 6 and 8 comprised the smallest cells exerting the least force, while clusters 2, 4, and 7 comprised the largest cells producing the most force (Fig. 6c). Cluster 5 comprised cells with the least distinct morphology and contractile profile, i.e., a grey zone of the cells most difficult to classify. Cluster 7 comprised the cells with a beating frequency smaller that the 1 Hz electrical stimulation imposed through external electrodes and displaying a prolonged contraction duration, while cluster 3 comprised the cells with high beating frequency that also displayed a low force production.

Strikingly, clustering identified different contractile phenotypes for the cells cultured on the 10 kPa or 35 kPa substrate stiffnesses, without a priori knowledge of the experimental conditions (Fig. 6d). In line with our earlier observations that stiffness, but not culture medium, impacted contractile function, clustering did not segregate cells cultured in different culture media (Fig. 6d). Clustering revealed

phenotypic differences linked to cell morphology. As expected, the cells with the largest spread areas generally produced the highest force, but, surprisingly, cells with the highest aspect ratios did not necessarily produce a high force. Clustering revealed a phenotypic difference linked to contractile dynamics. Differences in beating frequency and contraction duration resulted in two somewhat opposite phenotypes: one displaying a low force and high beat rate (cluster 3) and another displaying a slow beat rate and prolonged contraction duration (cluster 7). Interestingly, the 1-Hz electrical stimulation imposed through external electrodes was most accurately followed by cells with high aspect ratios, a behavior that could be linked to elongated patterning driving faster expression and function of beta-adrenergic receptors[50]. Clustering also revealed that the cells producing a particularly high force also displayed a high strain energy; these cells also displayed a high contraction velocity and were mostly cells cultured on softer 10 kPa substrates. In contrast, other cells that contracted with a high force but a slow contraction velocity were predominantly large cells cultured on stiffer 35 kPa substrates.

We observed converging phenotypes in lactate-rich medium. By exploiting CONTRAX's ability to follow single cells over time, we

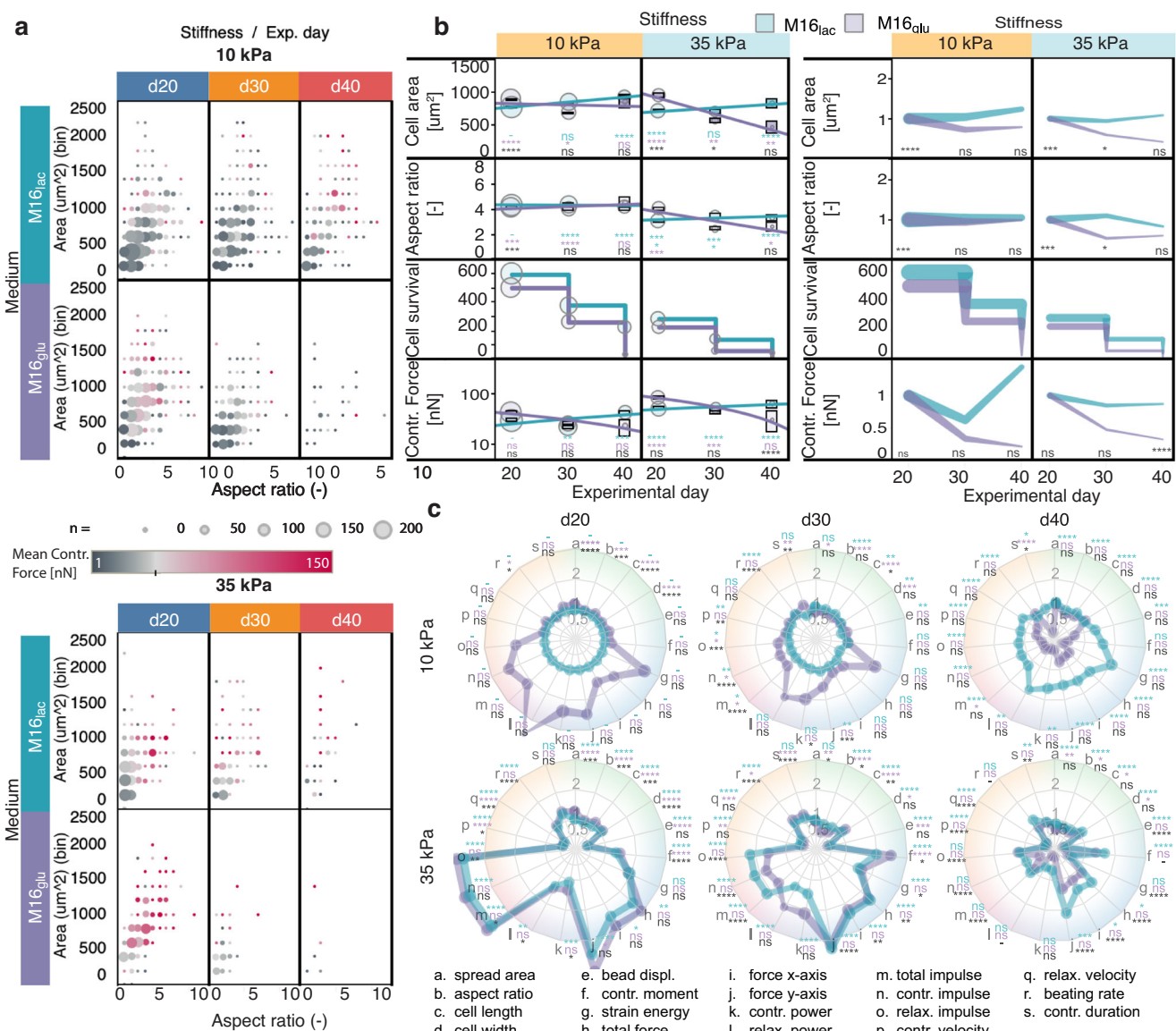

a. spread area    e. bead displ.      i. force x-axis      m. total impulse    q. relax. velocity
b. aspect ratio   f. contr. moment    j. force y-axis      n. contr. impulse   r. beating rate
c. cell length    g. strain energy    k. contr. power      o. relax. impulse   s. contr. duration
d. cell width     h. total force      l. relax. power      p. contr. velocity

**Fig. 5 | The contractile function in micropatterned hiPSC-CMs better develops over time in M16_lac versus M16_glu medium. a** Balloon plot showing that cells die sooner in M16_glu than in M16_lac medium, notably on stiffer 35 kPa substrate. Cells in M16_lac tend to increase in size. **b** CONTRAX measured changes in multiple parameters over 20 days, for both medium conditions and substrate stiffnesses. Left: Absolute value of the parameters at each time points with circle size proportional to the number of cells measured; the vertical bars showing the standard error; and the lines showing linear regressions. Linear regression models were tested against the null hypothesis of zero-slopes and compared using an F test using a 95% confidence interval: Cell area: $p$ value < 0.0001 against the null hypothesis and $p$ value < 0.0001 for the comparisons for both stiffnesses; Aspect ratio: $p$ value = 0.1235 and 0.7644, for each medium respectively, against null hypothesis and $p$ value < 0.1702 for the comparisons for 10 kPa stiffness, and $p$ value < 0.0001 against null hypothesis and $p$-value < 0.0001 for the comparisons for 35 kPa stiffnesses; Contr. Force: $p$ value < 0.0001 against null hypothesis and $p$ value < 0.0001

for the comparisons for both stiffnesses. Right: Value of the parameters normalized to d20 showing divergence over time as function of media composition. Line thickness represent the number of samples. See Supplementary Information Figure S4 for more parameters. **c** Spider plots providing a summary of the comparison of all measured parameters over time, normalized to control 10 kPa in M16_lac medium. The color shading groups parameters related to: green: cell morphology, blue: contractile stress, red: contractile dynamics, orange: temporality (see Figure S10). Line thickness is proportional to the standard deviation. Statistics show: turquoise: M16_lac vs. control M16_lac - d20 – 10kPa; purple: M16_gluc vs. control M16_lac - d20 – 10kPa; black: M16_lac vs. M16_gluc. In panel **b** and **c**, individual time point measurement data were compared with the corresponding control or corresponding condition for comparison using one-way ANOVA with Sidak's multiple comparison test with individual variance computed for each comparison. For all panels, \*$p$ < 0.05, \*\*$p$ < 0.005, \*\*\*$p$ < 0.001, \*\*\*\*$p$ < 0.0001. ns, not significant. See Figure S4, S5 and Table S3 for detailed data.

examined the maturation trajectories of the contractile phenotype in individual cells (Fig. 6b). Notably, the cells in our control group (*i.e.*, cultured in M16_lac at 10 kPa substrate stiffness) converged from relatively heterogeneous contractile phenotypes at d20 (cluster 5) to more homogeneous phenotypes with a larger contractile force (cluster 2). This single-cell resolution analysis confirmed the consistent better contractile performance over time across individual cells in our control group (Fig. 5b).

**Subpopulation analysis reveals potential gating strategies**
We questioned whether our large dataset and clustering analysis could help identify gating strategies. Such gating would indeed be very helpful in defining robust and applicable quality standards that could be applied across academic studies and industrial development of cell models and therapies. Here, we divided our dataset in four quadrants based on cell morphologies by defining two gates at the median values of aspect ratio (3.88:1 L:W) and spread area

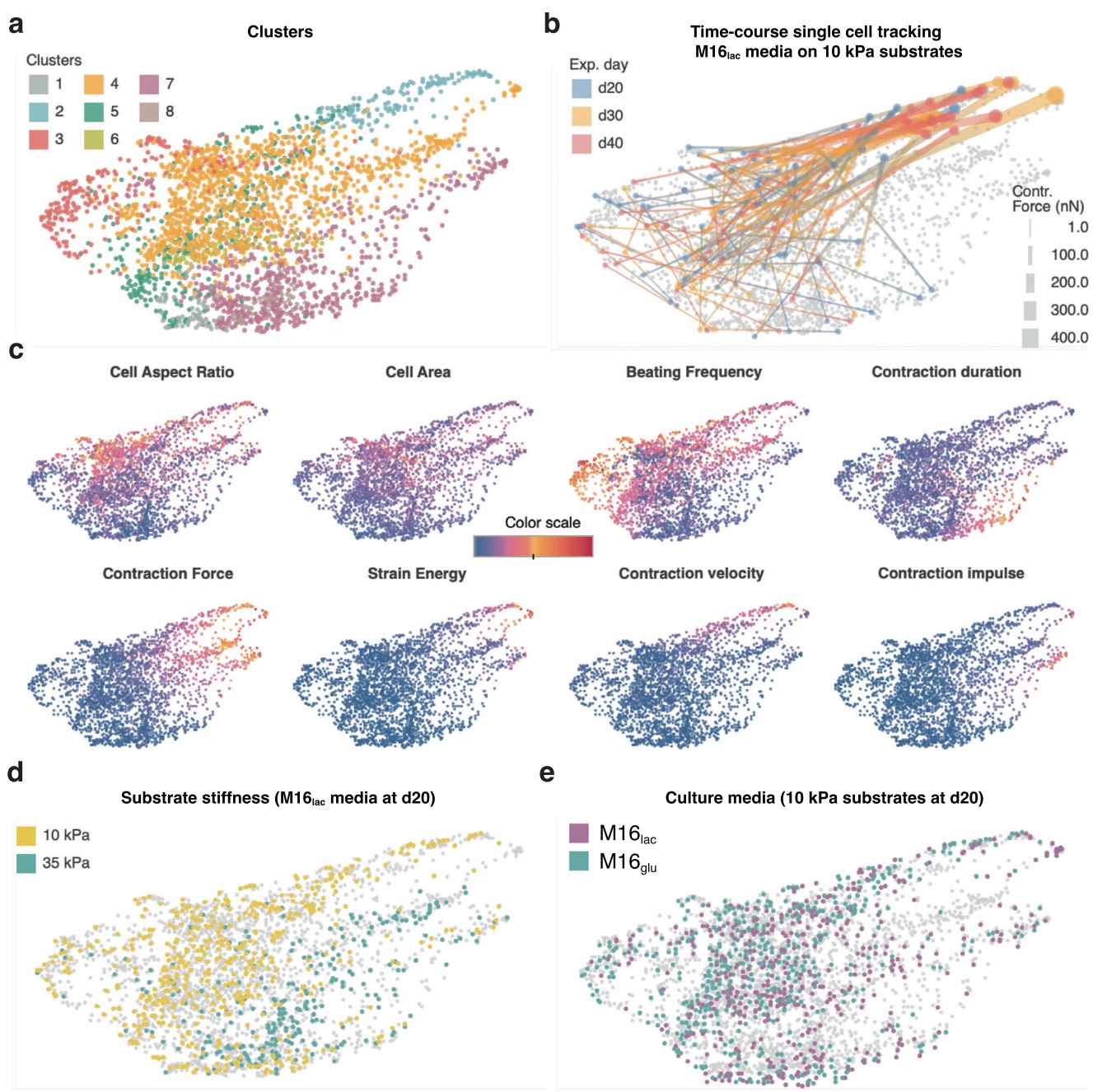

**Fig. 6 | Our large dataset enables cluster analysis, revealing the phenotypic complexity at the single-cell level. a** High-dimensional clustering by X-shift K-nearest-neighbor density estimation (Methods) yields eight clusters. **b** Single hiPSC-CMs in M16$_{lac}$ on 10 kPa substrates (control group) over time. Lines connect the different measurement time points for single cells, the colors correspond to experimental day, the line width correspond to contractile force. **c** Distinct distributions of the clustering parameters reveal the characteristics of each cluster. **d** At day 20, control hiPSC-CMs grown in M16$_{lac}$ mainly segregate into clusters 2 and 5 (10 kPa) and into clusters 4 and 8 (35 kPa). **e** In contrast, control cells at day 20 do not segregate into clusters based on medium composition.

(median = 704 μm$^2$) in our control cell group (*i.e.*, cultured in M16$_{lac}$ on 10 kPa substrates at d20) (Fig. 7a). Strikingly, a comparison of each quadrant with the clustering revealed that the cells in quadrant 1 almost entirely correspond to those in clusters 2 and 4, the two clusters segregating for phenotypes displaying a large contractile force (Fig. 7b); and > 80% of the cells in our control cell group had a morphology commensurate with quadrant 1. Interestingly, these cells constituted the bulk of the cells with a converging maturation trajectory (see Fig. 7c). Such gating could serve as a quality control or standardization strategy to select cell populations for drug safety and toxicology assessment.

We further examined how the trend over time of the mean contractile force and spread area differed in each quadrant (Fig. 7d). Qualitatively, the trends are generally preserved in each quadrant, with improving contractile function in lactate-rich medium compared to glucose-rich medium (Fig. 5b).

### Drug experiment: Mavacamten treatment reduces the contractile force

To demonstrate the ability to analyze the same cells before and after a drug treatment, we analyzed the response of single hiPSC-CMs to Mavacamten, a cardiac myosin inhibitor recently FDA-approved to

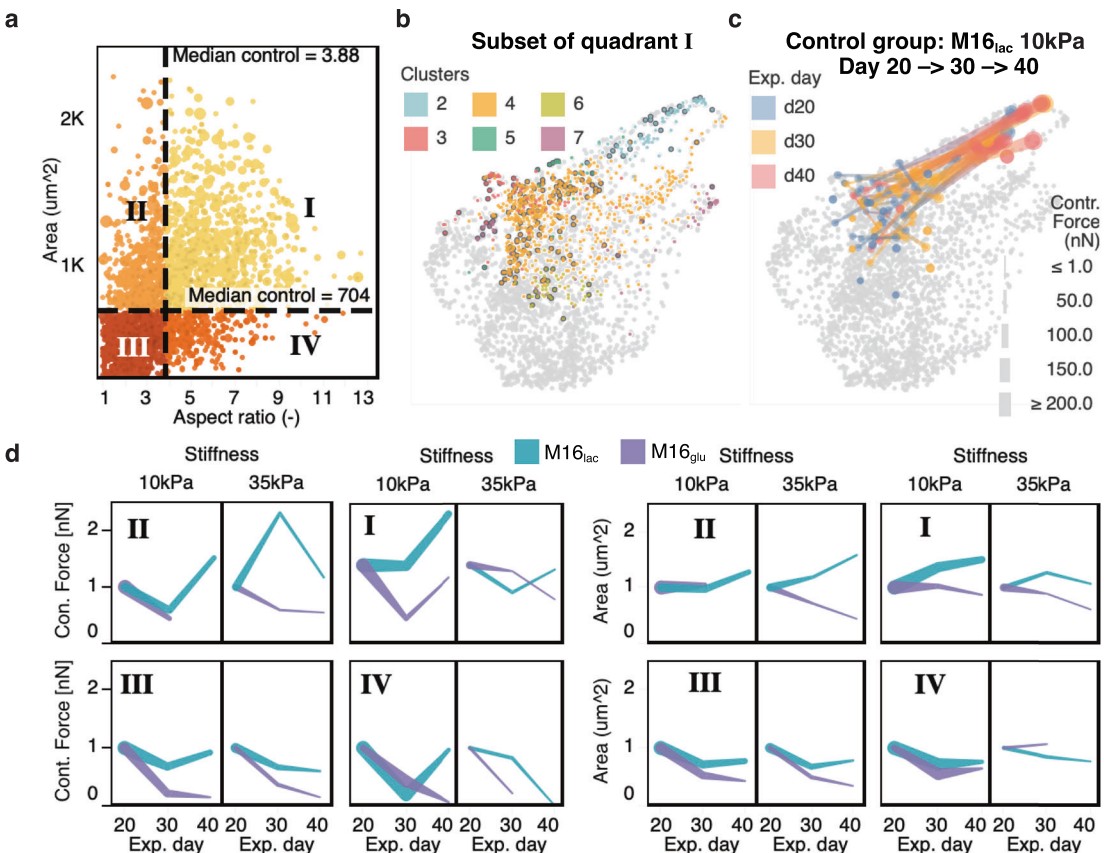

**Fig. 7 | Gating for subpopulations based on cell morphology correlates with difference phenotypic profile. a** Similar to FACS analysis, gates were defined to defined four subpopulations based on cell morphology. The median area and aspect ratio of control cells (M16_lac on 10 kPa at day 20) were used as gate values. **b** Cells in quadrant I correspond closely to cluster 2 and 4 (Fig. 6a). Control cells are marked with dark borders. Cells in clusters 1, 3, 5, and 7 are almost entirely absent from quadrant I. **c** Over time, cells in quadrant I progress toward a more homogeneous contractile phenotype. **d** Day 20-normalized force and area for each quadrant. Despite differences in cell morphology for each quadrant, the normalize contractile force display the best evolution and increase over time in quadrant I and II in the M16_lac medium on 10 kPa substrates. Line thickness is proportional to the sample number.

treat obstructive hypertrophic cardiomyopathy in the United States. Mavacamten previously demonstrated to reduce contractile force in wild type hiPSC-CMs (WTC) as in human cardiac tissue[51], a 35-57% reduction in contractility in 2D monolayer hiPSC-CMs after Mavacamten (0.3-1.0uM) treatment using an image based software inferring contractility from cell shortening measurements[52], and isolated murine cardiomyocytes[53,54].

We subjected patterned (1500 um²) single hiPSC-CMs cultured on 10 kPa substrates to treatment with 0.05 μM Mavacamten while measuring the contractile phenotype of the same single cells pre- and 1-hour post-treatment. We observed a consistent and significant reduction in contractile force and velocity under these conditions, despite an increase in cell area, confirming the potency of this drug treatment in reducing contractile force. (Fig. 8a, b). The effect of Mavacamten on the contractile output is summarize in a spider plot in Fig. 8c (Figure S8 & Table S4), where the measured parameters normalized to control clearly reveal how many of the various contractile parameter decrease post-treatment.

**Mutation experiment: DMD mutation severely affect contractility**

To demonstrate the value to observe distributions of morphological and contractile phenotypes across cell populations, we studied the impact of disease mutations using multiple cell lines with or without mutations in the dystrophin gene leading to Duchenne Muscular Dystrophy (DMD)-linked dilated cardiomyopathy. DMD is a deadly heritable disease affecting >3000 children per year, culminating with a

dilated cardiomyopathy. Interestingly, cardiac symptoms are delayed compared to skeletal muscle, but rapidly spiral towards cardiac failure, possibly due to a stiffening of the myocardium due to fibrosis[21]. We measured >1800 single hiPSCs with or with DMD mutation from multiple cell lines, including isogenic lines with CRISPR-induced/corrected mutations (Table S6). DMD mutations resulted in a general degradation of the contractile function, an effect that was notably more pronounced on stiffer fibrotic-like 35 kPa substrates compared to healthy 10 kPa substrates (Fig. 9a, b and Figure S9). Normalized comparison of the average parameters for these groups revealed minimal changes in cell morphology and small changes in most contractile parameters on 10kPa gels (with the largest changes related to power), but these changes are significantly magnified on stiffer 35kPa gels, as clearly visible in the spider plot summary in Fig. 9c (Figure S9 & Table S5).

## Discussion

CONTRAX enables the acquisition of large phenotypic datasets with single cell resolution (Fig. 1). Compared to previous approaches, CONTRAX's streamlined workflow and the order-of-magnitude improvement in data processing performance enable assaying hundreds to thousands of single cells within a couple of days, compared to multiple weeks or months, substantially shortening the time-to-results. Thus, CONTRAX facilitates the integration of quantitative TFM assays. It supports research workflows to characterize morphological and contractile phenotypes and makes such assays more accessible to the broader community. Further, CONTRAX's interlocking modules afford

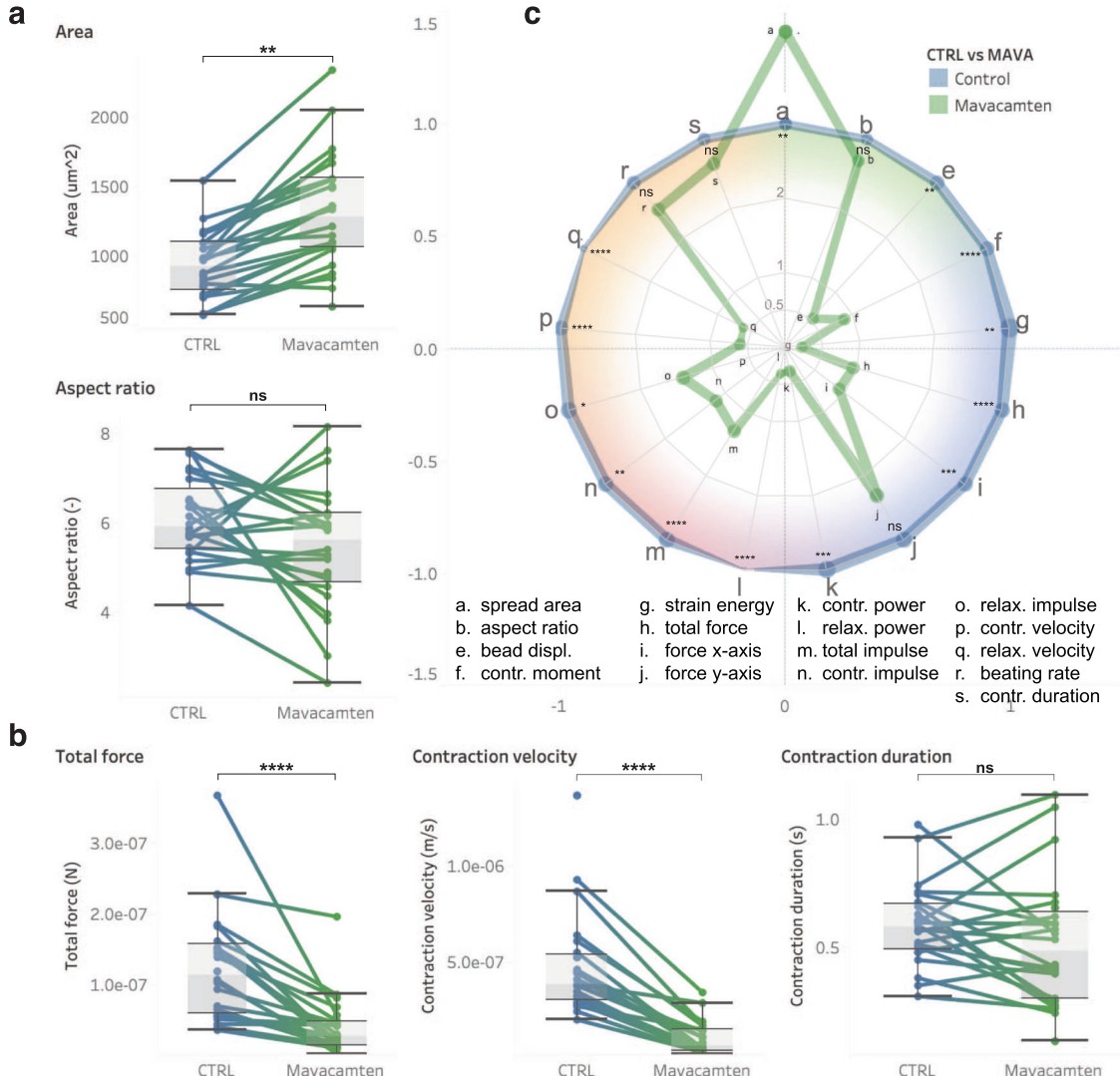

**Fig. 8 | Mavacamten treatment affects contractile function in single hiPSC-CM within one hour. a** Bar plots showing a consistent increase single cells area and no significant change in cell aspect ratio upon treatment. **b** Bar plots showing a consistent decrease in total force and contraction velocity in single cells but no change in contraction duration upon treatment. **c** Spider plot providing a summary of the comparison of all measured parameters normalized to control, pre-Mavacamten treatment. The color shading groups parameters related to: green: cell morphology, blue: contractile stress, red: contractile dynamics, orange: temporality (see Figure S10) Statistics show: Mavacamten 1 h post-treatment vs. control on 10 kPa substrates. In all panel, sample number n = 24 single cells for CTRL and 24 Mavacamten treatment. Boxplots show median value and the upper and lower quartile (25th and 75th percentile respectively), and the whiskers indicate data within 1.5 times the interquartile range. For all panels, data were compared with multiple parametric t-test with 95% confidence interval corrected for multiple comparison using Holm-Sidak method. *$p < 0.05$, **$p < 0.005$, ***$p < 0.001$, ****$p < 0.0001$. ns, not significant; n = 24. See Figure S8 and Table S4 for detailed data.

the ability to track changes in single cells (cardiomyocytes or other cell types as well) over time, a feature that will benefit many longitudinal studies (Fig. 6b).

Micropatterning constrains cell morphology into physiological elongated cardiomyocyte shapes (Fig. 2b) and, thereby, enables reduced hiPSC-CMs morphological heterogeneity. Micropatterning contributes to the development of a better developed phenotype, with increasing force with larger aspect ratio (Figs. 3 and 4) or area (Figure S1)[12]. The micropattern dimensions (1:7 aspect ratio and 2500 um²) were intentionally chosen to be larger than that for reported hiPSC-CMs spread areas[12,25] to enable characterizing the range of cell morphology found in heterogenous hiPSC-CM populations[50]. As a result, while the variance in cell morphology was large, the cell area scales proportionally with aspect ratio, even if the slope was not as steep as could be expected from the micropattern dimensions. This suggests that hiPSC-CMs favor elongation along the micropatterns over lateral spreading (Fig. 2b)[55]. Despite the physical constraints imposed

by patterns, this size enabled observing a significant increase in cell size upon Mavacamten treatment (Fig. 8 & Figure S8) and a decrease in cell sizes with disease-causing mutations (Fig. 9 & Figure S9)[21,22].

While micropatterning does contribute to reducing the heterogeneity found in hiPSC-CMs morphology, micropatterning is not a strict requirement for CONTRAX. To the contrary, the gating strategy enabled by the *Cell Locator* module enables to rapidly identify elongated cells on substates with or without micropatterns, which potentially alleviates the need to perform substrate micropatterning, a step that presents its own challenges despite the various methods now available[12,41].

Substrate stiffness and maturation time impacted cell morphology and contractile phenotype. Stiffer 35 kPa substrates resulted in the development of wider but not smaller hiPSC-CMs over time (Fig. 2c, Figure S1.a-b). This suggests that an increased stiffness and prolong maturation time contribute to promoting cell growth, similar to that reported in vivo in compensatory hypertrophy[56]. Stiffer 35 kPa

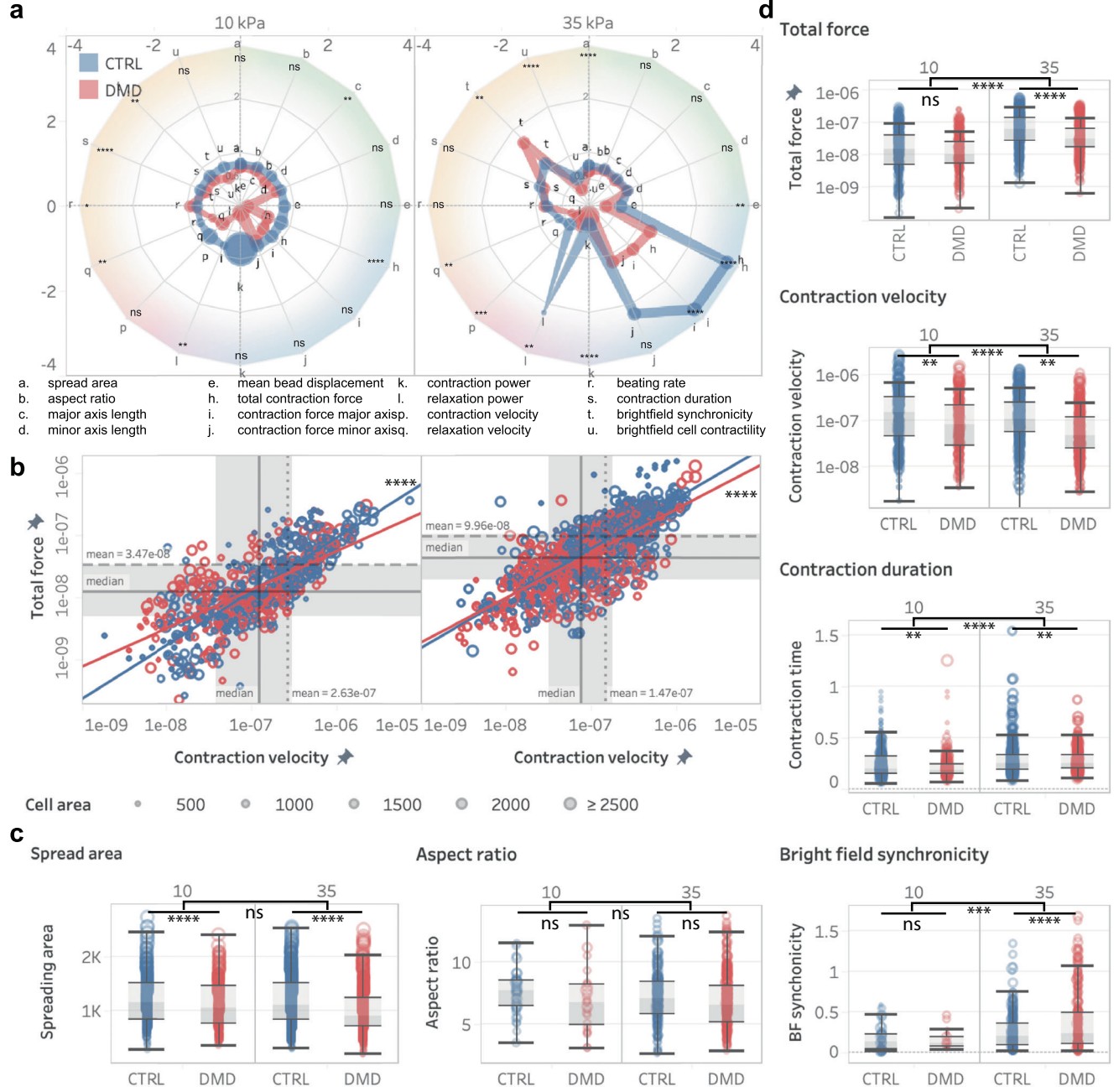

**Fig. 9 | Increased substrate stiffness exacerbated the effect of dilated cardio-myopathy causing Duchenne Muscular Dystrophy (DMD)-mutation in hiPSC-CMs. a** Spider plots providing a summary of the comparison of all measured parameters normalized control hiPSCs on 10 kPa substrates. The color shading groups parameters related to: green: cell morphology, blue: contractile stress, red: contractile dynamics, orange: temporality (see Figure S10). Statistics show: DMD vs. control healthy hiPSC-CMs on 10 kPa and 35 kPa substrates. **b** Log-log plot of the contraction force and contraction power versus contraction velocity with square-law regression, showing a significant reduction in the slope for DMD vs control. **c** Bar plots showing a decrease in single cell of the area, but not in aspect ratio.

Sample number n is 35 for CTRL-10kPa, 24 for DMD-10kPa, 177 for CTRL-35 kPa and 282 for DMD-35 kPa, analyzed in at least 3 independent experiments. **d** Bar plots showing that increased substrate stiffness leads to a dramatic loss of total force and synchronicity on stiffer fibrotic-like 35 kPa substrates, but not on healthy 10 kPa substrates, and a loss in contraction velocity, contraction duration in DMD-hiPSCs compared to control hiPSCs. Sample number is the same as for panel **c**. For all panels, data were compared using ordinary two-way ANOVA using Tukey's multiple comparison test with single pooled variance and a 95% confidence interval, *$p < 0.05$, **$p < 0.005$, ***$p < 0.001$, ****$p < 0.0001$. ns, not significant. See Figure S9 and Table S5 for detailed data.

substrates also led to a higher contractile stress (Fig. 3b), an expected physiological response to increased afterload, consistent with previous findings using other in vitro models[14,27,32,57,58]. Notably, an increased stiffness also affected the contractile performance over time and resulted in faster cell loss or death, revealing a possible link between mechanical stress and loss of contractile function in CMs (Fig. 5b and Figures S5 & S6). Additionally, increased stiffness affected the hiPSC-CMs electrical paceability or beating synchronicity, with a

mean beating rate higher than the 1 Hz pacing frequency on 35 kPa substrates. Taken together, these findings suggest an important role of increased tissue stiffness in displacing cardiomyocyte contractile homeostasis, similar to that observed in vivo in tissue undergoing fibrosis, as in dilated cardiomyopathy, including DMD (Fig. 9 & Figure S9)[21,32,33].

A metabolic switch promotes the maturation of contractile phenotype. At d20, cells cultured in M16$_{glu}$ medium produced more force

than control cells (M16$_{lac}$ on 10 kPa at d20) (Fig. 4), reflecting the favored glucose metabolism during early development. Over time, however, we observed a dramatic change: the contractile function of cells cultured in M16$_{glu}$ declined with a significant increase in cell death. In contrast, control cells cultured in lactate-rich M16$_{lac}$ displayed a slight but consistent increase in most contractile parameters (Fig. 5 and Figures S5 & S6). This behavior appears to be consistent with the known metabolic switch associated with CM maturation during development[36,45,46]. Accordingly, we observed an increased in mitochondrial respiration in control M16$_{lac}$ cells compared to M16$_{glu}$ in Seahorse assays (Figure S2).

Undirected clustering revealed distinct patterns in maturation trajectories. Our multidimensional clustering analysis (Fig. 6) provided insight into the diverse contractile phenotypes found in hiPSC-CM populations in vitro. Our analysis demonstrated distinct stiffness-dependent phenotypes that emerged over time: a high force and strain energy corresponds to cell with high contraction velocity on softer 10 kPa substrates whereas, on stiffer 35 kPa substrates, high force corresponds to predominantly larger cells with slower contraction velocity. Further, the larger and slower-contracting cells on 35 kPa substrates displayed a high contraction impulse (the area under the curve of force versus time), a metric that has been suggested as a potential indicator of cardiomyocyte hypertrophy in vivo and in vitro[32]. This demonstrates how CONTRAX's ability to measure such phenotypes and maturation trajectories has the potential, together with emerging in vitro models of cardiac diseases, to provide insights that can be better mapped to in vivo observations. For instance, in vivo, larger cells are characteristic of hypertrophic cardiomyopathy (characterized clinically by hyper-contractility), and cells of high aspect ratios are typical of dilated cardiomyopathy, such as DMD, characterized clinically by hypocontractility[59]. In vivo, chamber specificity (atrial versus ventricular) and deficient excitation-contraction coupling have also been linked to differences in contractile dynamics and excitability at the cardiomyocyte level[60,61]. Further, our observations of a, possible allometric, power-law relation between contractile force (or power) and contraction velocity is resembling the well-known in vivo force-velocity relationship in the heart and cardiomyocytes that links a higher afterload to a lower contraction velocity (Figs. 3c & 4c)[62–67].

Unbiased gating offers the potential to select for specific cell subpopulations. In drug testing, or in developmental and remodeling studies, it is important to include a maximum of cells of relevance, for example those most likely to reach specific maturity targets (e.g., minimum area, elongation, contractile function). CONTRAX enables applying such selection gates using the *Cell-Locator* module and pre-select subpopulations based on morphological criteria upstream of TFM data acquisition and analysis. Application of such selection gates is rapid and repeatable in contrast to current approaches that rely on manual and subjective cell selection; this process further increases the yield in CONTRAX and the overall specificity of the assay. In our study, selecting for more elongated and larger cells (quadrant 1) reduced some of the intrinsic heterogeneity (Fig. 7). Nevertheless, we recommend careful definition and up-stream application of morphological selection gates, as these choices can lead to inadvertent exclusion of certain phenotypes of interest, such as high contractile frequency (cluster 3) or hypercontractile phenotypes (cluster 7) of particular importance for certain longitudinal drug or disease studies.

CONTRAX reveals the effect of drugs and genetic mutations on hiPSC-CMs contractility. The results of our Mavacamten treatment (Fig. 8 & Figure S8) and DMD disease mutation (Fig. 9 & Figure S9) studies confirm the potency to apply CONTRAX to clinically relevant biological studies and yielded potentially important findings. The integration of the *cell locator* and acquisition modules improves the ease of rapid repeated imaging of a high number of cells pre- and post-drug treatment, which is important to capture the dynamics of specific compounds and to address the intrinsic variability within populations

of hiPSC-CMs. CONTRAX also enabled a high throughput analysis of cells with disease-causing mutations from multiple cell lines, which is instrumental in the consideration of patient-to-patient variability. As such, CONTRAX enabled a large study of the impact of mutations on the mechanobiology, telomere shortening and cardiomyocyte dysfunction in the context of DMD[21].

Overall, the ability to compare combinations of factors (cell genotypes and substrate stiffness) has important implications for investigating a complex mechanical system like the heart, and we expect that further use of the longitudinal tracking of cells enabled by this approach will also provide insights into the progression of disease phenotypes. We believe that our approach with CONTRAX may contribute to addressing the challenge of defining robust and applicable quality standards for academic studies and industrial development of cell models and therapies.

While we believe CONTRAX is a highly relevant approach, there remain some limitations. CONTRAX performs traction force microscopy measurement using 2D image videos. Recently, traction force microscopy has also been made possible using 3D images through advances in confocal imaging and computing power[30,31]. 3D TFM has the advantage of being more comprehensive and accurate in that it encompasses traction stresses occurring out-of-plane, something that is not possible with 2D images, and allows to consider more complex 3D in vitro model. Nevertheless, 3D imaging possesses limited time resolution due to the plane-by-plane imaging modality currently used, making this technique difficult to apply to fast-contracting cells such as cardiomyocytes and to reach higher throughput. Further, the sheer volume of data to process under such conditions makes the computing requirement another challenging aspect. Another potential concern exists in the implementation of the workflow in extended longitudinal studies regarding the stability of the mechanical properties of the hydrogel substrates and of the ECM micropatterns, and appropriate controlled must be used.

In summary, CONTRAX combines an innovative approach with three interlocking software modules for multiparameter quantification of the contractile function of single hiPSC-CMs over time. Our analysis demonstrates the power of CONTRAX in tracking changes in contractile phenotypes with single cell resolution over time. CONTRAX is a powerful tool that will benefit longitudinal studies of cardiac contractile deficiencies and contribute to the development of cardiac therapies and personalized medicines.

## Methods

Supplementary Methods appear in the online Supplementary information.

### Ethical statement
All protocols using hiPSC were reviewed and approved by the Stanford Stem Cell Research Oversight committee.

**Fabrication of micropatterned hydrogel substrates.** To manufacture our microphysiological assay substrates, we adapted previously published procedures[12]. In short, polydimethylsiloxane (PDMS, Sylgard® 184 Silicon Elastomer Kit, Dow Corning Corporation) was molded onto a microfabricated SU8-silicon mold by mixing the PDMS kit components at a 1:10 ratio of curing agent:prepolymer using a Thinky mixer, desiccating for 60 min using house vacuum, and curing at 60 ˚C for at least 60 min to obtain 1-cm$^2$ stamps able to print >4000 patterns of 132.3 μm x 18.9 μm $\cong$ 2500 μm$^2$ and aspect ratio 7:1, interspaced with 200 μm. Stamps (prechilled at 4 ˚C) were incubated at 4 ˚C overnight with Matrigel extracellular matrix (ECM) proteins diluted 1:10 in L-15 medium (Leibovitz), rinsed once with L15 medium, and dried under nitrogen flow. ECM protein micropatterns were microcontact-printed onto cleaned (O$_2$ plasma, 80 W, 60 s) 15-mm coverslips by gently placing the Matrigel-coated stamp face down onto the glass, applying a

constant pressure with a 50-g weight for 3 min, leaving the device without weight for another 2 min, then separating the coverslip from the stamp using forceps.

Polyacrylamide gel precursor solutions were prepared by mixing acrylamide (CAS # 79-06-1, Sigma) (10% w/v), *N,N′*-methylenebisacrylamide (CAS # 110-26-9, Sigma) (0.1% or 0.3% w/v for 10 kPa and 35 kPa, respectively), 3 mM HEPES (Life Technologies), 0.2 μm yellow-green or red fluorescent FluoSpheres™ Carboxylate-Modified Microspheres (#F8813, Invitrogen) (2.16% w/v to yield a final concentration of ~6 × 10$^9$ microbeads/mL), and Milli-Q water and desiccating the solution for 30-60 min in vacuum[12,43]. These polyacrylamide formulation were measured to maintain good mechanical properties for the duration of our experiments. While the solution was desiccating, glass-bottom 6-well plates (MatTek Corp.) were silanized with 0.3% 3-(trimethoxysilyl) propyl methacrylate (CAS # 2530-85-0, Sigma) in 200-proof ethanol, adjusted to pH ~3.5 with 5% acetic acid glacial for 5 min, washed twice with 200-proof ethanol, and dried with N$_2$ gas. Polyacrylamide hydrogels were formed by adding ammonium persulfate (Sigma) (0.1% w/v) and *N,N,N′,N′*-tetramethylethylenediamine (Sigma) (0.1% v/v) to the precursor solution, rapidly transferring 35 μL onto a silane-treated glass-bottom well, and placing a microcontact-printed coverslip face down onto the polymerizing gel precursor. The use of 6-well (or larger) plates enables optimal repositioning of the plate onto the microscope stage for acquisition of the same single cell during time-course experiments. In order to obtain reproducible hydrogel substrates, PDMS or Kapton® spacers of the desired thickness were placed between the bottom glass and the microcontact-printed coverslip to obtain a controlled hydrogel thickness of at least 100 μm. At smaller thicknesses, the human induced pluripotent stem cell-cardiomyocytes (hiPSC-CMs) would feel the rigid glass bottom, yielding inaccurate measurements of traction force. After 30 min for polymerization at room temperature, the wells were flooded with phosphate-buffered saline (PBS; pH 7.2, Gibco) plus 1:100 penicillin-streptomycin (10,000 U/mL) (#15140122, Thermo Fisher Scientific) and incubated overnight at 37 °C. Next, the top coverslip was removed to reveal the ECM micropattern transferred into the top surface of the hydrogel. To verify the transfer of the micropatterns, we spiked fluorescently labelled gelatin (Alexa fluor 488) at 1 μg/ml in the Matrigel and imaged using the corresponding channel (see Figs. 1b and 2a). This was only done for validation experiments and not for all devices used for TFM imaging. The PBS was aspirated, and hiPSC-CMs were seeded onto the micropatterned hydrogel.

**Culture, passaging, and differentiation of hiPSC-CMs.** Here we used hiPSC-CMs as an in vitro model of human cardiac biology. For the longitudinal and drug experiments, we used WTC-11 hiPSCs, a widely used and now commercially available that has been deep sequenced with no known mutation, that was obtained as a generous gift from collaborators[36–38] and can be purchased through the Coriell Institute Biobank (GM26256). For the DMD study, we used the lines described in Table S6.

hiPSCs from the various lines were grown on Matrigel-coated (Corning, 356231) plates using Nutristem medium. All protocols using hiPSC were reviewed and approved by the Stanford Stem Cell Research Oversight committee (#602). The medium was changed daily, and cells were passaged every 4–6 days using Accutase (Sigma, A6964) and seeded in 1:4 or 1:8 dilution with the addition of 5 mM Y-27632 2HCl (Selleck Chem, S1049). hiPSCs were grown to 70–90% confluence and subsequently differentiated into beating cardiomyocytes, as demonstrated previously[68].

WTC hiPSCs were cultured in Gibco™ Essential 8™ Medium (#A1517001, Thermo Fisher Scientific) supplemented with E8 supplement with daily medium change with Falcon multi-well (6 or 12) culture plates (#353046, #353043, Corning). These plates were previously incubated for at least 1 h with Corning™ Matrigel™ GFR

Membrane Matrix (#CB-40230, Fischer Scientific) diluted 1:100 in Leibovitz's L-15 Medium (#11415064, Thermo Fisher Scientific) at 4 °C. hiPSCs were passaged at 1:30 dilution every 3–5 days or before reaching 90% confluency using 0.5 mM ethylenediaminetetraacetic acid (EDTA) in PBS pH 7.2 (#10010023, Thermo Fisher Scientific) or Accutase solution (#A6964, Sigma-Aldrich). Before passaging hiPSCs, the Matrigel incubation solution was aspirated from a new culture plate and cells were passaged into E8 medium supplemented with 5 μM Rock inhibitor (#Y-27632, StemCell Technologies).

hiPSCs were differentiated into beating CMs according to established protocols[15,21,69]. When hiPSCs reached ~60-80% confluency (day 0), E8 culture medium was replaced with RPMI-1640 culture medium containing L-glutamine and glucose (#11875093, Thermo Fisher Scientific) supplemented with B-27™ Supplement minus insulin (#A1895601, Thermo Fisher Scientific) and 6 μM GSK-3α/β inhibitor CHIR-99021 (Chir, #S2924, SelleckChem). Forty-eight hours after the start of differentiation, the culture medium was replaced with fresh medium in which Chir was replaced with 2 μM Wnt-C59 (C59, #S7037, SelleckChem), a PORCN inhibitor for Wnt3A-mediated activation. On day 4, the medium was exchanged again with fresh RPMI-1640 without any inhibitor or activator. On day 6, the medium was exchanged again with fresh RPMI-1640 with L-glutamine and glucose (#11875093, Thermo Fisher Scientific) supplemented with B-27™ Supplement (50X) and without serum (#17504001, Thermo Fischer Scientific); the medium was exchanged again on day 8 with the same medium.

On days 10-12, robust beating of the cell monolayer was typically obtained (data not shown) and differentiated hiPSC-CMs were passaged at 1:4 or 1:6 into new 12-well plates freshly coated with Matrigel. Accutase (0.5 ml in 12-well plates, 1 ml in 6-well plates) was used to detach cells, then quenched with RPMI 1640 culture medium with L-glutamine and no glucose (#11879020, Thermo Fisher Scientific) supplemented with B-27™ Supplement (50X) without serum (#17504001, Thermo Fischer Scientific), 4 mM sodium DL-lactate solution (#L7900, Lot#LBR6294V, Sigma-Aldrich), and 5% Knockout Serum (#10828010, Thermo Scientific). This solution was spun for 3 min at 300 x *g* at room temperature, and cells were resuspended in 1 mL of the same medium supplemented with 5 μM Rock inhibitor (#Y-27632, StemCell Technologies). Cells were replated onto a new plate, and the medium was changed after 2 days to remove the Rock inhibitor. Spontaneous beating reoccurred soon after replating (data not shown).

Longitudinal study: WTC hiPSC-CMs were differentiated in five differentiation batches prepared from different passage numbers of WTC-iPSCs line (one differentiation batch from p37, one from p41, two from p46, one from p47)[15,35,36]. hiPSCs were differentiated into beating CMs according to established protocols (see below and Fig. 2a)[15,21,69]. At day 12 post-differentiation (d12), the hiPSC-CMs were subjected to 4 days of glucose starvation to purify against fibroblasts[70], then were replated onto freshly prepared Matrigel-coated tissue culture plates. At d16 ± 1 on average, half of the cultured wells were either switched to RPMI-1640 + B27 + D-glucose medium or maintained with RPMI-1640 + B27 + DL-lactate medium[15,36]. These media are called M16$_{glu}$ (glucose-rich) and M16$_{lac}$ (lactate-rich), respectively, in the rest of this report. On d17, the hiPSC-CMs were replated onto micropatterned hydrogel devices for TFM. Cells (~20 × 10$^5$ cells/cm$^2$) were replated using the medium in which they were last cultured plus 5 μM ROCK inhibitor (#Y-27632, StemCell Technologies) and 5% Gibco™ Knock-Out™ Serum Replacement (KSR, #10828010, ThermoFischer Scientific). Medium was replaced, without ROCK inhibitor and KSR, after 24 h and thereafter every 2 days. Cells were allowed to recover for at least 48 h after seeding before measurements. We only measured beating cardiomyocyte to alleviate for potential low efficiency of some differentiation batches.

Drug study: hiPSC-CMs were differentiated as detailed above from the WTC-iPSCs line. CMs were plated on day 33 on Matrigel patterned

(1500 um) hydrogels (10kPa) as previously mentioned. After two additional days of culture (d35), CONTRAX was used to detect and map single beating hiPSC-CMs (~270 cells were detected with the *Cell Locator* module). TFM was performed on 42 cells prior to Mavacamten treatment (DMSO vehicle, Paced 1 Hz). Warmed media containing 0.05 μM Mavacamten was then added to the cells and placed back in the incubator for 1 hour. TFM was performed on the same 42 cells after 1 hour treatment with Mavacamten under electrical stimulation pacing at 1 Hz. Cells that did not respond to electrical pacing in a range of 0.4-1.6hz and/or ceased beating due to drug treatment were removed. A total of 26 cells were analyzed and plotted.

Mutation experiment: All protocols using hiPSC were reviewed and approved by the Stan- ford Stem Cell Research Oversight committee (#602). HiPSCs were differentiated into beating cardiomyocytes, as demonstrated previously and detailed in Supplementary Information[15,35,36,68]. Beating hiPSC- CMs were purified against noncardiomyocytes and matured by culturing in glucose-free conditions using RPMI-1640 medium without glucose with B27 supplement and 5 mM lactate (Life Technologies) until day 30[26,36,70]. Micropatterned hydrogel devices of 10 kPa and 35 kPa stiffness were manufactured 1-day prior to passaging and seeding of day 25 hiPSC-CMs. Imaging and data acquisition were consistently performed 4-5 days after seeding the cells on the devices. HiPSC-CM derived from Isogenic pairs were always measured consecutively on the same day. HiPSC-CM derived from each cell line were measured in differentiation triplicates. HiPSC-CMs were passaged onto micropatterned polyacrylamide devices by using 4:1 Accutase:TrypLE™ for dissociation and RPMI-1640 B27+ glucose- lactate+ medium with 5 μM ROCK inhibitor and 10% Knock-out Serum (KSR) for resuspension. After counting, cells were deposited onto the 18 mm diameter hydrogels at a density of ~30,000 cells/cm² in a final 150 μl volume at a concentration of 200,000 cell/ml. After 30 min incubation, an additional 2 ml of medium was added. After overnight incubation, the medium was replaced with fresh RPMI-1640 B27+ glucose-lactate+ medium and replaced every 2 days thereafter.

**Live-cell microscopy.** We used a Leica DMi6000b epifluorescence microscope equipped with a motorized x-y stage, a motorized focus drive, motorized objectives, motorized filter cubes, a motorized condenser turret, a high-speed/sensitivity Photometrics PRIME 95B sCMOS camera, and a live-cell incubator enclosure to maintain optimal cell viability at 37 °C and 5% CO₂ supplied from a premixed air-CO₂ gas bottle (Praxair) during imaging. The free and open-source *Micro-Manager* (version 1.4.23 64 bit) software was used to control microscope acquisition. The multi-well plate format allowed for culturing cells over long periods in well-controlled conventional cell-culture incubators, while reproducibly repositioning assay substrates on the imaging platform for time-course measurements.

During acquisition, the hiPSC-CMs were subjected to electrical pacing using a single-channel Myopacer Cell Stimulator (IonOptix) and a custom single-well large-area carbon electrode (IonOptix), with a programmed bipolar pulse of 20 V at a frequency of 1 Hz. For video imaging, we used an acquisition cycle of either ~23 s/cell (autofocus, 7 s brightfield imaging, 7 s fluorescence imaging, and microscope control overhead), ~26 s/cell (autofocus, a brightfield snapshot, 7 s fluorescence imaging, and microscope control overhead—only a single bright field image is strictly required for cell outlining, not an entire movie). If hardware allows, synchronizing electrical pacing and video acquisition could allow to further study the excitation-contraction coupling for example.

**Seahorse assay.** hiPSC-CMs were seeded at 30,000 per well in Matrigel-coated Seahorse plates in M16$_{lac}$ or M16$_{glu}$ medium five days prior to assay. The oxygen consumption rate responses of cells were measured with the Seahorse Bioscience XF96 flux Analyzer following instruction in the XF cell Mito Stress Test Kit User Guider. All measurements of oxygen consumption rate (OCR) were acquired at 5-min intervals with 1-min mixing between intervals. Three baseline measurements were acquired followed by injection of oligomycin to a final concentration of 2.5 μM. After three measurements in the presence of oligomycin, FCCP was injected to a final concentration of 1 μM and three measurements were recorded. Last, rotenone and antimycin A were injected to a final concentration of 2 μM each, followed by three measurements. CMs were subjected to the same substrate conditions as for culture (M16$_{lac}$ or M16$_{glu}$ medium) but prepared with Phenol red-free unbuffered RPMI-1640 (Agilent) as basal medium instead of RPMI-1640 (Thermo Fisher Scientific). OCR was normalized to live cell count using a PrestoBlue dye in accordance with the manufacturer's protocol (10 min incubation at 37 °C) and quantified on a TECAN Pro1000 plate reader (Stanford High-Throughput Bioscience Center) at 560 nm excitation and 590 nm emission.

**Longitudinal experiments on single hiPSC-CMs.** To evaluate the power of CONTRAX, we designed a 2-dimensional time course experiment that aimed to reveal the effects of stiffness and of culture-medium composition on the contractile maturation of micropatterned hiPSC-CMs over 20 days (Fig. 2a). We employed two substrate stiffnesses, 10 kPa and 35 kPa, which correspond to healthy and fibrotic tissue, respectively[14]. For consistency, all hydrogel devices were prepared in batches two days before seeding d17 hiPSC-CMs on them, using the same stock solution of hydrogel precursors and reagents to minimize experimental variability. Micropatterns of ECM protein with a 2500 μm² rectangular area and a 7:1 aspect ratio were defined via microcontact printing and transferred onto the hydrogel via copolymerization, spatially confining single hiPSC-CMs into a physiologically relevant, elongated shape[12,32]. We designed sufficient distance ( > 200 μm) between the micropatterns to ensure adequate decoupling of cell from other nearby contracting cells[71]. We measured 3,366 single hiPSC-CMs across five differentiation batches and, after excluding 613 measurements due to cell doublets or unidentifiable contraction peaks (caused by a lack of contraction or low signal-to-noise ratio), our final dataset includes 2,753 single micropatterned hiPSC-CMs. For statistical comparison, our control group is that of cells cultured in M16$_{lac}$ media on substrates with 10 kPa stiffness at d20 (Fig. 2a).

On d20, d30, and d40, TFM videos of single cells were acquired using CONTRAX. For each cell batch, 10x-magnification surveys and 40X-magnification traction force videos were acquired for all conditions on the same day. Cells were electrically paced at 1 Hz using a bipolar pulse of 20 V and single-well carbon electrodes during video acquisition. A 5% CO₂, 37 °C, full stage enclosure incubation chamber was used on the microscope setup to ensure optimal environmental conditions.

At d20, we used CONTRAX's *Cell-Locator* module to locate single micropatterned hiPSC-CMs. On subsequent imaging days, we reused the position list from this initial device survey to automatically locate the same single cells and image them again. To use the *Cell-Locator* module, we acquired tiled image surveys of our micropatterned hiPSC-CMs on the TFM substrates using bright field microscopy (transmitted light) and a 10X, 0.3 NA air objective, using the built-in *Acquire Multiple Regions* plugin of *Micro-Manager* software v1.4 with autofocusing (Fig. 1, middle). The *Stage Position List* of the absolute stage XYZ coordinate for each tile was saved as a position file (.pos) file. Tiled images (.tif or.czi) and the corresponding position list were loaded into our *Cell-Locator GUI*. The software identified single cells were automatically based on user-defined selection criteria. For this study, we defined broad search criteria (cellular aspect ratio of 1.5:1 to 12:1 (length:width) and an area of 300-3500 μm²) to assay the heterogeneity of the hiPSC-CMs on our micropatterned substrate. The *Cell-Locator* software module outputs a position list of all the cells meeting these criteria. Details of the workflow in the *Cell-Locator* module are found in the online Supplementary Methods in Supplementary information.

Next, we used CONTRAX's *Automated TFM acquisition* module to automatically acquire TFM videos of these cells. We first calibrated the positions for any x-y-z offset resulting from differences between objectives or stage drift. We recorded 7-s videos using a 40X, 0.6 NA air objective, with a 3-ms exposure for the bright field channel and a 20-ms exposure for the fluorescence channel, using stream acquisition (the maximum camera speed for a given exposure). The dose of fluorescence illumination was minimized to limit phototoxicity. We applied a predefined crop factor of 600 ×300 pixels (no pixel binning). These parameters yielded an acquisition frame rate of 30 frames/s on our setup, although a higher frame rate may be possible and sometime advantageous. At each cell position, software autofocusing was performed using the *OughtaFocus* as *Autofocus properties* in *Micro-Manager*, or with the software autofocus in *Zen* (Zeiss) Settings were optimized for our experimental conditions (search range 40 μm; tolerance 1 μm; crop factor 0.5; exposure 1 ms; default lower and upper FFT cutoffs of 2.5% and 14%, respectively; maximizing for Edges detection).

Finally, we analyzed TFM videos using CONTRAX's *Streamlined-TFM* module. Our *Auto-draw* function enabled automated detection of cell outlines. We used the corresponding video parameters (frames per second = 30, pixel-to-micron conversion = 0.275) and applied our built-in cropping and 2-by-2 binning functions to reduce file size using our default 200 × 85 μm cropping window, leaving enough space around each cell to capture substrate deformation far from the cell itself. These is important for the accuracy of TFM analysis. We used the following displacement parameters: subset radius = 15 px, spacing coefficient = 5 px, cutoff = $10^6$, maximum iteration = 20. For computation of TFM via FTTC, the regularization parameter was automatically computed by the software using L-curve identification[72]. Contraction peaks are detected automatically using our built-in peak detection and averaging algorithm. Details of the *Streamlined-TFM* module workflow appear in the Supplementary Methods online in Supplementary information.

**Treatment of single hiPSC-CMs with myosin inhibitor drug**. hiPSC-CMs (D33) were plated on Matrigel patterned (1500um) hydrogels (10kPa) as previously mentioned. After two additional days of culture (D35), CONTRAX was used to detect and map single beating hiPSC-CMs (~270 cells were detected). TFM was performed on 42 cells prior to Mavacamten treatment (DMSO vehicle, Paced 1 Hz). Warmed media containing .05uM Mavacamten was then added to the cells and placed back in the incubator for 1 hour. TFM was performed on the same 42 cells after 1 hour treatment with mavacamten (Paced 1 Hz).

Cells that didn't pace in a range of 0.4-1.6hz and/or ceased beating due to drug treatment were removed. A total of 26 cells were analyzed and plotted.

**Analysis of single hiPSC-CMs with disease-causing mutations**. To determine the effects of disease-causing mutations in dystrophin, we differentiated six DMD and six control hiPSC lines from four different laboratories derived from disparate cell types, including two isogenic pairs, in which mutations were corrected by CRISPR-Cas9[73], and one isogenic pair in which a deletion of the first six exons was introduced into healthy cells by CRISPR-Cas9[74,75], three non-familial healthy controls, and three DMD lines, and we have previously published a summary of some of these experiments[21]. At least 3 independent experiments were performed and 19–143 cells analyzed per line. For the additional analysis reported here, we have averaged together the control cells from all of these lines to compare with the average data from all of the DMD mutant lines.

**Statistics**. Data analysis and statistical analyses were performed with *Tableau Desktop 2019.1* (Tableau Software Inc.) and *Prism 8* (GraphPad Software LLC). Data were tested for normality versus lognormality using the Anderson-Darling test, the D'Agostino-Pearson omnibus normality test, the Shapiro- Wilk normality test, and the Kolmogorov-Smirnov normality test with Dallal-Wilkinson-Lillie for P value. The appropriate log transformation was applied to the data when log-normality was statistically motivated ($p < 0.05$). Population means were compared using two-tailed parametric t-tests, one-way analysis of variance (ANOVA) for unpaired measurements plus Tukey's correction for multiple comparisons, or two-way ANOVA with correction for multiple comparisons using Sidak hypothesis testing. $p$-values < 0.05 were considered significant. Least-square regressions were performed on the data without constraints and regression models and slope differences between treatments were analyzed using a sum-of-square F test and the ANCOVA method with significance at $p < 0.05$. All measurement were taken from distinct samples and multiple experiment and technical repeats, except for the longitudinal study and dug study where the same sample was measured repeatedly over time or after treatment.

Single-cell clustering was performed in the *Vortex clustering environment*, a freely accessible software that automatically defines high-dimensional clusters within single-cell datasets[47–49]. We used *X-shift* gradient assignment by Euclidean distance and weighted k-nearest-neighbor density estimation with the following data fields as clustering parameters: cell area, cell aspect ratio, average microsphere displacement, total contractile force, contraction and relaxation velocities, strain energy, contraction and relaxation powers, contraction and relaxation impulses, and contraction frequency and duration. The optimal number of clusters was determined via the built-in elbow detection, which was found at the free parameter value $K = 20$.

**Reporting summary**
Further information on research design is available in the Nature Portfolio Reporting Summary linked to this article.

## Data availability
The source data are provided with this paper and are available via Open Science Framework (OSF) (https://osf.io/785vp/)[76]. Source data are also provided with this paper. Source data are provided with this paper.

## Code availability
The three software modules are open-source and freely available through our GitHub repository [https://github.com/MicrosystemsLab/ContraX][77].

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

## Acknowledgements

We thank all members of the Pruitt laboratory at Stanford University and UC Santa Barbara and members of the Blau laboratory at Stanford University for helpful discussions and support. Wild-type hiPSCs (WTC cell line) were obtained as a generous gift from collaborators, and from the Allen Cell Collection[36,38,78].

This research was supported by the Swiss National Science Foundation (SNSF) Early Postdoc Mobility Fellowship (#P2SKP2_164954 to G.P.) and Postdoc Mobility Fellowship (#P400PM_180825 to G.P.), the American Heart Association (AHA Award 18POST34080160 to G.P., 20POST35211011 to A.S.V.R., and 17CSA33590101 to H.M.B. and B.L.P.); the National Institutes of Health (NIH 1R21HL13099301 and RM1GM131981 to B.L.P., and K99HL153679 to A.S.V.R); and the Baxter Foundation, Li Ka Shing Foundation and The Stanford Cardiovascular Institute to H.M.B. We also acknowledge support from the National Science Foundation GRFP (to A.K.D and R.E.W., E. A. C.), the Stanford Office of the Vice Provost for Graduate Education (to A.K.D.), Ford Foundation Pre-doctoral Fellowship (E. A. C.); the Stanford Bio-X Summer Undergraduate Research Program (to F. Birnbaum), a Major Grant from the Stanford University Vice Provost for Undergraduate Education (to F.B.), and NIH 1F31HL158227 (O.C.). This research was also supported by the National Natural Science Foundation of China (82070248 to A.C.Y.C.) and Shanghai Pujiang Program (19PJ1407000 to A.C.Y.C.); the Program for Professor of Special Appointment (Eastern Scholar) at Shanghai Institutions of Higher Learning (0900000024 to A.C.Y.C.); Innovative Research Team of High-Level Local Universities in Shanghai (A.C.Y.C.); the American Heart Association (13POST14480004 and 18CDA34110411 to A.C.Y.C.); and the Canadian Institutes of Health Research Fellowship (201411MFE-338745-169197 to A.C.Y.C.).

## Author contributions

G.P., H.L., A.S.V.R., A.C.Y.C., H.M.B., and B.L.P. conceived and designed experiments. G.P., H.L., and F.B. developed and designed the computational algorithms and software. G.P., O.C. and H.L. carried out the experiments. G.P., H.L., and K.K. carried out the computation. G.P., A.S.V.R., E.A.C., C.A.B., O.C., R.W., A.K.D., and C.H. helped with cell culture and with testing of algorithms. G.P., A.S.V.R., E.A.C., O.C., and B.L.P. analyzed data. All authors contributed scientific insights and to the writing of the manuscript.

## Competing interests

The authors declare no competing interests.
