## [Peer Review File · Nature Communications]

Reviewers' Comments:

Reviewer #1:

Remarks to the Author:

In their manuscript "Insights into single hiPSC-derived cardiomyocyte phenotypes and maturation using ConTraX, an efficient pipeline for tracking contractile dynamics" Professor Pruitt's team introduces a new tool for tracking the contractility and morphology of cultured striated muscle cells.

The authors go on to use the method, very well developed and clearly communicated in Fig 1 and accompanying text, to examine the contractility of iPS-derived cardiac myocytes in culture. Further, they go on to examine the response of these cells to growth on micropatterned ECM 'islands', variable stiffness substrates, metabolic substrate, and clustering.

By virtue of their tool, the investigators are able to track cells and cell clusters over time. This is a very important capability, especially with large populations of cells, if we are to move forward with cellular agricultural methods for supplying myocytes for cardiac cell therapies, microphysiological systems, and other such assays.

My only challenge with the paper was the section on subpopulation analysis that might point to gating strategies. I felt like this was the least developed idea in the paper but that it has, potentially, the greatest impact for the cell therapy industry's attempt to mass produce cells within certain quality specifications.

Finally, the imagery of the data-dense manuscript could use a boost. Microscopy images are lacking that would help communicate what is being tracked by ConTraX in vitro. Certainly these images exist and would be a great supplement to the figures. Figure 1 is well done but Figure 2 is so dense that it took this reviewer a considerable period of time to go back and forth from the text to the graph to see what the authors were talking about. At times, I feel as if the authors thought the figures were self explanatory. So to summarize, this is great data! Perhaps improving the imagery and corresponding text, not much but a bit, would help tell what is looking like a great story, empowering to the field.

Reviewer #2:

Remarks to the Author:

OVERALL ASSESSMENT:

The manuscript reports the tracking of the contractile cycles of patterned single cardiomyocytes derived from iPS cells, at 3 time points during an important 20 day period for their maturation. The strong point of this work is that it provides an automated workflow to analyze large amounts of data and extract useful information in short amounts of time. By streamlining an automation pipeline, the authors were able to analyze more cells, while keeping imaging data acquisition to the minimum necessary, and use the thus enabled enhanced throughput to improve the statistical insights from the analysis. Additionally, using traction force microscopy, they were able to move such analysis from a mere morphological analysis to a functional readout, improving its significance. Overall the paper is clearly written and provides useful new insights on iPS derived cardiomyocyte heterogeneity at the single cell level, as well as interplay between substrate stiffness, culture medium, cell morphology and cellular force exertion. The representation and statistical analysis of the fairly large dataset is well done, and is the highlight of the paper. The components of the software developed or the analysis they were used for, are not novel, but have not been combined and used as reported before. Thus, having the software openly available is of value, even if the utility is currently limited to compatible microscopes.

Weak points have to do with the following:

- Information is sometimes lacking on (sometimes important) aspects of the workflow, so that one does not always fully understand what was implemented, making it sometimes difficult to fully assess merits and limitations of the work.
- Some important references to the literature are lacking, which is especially true for previous TFM studies. Authors should compare their work to these studies in order to better support their claims on enhanced efficiency.
- Limitations of their work should be clearly mentioned and discussed. This is particularly true for the fact that the entire imaging and TFM analysis seem to be 2D and not 3D. The authors should elaborate more on the accuracy of the TFM workflow, and to what extent the emphasis on efficiency might negatively affect accuracy. Moreover, it is not clear whether the entire workflow necessitates patterned substrates, or whether it would also work with non-patterned substrates. These weaknesses must be addressed, as well as some other (more minor) concerns. More information on the various points that require modifications can be found below.

DETAILED COMMENTS:

1. The authors emphasise the efficiency of their workflow and at the same time refer to the inefficiency of existing workflows. Important information is missing to support these claims:
1.a. Line 75: "Despite recent progress,15,16,23 TFM data acquisition and downstream analysis remain a demanding, time consuming, and tedious process." Later, in line 77: "... data processing with currently available TFM computational packages is slow and user-input intensive". These sentences are vague. First, the authors refer to currently available TFM computational packages but they do not cite any examples that can support this statement. References 15 and 23 are articles from these same authors, the second one being a preprint. In Reference 16 one finds that with their algorithm they "improve single-cell force measurements at throughputs 100 fold higher than previously". The authors should give more specific details to illustrate in which sense their approach is more efficient than Ref. 16. At the same time, there are multiple open source 2D TFM tools that are well established and that are already user-friendly and quite optimized. To list a few: there is an ImageJ plugin (<https://sites.google.com/site/qingzongtseng/tfm>) and the works of the groups of Christian Franck (<https://www.franck.engin.brown.edu/downloads>), Dufrense (Style et al Soft Matter 2014), Sabass (Huan et al Comput. Phys. Commun 2020) or Danuser (Han et al Nat. Methods 2015). The authors should refer in their manuscript to these tools and associated papers. Moreover, the article would be stronger if the authors provide more information on how ConTraX compares to all these previous works. In this respect, claims of order of magnitude efficiency improvement by the developed software do not seem to be supported by any quantitative comparison with any of these previous work. Unless the authors would include such quantitative comparison in their manuscript, they should remove those claims.

1.b. The authors should better address how the concern about data size brought up in the introduction was addressed in this work. Eg. how much saving did the ROI cropping provide? Provide some numbers. What fraction of islands were occupied by cells? Is location detection really an advantage over imaging all islands? Since it is a fixed pattern, most automated microscopes can be asked to image at regular xy-spacing. Why go through the trouble of the first software module? Pre-selection based on parameters does not sound like a strong motive for the first module in data collection either, since at that stage one generally does not want to miss data, and selections make more sense for post-processing. Please explain the rationale, also with respect to the claimed efficiency enhancement.

2. While the focus of the manuscript concerns high throughput and efficiency of the proposed TFM procedures and routines, the authors should provide more information on the accuracy and validation of their TFM routines, as this is clearly important as well for the usefulness of the proposed tools. Even if the aspect of accuracy and validation seem to be covered more in depth in

a second parallel manuscript (preprint) of the same group (see below), more information need to be included in the current manuscript to better understand how efficiency relates to accuracy (and whether e.g. the enhanced efficiency leads to reduced accuracy). The fact that their TFM procedures seem to rely on 2D images (brightfield imaging) to achieve sufficiently fast imaging (for reasons of efficiency), raises a concern with respect to accuracy, compared to TFM procedures based on 3D imaging (such as confocal microscopy). The aspect of finding a compromise between efficiency (and the necessity of 2D imaging techniques) and accuracy must be better elaborated in the manuscript. See also below for more detailed questions:

2.a. Line 110: "The Streamlined TFM module, innovating on prior work,¹⁵ uses Digital Image Correlation (DIC) to track fluorescent fiducial markers (i.e., fluorescent microspheres) and measure the deformation of the hydrogel." In this sentence, the authors claim that the use of DIC for TFM is innovative with respect to previous work. They cite a previous paper by these authors where they used Ncorr, a DIC open source algorithm for Matlab. It is not clear in which sense the authors are innovating their previous work. In any case, while using DIC might be an innovation for the author's prior work, it is not innovative in the context of Traction Force Microscopy. Rather, DIC has been used in the field of TFM for around 20 years (again, see work of James Butler, Micah Dembo, or Christian Franck, to name a few; again, the authors should refer to these works). In fact, DIC is rather obsolete in the field as the current tendency is to acquire 3D stacks and use 3D displacement measurement algorithms such as Digital Volume Correlation (DVC, which is the 3D version of DIC). As it is well known that cells exert non-negligible out of plane tractions (i.e. with a non-zero Z component), a 2D approach (as followed in this manuscript) cannot recover these out of plane tractions. The authors must mention this as a limitation of their work and in addition must report on the traction recovery errors made by simplifying a 3D displacement field to a 2D field (see also below)

2.b. Line 980: "... by tracking the displacement of the fluorescent microspheres using digital image correlation (DIC). In this step, the user should verify that adequate DIC parameters are used; again, the TFM Benchmarking Model is helpful." The authors provide reference [64] to cite their Benchmarking Model. For reasons of methodological clarity the authors should elaborate as follows:

2.b.i. While the authors have written a parallel manuscript presenting their TFM validation tool in more detail, the authors should provide some basic information about how to tune the DIC parameters, and therefore, more details on how this Benchmarking Model works, without the need for having access to the other manuscript. This is even more important since the provided reference is a preprint that has not been peer reviewed. The authors should provide some lines describing how the ground truth displacements and tractions were generated for TFM validation purposes, which cell geometry was used for it, how this ground truth is corrupted to simulate experimental measurements and how errors are addressed. The effect of simplifying a 3D to a 2D displacement field on the recovered tractions must be addressed here.

2.b.ii. Please provide some basic guidelines on the parameters to be tuned. The paper would be stronger if some recommendations to the user would be provided. This information also seems to be missing in the preprint [64]. This information is crucial if the authors want to highlight the user friendliness of ConTraX.

2.b.iii. While I have not read the preprint [64] in detail, it seems that this is a relatively simplified simulation platform. The authors should also be aware of more recent and sophisticated ways of validating TFM methods and cite them in their manuscript. See e.g. Holenstein et al *Comput. Meth. Biomech. Biomed. Eng.*, 2019 and Barrasa-Fano et al *Acta Biomaterialia* 2021.

3. At various instances throughout the manuscript, information is missing to understand exactly what was done. Please provide more detailed information at all those instances (either within the main text, or as supplementary material):

3.a. A very important aspect of TFM is the fact that matrix (hydrogel) displacements must be calculated with respect to a stress-free hydrogel state in order to come up with the total force a cell is exerting. Typically, this state is obtained at the end of the experiment, by detaching (lysing) the cells, or alternatively at the start of the experiment before cell seeding. Did the authors do

this? Or alternatively, were hydrogel displacements calculated with respect to the least deformed hydrogel state (i.e. for the cycle time at which cardiomyocytes exert the least force)? If the latter is the case, the authors cannot calculate the total force, but only a 'relative' force (force increment, or dynamic force component with respect to baseline value). This would be an important limitation that in this case must be clearly mentioned in the manuscript.

3.b. In order to compare cellular forces and tractions (and associated parameters) over such long (20 days) culture period, it is important that the mechanical (elastic) properties of the hydrogel (polyacrylamide) as well as the adhesiveness of the coating (microcontact printing with GFP-labelled gelatin + Matrigel) do not change substantially over the 20 day period. Please provide experimental evidence that both the gel mechanical properties and coating remain sufficiently stable over the 20 day period (under the same culture conditions as for the reported study), so that their changes do not interfere with the reported force data and their evolution (maturation) in time. As to the stability of the elastic properties of polyacrylamide, the senior author of the manuscript has previously reported on non-negligible changes of polyacrylamide gels stored in PBS for 10 days (changes of elastic modulus of more than 50%, as assessed by means of AFM, see Denisin and Pruitt, ACS Applied Materials & Interfaces 2016; 8: 21893–21902). How can the authors rule out that such changes (which may be even larger in the presence of cells) are not interfering with the reported changes in cellular force exertion over time? The same is true for the ECM coating: if the coating is not stable over the reported 20 day period, cell-matrix adhesion may change, in turn affecting cellular force exertion and interfering with data interpretation concerning cell maturation.

3.c. Was the fluorescent labeling of the islands used to confirm that the variability in the cell shapes was not due to incomplete transfer of proteins during the patterning process? Please mention or describe this. See also previous question on coating stability, where the fluorescent labeling can also help in assessing the stability with time. In this respect, please clarify whether the fluorescent imaging was done only for the TFM beads or the ECM patterns as well. If both, did it increase imaging time or multiple band filters were available for simultaneous imaging? Provide sample movies used for data analysis in supplementary material.

3.d. Line 816: "Using a sequence of image-processing steps including thresholding, edge detection, masking, and filtering, thousands of relevant single hiPSC-CMs are automatically identified and located within tens of seconds." Please provide details on which edge detection algorithm was used and if there are any important parameters to be tuned. Please also mention which image filter was used.

3.e. Line 934: "To solve these problems, our algorithm identifies cell contours through a series of image-processing steps that detect both weak and strong edges despite the presence of image artifacts or fluorescent microspheres visible as black dots in bright field images." Please provide the mathematical/technical details of this algorithm.

3.f. Line 950: "From this force-versus-time trace, the algorithm identifies the peak and baseline and computes metrics including peak amplitude, duration, mid-peak duration, maximum and minimum derivatives (for example contraction and relaxation velocities), frequency, and integral under the curve (impulse)." A user would like to know what he/she is calculating. For example, were the forces compared to the maximum forces? Were the velocities tracked from cell edges or from the beads? Again, was the maximum velocity locally on a cell used? Please specify how every parameter is calculated (if applicable, provide a formula). Moreover, Figure 1.4 is not referenced in the text. As this figure seems to be a good visual support to explain these equations and associated parameter, it should be referenced.

3.g. Cells are first located with the Cell-Locator. This module analyses an image acquired at low resolution, detects the positions of the cells and, after filtering them based on filtering criteria, stores them in a file to be used in the second module. The second module then images these positions at a higher resolution. Please provide more information on the following:

3.g.i. Which positions are stored here? Is it the centroid of the cell? In this regard, since the cell segmentation is done via a Matlab script, the output will be pixel coordinates (x,y). How is that translated back into microscope coordinates? And more importantly, if those pixel coordinates have been calculated at 10x magnification, how do they translate into microscope coordinates of a

higher magnification (at 40x)? How does the user ensure that the cell will be located in the center of the image during the TFM acquisition module?

3.g.ii. The authors mention that the Automated TFM Acquisition script has an option for doing autofocus at each cell position. Does this step only corrects for possible z shifts or also realigns the ROI to keep the cell at the center of the image?

3.h. Based on Figure 1 and some parts of the text, the ConTraX workflow proposes two imaging steps: a first one using a 10x magnification and a second one using 40x. While the low magnification is mentioned in the description of the Cell-Locator step, 40x is not mentioned in the description of the Automated TFM acquisition step. Please include this here as well. Moreover, please indicate the equivalent pixel size that one obtains with each magnification.

3.i. Please provide a video of the usage pipeline of the Contrax software, as this will help the reader and potential future user of the software to better understand.

4. Please also comment on the following (minor) questions, and adapt the manuscript where necessary:

4.a. Line 97-101: "The Cell-Locator module uses a low-magnification (10x) survey of the cells on the hydrogel and generates a position list of cells matching user-defined criteria: cell area, elongation, and orientation; and fluorescence intensity and contractility thresholds. This targeted and reproducible selection alleviates user-induced cell selection bias, provides population-wide cell-morphology metrics, and increases the robustness and throughput of the assay." These two sentences are somehow contradictory. On the one hand, the authors say that the user can define parameters to select which cells to pick and immediately after, they claim that the results are free of user-induced bias. This is not entirely true. For example, the software will not provide results of cells with an area lower than the threshold that the user defines. This still biases the results. The authors should rephrase this second sentence. The main advantage of Cell-Locator is that one can have a quantitative assessment of certain parameters during imaging. This avoids selecting ROIs "by eye" and allows for selecting them with some extra quantitative assessment, but it is not completely bias free, as still dependent on user-defined thresholds.

4.b. Which software were used to make the various plots? Please mention in the manuscript. If open source tools were used, then making the plotting codes available will also help others doing similar work, especially since the plots have a clear added value in visualizing complex results in an accessible way.

4.c. Utility of the software would be enhanced if the start of the imaging was synchronized with the electrical signal to check the phase of the contractile cycles of each cell, to comment on any mechanically aided synchronization effects, even if it was ensured that cells were sufficiently far apart to not interact mechanically through the gel.

4.d. Were there any spatial patterns between cells in proximity to each other or based on location on the gel? Was there an influence of the proximity to the electrodes? These factors should at least be acknowledged, and better, checked in the data.

4.e. Choice between standard error and standard deviations is not justified in Tables 1 and 2.

4.f. Interpretations based on the highest aspect ratios or areas might be limited by the lower number of cells exhibiting them. Was a threshold used to consider a phenotype useful for the analysis (i.e. say >10 cells exhibiting it)?

4.g. How is statistical significance determined between the trend lines? Was it the sum of squares F test? Specify in image captions for clarity.

4.h. Please specify what state of cell contractile cycle was the area calculated for? The average of the maximum and minimum during the cycle?

4.i. In fig.1 time traces, when displacements are dropping to 0, why are the forces not also dropping to 0?

4.j. Please provide distance between micropatterned islands.

4.k. Fig 5B: Contraction times are reported in ms. Probably this must be s.

reviewed by Hans Van Oosterwyck (with detailed input from Ravi Sinha and Jorge Barrasa Fano)

Reviewer #3:

Remarks to the Author:

It was a pleasure to read this manuscript, which was well written and contains clear and beautiful figures. The topic is highly interesting to the field because there is a lack of automated (and therefore unbiased) approaches to quantify hiPSC-CM contraction. The authors have developed an impressive tool that allows them to automate three important steps in the experimental procedure: cell selection, video acquisition and analysis. This will contribute to more reproducible and unbiased experiments for people interested in doing traction force microscopy in single cells. With this tool they have acquired an impressive amount of data. However, there are still a number of concerns that need to be addressed.

Major points

- ConTraX is presented as the solution for measuring contractility in hiPSC-CMs and as an alternative to for example high-speed microscopy and engineered heart tissues. In reality this is not a useful head-to-head comparison. While ConTraX might be valuable to study single cells, the other approaches are able to study monolayers and/or 3D tissues. This nuance is extremely important, especially since in most recent publications it is shown that 3D tissues, often composed of multiple cell types, mature the hiPSC-CMs significantly and therefore the field is moving towards drug testing and disease modelling in 3D. The authors should acknowledge the differences and limitations of the different measurement tools and elaborate extensively on what type of measurement are only possible in single cells. This is where ConTraX will provide an advantage.
- This advantage should be demonstrated in the following sections of the manuscript. While the authors present an overwhelming amount of data new insights are lacking. The authors show that contractility changes depending on the stiffness but this is already widely known. The investigation of the power-law is interesting but limited: why is the power-law the best trend? Please explain in more detail than only the highest Rsquare. What is the adjusted Rsquare of all different fitted relationships in Figure S2? If the power-law trend is correct, how can this be used or applied?
- The main applications of hiPSC-CMs are disease modelling and drug screening. It would be useful to demonstrate how this tool can be used for one of these applications. Specifically, it would be useful to test different clinically known drugs to validate the response seen in the hiPSC-CMs.
- Cell survival & longitudinal studies: it is clear (Fig 5B, panel cell survival) that the cell population in the experiment is (slowly) dying which is unsurprisingly since cardiomyocytes favour cell-to-cell contacts. If the authors are not measuring in a stable population, how can it be clear that the changes in function over time are due to the experimental conditions instead of due to their inherent instability? Authors claim in some conditions the cells provide a more mature phenotype, but it should be made very clear whether just the average changes (for example if the already 'weaker' cells die off earlier) or whether individual cells actually improve in function. For that reason it would be an absolute necessity to normalize each measurement point against its own baseline. This will also reduce the variability of the data due to the inherent heterogeneity of the hiPSC-CM differentiations. Since the system is able to track the cells over time in such studies, this should be relatively easy to do.
- How do the different batches of differentiations map across the different parameters? It would be interesting to get insight in the reproducibility. What was the efficiency of the differentiations in terms FACS analysis of cardiac specific markers?
- Seahorse assay (Figure S4): the y-axis should report the normalization. Are the error bars std error or deviation? It seems very odd that the last two points are not significant compared to the other points: the error bars are smaller compared to other significant points. Moreover, proper analysis of this assay is lacking. It is important to calculate the different activities of the respiration: basal respiration (=first 3 points – last three points), ATP Production (=first 3 points – point 4,5,6), etc. (see https://www.agilent.com/cs/publishingimages/Cell_Mito__Profile.jpg). If there is any difference between the two groups of the calculated parameters, that is relevant. The difference between two points in the graph (as tested now) is not relevant. Pay special attention to

the non-mitochondrial activity: why is the non-mitochondrial activity so much higher in Lact versus Gluc?

- Figure 5B: calculating significant differences based on a 3-point regression slope is not accurate. The linear fit needs more points to be used for this. In addition, the opaque measurement points based on interpolation should be removed because they are not real values. It should just be a group comparison. If done, do the claims 'lactate-rich medium promotes contractile function' and 'stiffer substrate attenuate contractile function' still hold? Also, it seems that the stiffer substrates still have a higher level of contractile force if they can be measured (before they die off): it is not clear to the reader if the contractile function is actually attenuated. Again, please use a normalisation of each measurement point to its own baseline.

- "(...) maximum around aspect ratio 7:1, beyond which force decreases." This does not seem to be significant so this claim should not be made.

- "(...) preselects for the cells that are the most able to further mature over time (Figure 7)." This is not clear from Fig 7. In fact, if we look only at 10kPa there is a similar trend in quadrant I and II. It would also be interesting to know whether cells are moving from one quadrant to another.
Minor:

- In-vitro in the abstract should be in vitro.

- Page 5: .. criteria .. contractility thresholds. Is it correct to assume this is not defined by Cell-Locator but checked after full analysis?

- The beginning of the discussion just describes all the panels of the figures (and even mentions the legends of asterisks) but this should be a more integral discussion of the results.

- "These parameters yielded an acquisition frame rate of 30 frames/s on our setup, which is above the minimum necessary rate (>15 frames/s) to avoid undersampling and adequately quantify of the contractile dynamics of CMs." This claim is not backed up with data while others report (and calculate) minimum required sample rates of 70 or even 100 fps (10.3791/55461, 10.1002/cphg.67). Especially when parameters change a frame rate of 30 FPS is low and in other optical approaches ~100 fps is the standard. It is advised to try and increase this. While perhaps not possible in this work, please adjust this statement and it is advised strongly to increase the framerate in future work.

ANSWERS TO REVIEWERS' COMMENTS

REVIEWER #1 (REMARKS TO THE AUTHOR):

In their manuscript "Insights into single hiPSC-derived cardiomyocyte phenotypes and maturation using CONTRAX, an efficient pipeline for tracking contractile dynamics" Professor Pruitt's team introduces a new tool for tracking the contractility and morphology of cultured striated muscle cells. The authors go on to use the method, very well developed and clearly communicated in Fig 1 and accompanying text, to examine the contractility of iPS-derived cardiac myocytes in culture. Further, they go on to examine the response of these cells to growth on micropatterned ECM 'islands', variable stiffness substrates, metabolic substrate, and clustering. By virtue of their tool, the investigators are able to track cells and cell clusters over time. This is a very important capability, especially with large populations of cells, if we are to move forward with cellular agricultural methods for supplying myocytes for cardiac cell therapies, microphysiological systems, and other such assays.

OVERALL ANSWER TO REVIEWER 1'S COMMENTS:

We thank reviewer 1 for his/her time in reviewing our manuscript and for his/her very positive and supportive comments. We hope that our answers below, and the modification made to the manuscript will have now addressed the points raised.

POINT-BY-POINT ANSWERS TO REVIEWER 1'S COMMENTS:

My only challenge with the paper was the section on subpopulation analysis that might point to gating strategies. I felt like this was the least developed idea in the paper but that it has, potentially, the greatest impact for the cell therapy industry's attempt to mass produce cells within certain quality specifications.

Detailed Answer

Reviewer 1 made the comment that our manuscript somewhat fails in highlighting the importance and strength of our subpopulation analysis, while being to his/her opinion one of the most impactful aspects of our work. We thank Reviewer 1 for this helpful feedback. We have modified our manuscript in several places to reflect this aspect. Notably, we have added the following sentences:

- In the section CONTRAX workflow:
 - This targeted and reproducible selection, which can be applied as gates in a similar way to what is done in Flow Cytometry and Fluorescence-Activated Cell Sorting (FACS) sorting, alleviates user-induced cell selection bias, provides population-wide cell-morphology metrics, and increases the robustness and throughput of the assay.
- In the results subsection: Subpopulation analysis reveals potential gating strategies
 - Such gating would indeed be very helpful in defining robust and applicable quality standards that could be applied across academic studies and industrial development of cell models and therapies.

- In the Discussion subsection Unbiased gating potentially allows selection for the cells most able to mature:

We believe that our approach with CONTRAX and its ability to apply selection gates based on standardizable criteria may contribute to addressing the challenge of defining robust and applicable quality standards for academic studies and industrial development of cell models and therapies.

Finally, the imagery of the data-dense manuscript could use a boost. Microscopy images are lacking that would help communicate what is being tracked by CONTRAX in vitro. Certainly these images exist and would be a great supplement to the figures. Figure 1 is well done but Figure 2 is so dense that it took this reviewer a considerable period of time to go back and forth from the text to the graph to see what the authors were talking about. At times, I feel as if the authors thought the figures were self explanatory.

So to summarize, this is great data! Perhaps improving the imagery and corresponding text, not much but a bit, would help tell what is looking like a great story, empowering to the field.

Detailed Answer

Reviewer 1 also commented that the imagery could use a boost. To address this concern, we have added microscopy images in the Supplementary Information document. We have also revised Figure 2 to facilitate the intuitive understanding of the data presented and moved some data into a new Figure S2 and we added Figure S10, as well.

We thank again Reviewer 1 for his/her feedback and suggestion, which we feel have helped make our manuscript clearer, and its usefulness and impact more evident.

REVIEWER #2 (REMARKS TO THE AUTHOR):

OVERALL ASSESSMENT:

The manuscript reports the tracking of the contractile cycles of patterned single cardiomyocytes derived from iPS cells, at 3 time points during an important 20 day period for their maturation. The strong point of this work is that it provides an automated workflow to analyze large amounts of data and extract useful information in short amounts of time.

By streamlining an automation pipeline, the authors were able to analyze more cells, while keeping imaging data acquisition to the minimum necessary, and use the thus enabled enhanced throughput to improve the statistical insights from the analysis. Additionally, using traction force microscopy, they were able to move such analysis from a mere morphological analysis to a functional readout, improving its significance.

Overall the paper is clearly written and provides useful new insights on iPS derived cardiomyocyte heterogeneity at the single cell level, as well as interplay between substrate stiffness, culture medium, cell morphology and cellular force exertion. The representation and statistical analysis of the fairly large dataset is well done, and is the highlight of the paper.

OVERALL ANSWER TO REVIEWER 2'S COMMENTS:

We thank Reviewer 2 for his/her time in reviewing our manuscript and for his/her positive, supportive and detailed comments. Below we provide point-by-point answers that we hope will address the points raised.

POINT-BY-POINT ANSWERS TO REVIEWER 2'S COMMENTS:

The components of the software developed or the analysis they were used for, are not novel, but have not been combined and used as reported before. Thus, having the software openly available is of value, even if the utility is currently limited to compatible microscopes.

Detailed Answer

Regarding above general comment that our method is limited to currently compatible microscopes, we would like to answer that there is unfortunately no universal standard for microscopy. Our choice was then logically placed on the most widely applicable microscopy software, i.e. *Micro-Manager*, as well as one well establish microscopy platform from Carl Zeiss. Our custom-made scripts are transposable to other platform with some code modification, but it would be difficult to provide a universal robustly-tested solution.

Weak points have to do with the following:

- Information is sometimes lacking on (sometimes important) aspects of the workflow, so that one does not always fully understand what was implemented, making it sometimes difficult to fully assess merits and limitations of the work.

Detailed Answer

Workflow: We agree with reviewer 1 that there remains some complexity in the overall workflow. However, we feel that the section CONTRAX workflow, Figure 1 and the details provided in Supplementary Information are rather complete and clear. Further, the workflows for using the software are described step-by-step in our GitHub release and

we added the User Guide to our Supplementary documentation. Further, we have carefully reviewed our method section and Supplementary Information documents to improve our description where we found it necessary (changes were made scattered throughout the text and supplementary material and, hence, we did not report them here).

- Some important references to the literature are lacking, which is especially true for previous TFM studies. Authors should compare their work to these studies in order to better support their claims on enhanced efficiency.

Detailed Answer

References: Our choice was to cite, to the best of our knowledge, the most relevant references in the context of this work. Based on the Reviewers comments, we have added a few references that we felt were missing.

- Limitations of their work should be clearly mentioned and discussed. This is particularly true for the fact that the entire imaging and TFM analysis seem to be 2D and not 3D. The authors should elaborate more on the accuracy of the TFM workflow, and to what extent the emphasis on efficiency might negatively affect accuracy. Moreover, it is not clear whether the entire workflow necessitates patterned substrates, or whether it would also work with non-patterned substrates. These weaknesses must be addressed, as well as some other (more minor) concerns. More information on the various points that require modifications can be found below.

Detailed Answer

Limitation of the work: We thank the reviewer for his/her comments, and in line with these, we have modified the discussion section to include the following text:

Potential limitations of the method. CONTRAX perform traction force microscopy measurement using 2D image videos. Recently, traction force microscopy has also been made possible using 3D images through advances in imaging and computing. 3D TFM has the advantage of being more comprehensive and accurate in that it encompasses traction stresses occurring out-of-plane, something that is not possible with 2D images. Nevertheless, 3D imaging possesses limits time-resolution due to the plane-by-plane imaging modality currently used, making this technique difficult to apply to fast contracting cell such as cardiomyocytes. Further, the sheer volume of data to process under such conditions makes the computing requirement another challenging aspect.

In this work, we made use of micropatterned hydrogel substrates to induce our hiPSC-CMs to adopt a physiological elongated shape. While micropatterning does contribute to reducing the heterogeneity found in hiPSC-CMs morphology, micropatterning is not a strict requirement for CONTRAX. To the contrary, the gating strategy enabled by the Cell Locator module enables to rapidly identify elongated cells on substrates with or without micropatterns, which potentially alleviates the need to perform substrate micropatterning, a step that presents its own challenges despite the various methods now available.

1. The authors emphasize the efficiency of their workflow and at the same time refer to the inefficiency of existing workflows. Important information is missing to support these claims:
 - a. Line 75: “Despite recent progress,^{15,16,23} TFM data acquisition and downstream analysis remain a demanding, time consuming, and tedious process.” Later, in line 77: “... data processing with currently available TFM computational packages is slow and user-input intensive”. These sentences are vague.

Detailed Answer

We thank Reviewer 2 for his/her comment, and we agree that these two sentences can be seen as vague. We tried to make these sentences more specific with the following modifications:

- Line 75:

Despite recent progress,^{15,16,23} TFM data acquisition and downstream analysis remain time-consuming processes requiring a significant level of expertise.
- Line 77:

data processing with currently available TFM computational packages remains demanding in computing power and on user involvement.

- b. First, the authors refer to currently available TFM computational packages, but they do not cite any examples that can support this statement.

Detailed Answer

We thank Reviewer 2 for noticing this. We addressed this comment by adding an example of a widely used TFM package and a reference to existing publication proposing TFM computational package, as well as a review article where readers will be able to find more information and reference regarding force measurements tools and techniques.

- c. References 15 and 23 are articles from these same authors, the second one being a preprint.

Detailed Answer

We thank Reviewer 2 for noticing that the second reference pointed towards the pre-print of a manuscript, which was a referencing error. We have now corrected the reference to cite the published paper of this pre-print. Reviewer 2 is right that these references are from the same laboratory, but the list of authors is largely different, and the results are from past research. These are valid and relevant references.

- d. In Reference 16 one finds that with their algorithm they “improve single-cell force measurements at throughputs 100 fold higher than previously”. The authors should give more specific details to illustrate in which sense their approach is more efficient than Ref. 16.

Detailed Answer

We thank Reviewer 2 for his/her comment. It is correct that this reference claims this in reference 16 (Pushkarsky, I., Tseng, P., Black, D., France, B., Warfe, L., Koziol-White, C. J., et al. (2018). Elastomeric sensor surfaces for high-throughput single-cell force cytometry. *Nat Biomed Eng* 2, 124–137. doi:10.1038/s41551-018-0193-2.). In that manuscript, the authors propose an approach relying on an integrated biosensor material comprising fluorescently labelled elastomeric contractible surfaces to measure force from single cells. Table 1 compares the proposed technology with other methods for single cell force measurements in terms of fabrication, other requirements, image analysis, spatial resolution, time course, automation workflow, throughput/experiment, top application area. This paper notably discusses the fact that:

"TFM and micropost array methods offer superior spatial resolution that is able to assign force vectors to specific focal adhesions and measure very subtle forces in small cells like T cells. Advanced TFM techniques are now able to map complex sub-cellular forces in three-dimensional polymer and gel networks. In general, both methods should remain the standard for addressing specific biological questions relating to sub-cellular force generation by small numbers of cells. However, this resolution comes at the cost of simplicity and throughput, as these methods typically require high magnification imaging and manual user input during analysis workflows. Thus, as shown in Table 1, the experimental data throughputs have generally been limited to 10–50 cells."

To account for the Reviewer 2's comment, we therefore modified our manuscript text to provide additional details related to how our approach is addressing the issue of throughput in single cell force measurement by TFM, without compromising on spatial resolution, accuracy and standardization. On line 158, the sentence now reads:

[...], which is an order of magnitude higher than conventional manual acquisition, without compromising on spatial resolution and accuracy, and significantly reduces the burden on users.

- e. At the same time, there are multiple open source 2D TFM tools that are well established and that are already user-friendly and quite optimized. The authors should refer in their manuscript to these tools and associated papers. To list a few:
 - i. there is an ImageJ plugin (<https://sites.google.com/site/qingzongtseng/tfm>)
 - ii. the works of the groups of Christian Franck (<https://www.franck.engin.brown.edu/downloads>), Dufrense (Style et al Soft Matter 2014),
 - iii. Sabass (Huan et al Comput. Phys. Commun 2020) or
 - iv. Danuser (Han et al Nat. Methods 2015).

Detailed Answer

We thank Reviewer 2 for this comment. We have added these valid references.

- f. Moreover, the article would be stronger if the authors provide more information on how CONTRAX compares to all these previous works. In this respect, claims of order of magnitude efficiency improvement by the developed software do not seem to be supported by any quantitative comparison with any of these previous work. Unless the authors would include

such quantitative comparison in their manuscript, they should remove those claims.

Detailed Answer

We thank Reviewer 2 for this comment. In addition to our readily available quantification of the acquisition throughput in terms of >200 single cell/hour of TFM video acquisition (line 156, we have added a reference to the reference 16 (see above), where the readers will find a good discussion and Table 1 with a quantitative comparison between various single cell measurement approaches and interesting detail as per what are the advantages and disadvantages of the various approaches.

- g. The authors should better address how the concern about data size brought up in the introduction was addressed in this work.
 - i. Eg. how much saving did the ROI cropping provide? Provide some numbers.

Detailed Answer

We have added the following sentence on line 160

“Automate cell outline detection is done using a single bright field image, alleviating the use of a fluorescent die, thereby preserving a fluorescent channel for additional labelling observations, and reducing the acquisition time and the amount of user mouse clicks by >>5-10x during video analysis preprocessing.”

Cropping of the image indeed reduces the data file size, and, hence, the computation time. We did not proceed with a systematic evaluation of the computation time reduction for specific reduction in file size, but to our experience, the scaling is linear if not higher.

- ii. What fraction of islands were occupied by cells?

Detailed Answer

In our hands, we estimated that on average 10-50% of patterns are occupied by cells.

The CellLocator module could also allow the measurement of patterns occupancy or at minimum, the number of cells per substrate. However, we did not focus our manuscript on this particular aspect, which we felt could be the subject of a separate study. Many parameters may influence cell attachment. TFM substrates can be produced with or without micropatterns, with various hydrogel materials, bead colors and densities, and with different pattern dimensions and ECM proteins. Further, cell attachment depends on cell seeding density and seeding protocol. In this manuscript, we have used a published and widely-used microcontact printing protocol from Ribeiro, A. J. S., Ang, Y.-S., Fu, J.-D., Rivas, R. N., Mohamed, T. M. A., Higgs, G. C., et al. (2015). Contractility of single cardiomyocytes differentiated from pluripotent stem cells depends on physiological shape and substrate stiffness. *Proceedings of the National Academy of Sciences* 112, 201508073–12710. doi:10.1073/pnas.1508073112. The CellLocator module could also allow the measurement of the micropattern quality, but this would require to label the EMC protein and acquire additional images.

In our manuscript, we focused on the measurement and analysis of traction stress in single cells and is independent on the use of micropatterns and on the method used for the manufacturing of the micropatterns.

To make it clear that micropatterns are not strictly required, we modified the text of our manuscript on line 176:

“We note that micropatterns are not strictly required for TFM measurements, and that our measurement and analytical pipeline can be used on unpatterned single cells, as long as cell interspacing is sufficient to alleviate for unwanted cell-cell disturbances.”

- iii. Is location detection really an advantage over imaging all islands?
Since it is a fixed pattern, most automated microscopes can be asked to image at regular xy-spacing. Why go through the trouble of the first software module?

Detailed Answer

We thank Reviewer #2 for this relevant question. As Reviewer #2 points out, one approach could be to image every single islands/micropattern using fixed step increments on the microscope stage control in a deterministic fashion. However, as Reviewer 2 rightfully points out in his/her previous question, not all patterns contain a single cell due to several reasons discussed above. Micropatterns can be empty or contain multiple cells, as cell seeding is a stochastic process. Hence, imaging every micropatterns would result in the acquisition of a tremendous amount of irrelevant data. This not only would significantly lengthen the image acquisition, especially for cardiomyocyte for which second long videos must be acquired, but also create the unnecessary burden of pre-analysis data filtering to avoid analyzing videos that would not contain relevant cells. Further, as we exemplify in our manuscript, the Cell Locator module can be used upstream of TFM video acquisition to apply selection gates, further contributing to a more targeted acquisition of relevant cells. Finally, practical constraints resulting from the positioning of the plates on the microscope stage and manufacturing of micropatterned substrates often results in non-perfect alignment of the micropatterns grids with the microscope stage, which would result, in the case one would use a deterministic regular-xy-spacing image acquisition approach, in a necessary pre-acquisition calibration of the grid position to ensure that the micropattern falls in the center of the field of view at each step.

Based on Reviewer #2's question, we modified the text in our manuscript on line 120-125:

“This targeted and reproducible selection, which can be applied as gates in a similar way to what is done in Flow Cytometry and Fluorescence-Activated Cell Sorting (FACS) sorting, alleviates user-induced cell selection bias, provides population-wide cell-morphology metrics. Further, this module increases the robustness and throughput of the assay by limited and targeted acquisition of TFM videos of strictly relevant cells.”

- iv. Pre-selection based on parameters does not sound like a strong motive for the first module in data collection either, since at that stage one generally does not want to miss data, and selections make more sense for post-processing. Please explain the rationale, also with respect to the claimed efficiency enhancement.

Detailed Answer

We thank Reviewer #2 for this comment. It is correct that in a perfect world, with unlimited microscopy and computing resources, time constraints can be alleviated through imaging parallelization and computing power. However, as we mentioned in our manuscript (line 67-74):

“Despite recent progress, TFM data acquisition and downstream analysis remain time-consuming processes requiring a significant level of expertise. The acquisition of video recordings requires several hours per experiment; most of that time spent manually selecting cells of interest; and data processing with currently available TFM computational packages is demanding in computing power and on user involvement. These challenges are compounded by the size of the datasets when measuring dynamic contractions in CMs that requires seconds-long, high-framerate videos (5-10 s, 150-300 frames).”

Under these circumstances and because of the serial nature of TFM video acquisition, acquiring irrelevant data directly contributes to decreasing data acquisition throughput. Hence, pre-selection does directly contribute to increasing throughput.

Until now, this problem was compounded by the manual acquisition, where selection was made subjectively by the researcher, leading to uncontrolled and potentially biased cell selection.

Finally, in addition to the location of cells of interests, the *Cell-Locator* module yields population wide statistics on cell morphology, as additional options, freely customizable, on fluorescence and detection of cell contraction. These results are of direct value independently to the subsequent acquisition of TFM videos or of other high-resolution imaging.

2. While the focus of the manuscript concerns high throughput and efficiency of the proposed TFM procedures and routines, the authors should provide more information on the accuracy and validation of their TFM routines, as this is clearly important as well for the usefulness of the proposed tools. Even if the aspect of accuracy and validation seem to be covered more in depth in a second parallel manuscript (preprint) of the same group (see below), more information need to be included in the current manuscript to better understand how efficiency relates to accuracy (and whether e.g. the enhanced efficiency leads to reduced accuracy). The fact that their TFM procedures seem to rely on 2D imawges (brightfield imaging) to achieve sufficiently fast imaging (for reasons of efficiency), raises a concern with respect to accuracy, compared to TFM procedures based on 3D imaging (such as confocal microscopy). The aspect of finding a compromise between efficiency (and the necessity of 2D imaging techniques) and accuracy must be better elaborated in the manuscript. See also below for more detailed questions:

Detailed Answer

We thank the Reviewer #2 for his/her comment on how efficiency relates to the accuracy of the method and the potential conflict with increased efficiency. As Reviewer #2 points out, we have referred to a pre-print, where we detailed a tool for validation of our TFM analysis tool. We agree that our manuscript would benefit from more information regarding this aspect and we have therefore added extra detail in Detailed Methods in the Supplementary Information.

Reviewer #2 also raises a valid point regarding potential differences in terms of accuracy between approaches based on 2D versus 3D images and TFM analysis. It is true that 3D TFM would provide more accurate results, in absolute value, as there exists an out-of-

plane traction force component, even for single cells on planar substrates. However, as Reviewer #2 mentions, today, 3D TFM relies on z-stack imaging through confocal microscopy, for example. Because of this, this technique is difficult to apply to the measurement of rapid contractions, such as found in cardiomyocyte for example, because of limited imaging speed. Further, such microscopes are much more expensive, limiting the broad applicability of these methods. These aspects are now discussed in our manuscript in lines 479-488, with the discussion section that we added to answer an earlier comment above:

Potential limitations of the method

CONTRAX perform traction force microscopy measurement using 2D image videos. Recently, traction force microscopy has also been made possible using 3D images through advances in imaging and computing. 3D TFM has the advantage of being more comprehensive and accurate in that it encompasses traction stresses occurring out-of-plane, something that is not possible with 2D images. Nevertheless, 3D imaging possesses limits time-resolution due to the plane-by-plane imaging modality currently used, making this technique difficult to apply to fast contracting cell such as cardiomyocytes. Further, the sheer volume of data to process under such conditions makes the computing requirement another challenging aspect.

In this work, we made use of micropatterned hydrogel substrates to induce our hiPSC-CMs to adopt a physiological elongated shape. While micropatterning does contribute to reducing the heterogeneity found in hiPSC-CMs morphology, micropatterning is not a strict requirement for CONTRAX. To the contrary, the gating strategy enabled by the Cell Locator module enables to rapidly identify elongated cells on substrates with or without micropatterns, which potentially alleviates for the need to perform substrate micropatterning, a step that presents its own challenges despite the various methods now available.

Finally, we note that brightfield imaging was only used to obtain cell outline using a single image, while fluorescence was used to acquire images of the microsphere in the hydrogel from which the gel deformation and traction stress are computed.

- a. Line 110: “The Streamlined TFM module, innovating on prior work,¹⁵ uses Digital Image Correlation (DIC) to track fluorescent fiducial markers (i.e., fluorescent microspheres) and measure the deformation of the hydrogel.” In this sentence, the authors claim that the use of DIC for TFM is innovative with respect to previous work. They cite a previous paper by these authors where they used Ncorr, a DIC open source algorithm for Matlab. It is not clear in which sense the authors are innovating their previous work. In any case, while using DIC might be an innovation for the author’s prior work, it is not innovative in the context of Traction Force Microscopy. Rather, DIC has been used in the field of TFM for around 20 years (again, see work of James Butler, Micah Dembo, or Christian Franck, to name a few; again, the authors should refer to these works). In fact, DIC is rather obsolete in the field as the current tendency is to acquire 3D stacks and use 3D displacement measurement algorithms such as Digital Volume Correlation (DVC, which is the 3D version of DIC). As it is well known that cells exert non-negligible out of plane tractions (i.e. with a non-zero Z component), a 2D approach (as followed in this manuscript) cannot recover these out of plane tractions. The authors must mention this as a limitation of their work and in addition must report on the

traction recovery errors made by simplifying a 3D displacement field to a 2D field (see also below)

Detailed Answer

We thank Reviewer #2 for pointing out the potentially confusing explanation in our manuscript regarding the use of DIC. Reviewer #2 is correct to say that this has been used previously, and we did not mean to say otherwise. We modified our sentence to remove the confusion as follow (line 130):

“The Streamlined TFM module computes the traction stress from the acquired TFM videos using Digital Image Correlation (DIC) to track fluorescent fiducial markers (i.e., fluorescent microspheres) and measure the deformation of the hydrogel. Fourier Transform Traction Cytometry (FTTC) to back calculate the traction stress from the hydrogel deformation.{Butler:2002eo, Sabass:2008bj, Ribeiro:2017hm}. It innovates on prior work, {Ribeiro:2017hm} by taking advantage of parallel computing to speed up calculations, by proposing a streamlined interface to reduce the burden on the users without compromising on the control of the analysis parameters, and proposes custom-built algorithms, including automated outline detection and contraction peaks detection to automate the quantification of multiple contractile parameters: [...]”

We note that we do refer to the work of James Butler, which we believe is the first demonstration of the use of Fourier Transform Traction Cytometry for TFM.

Regarding the fact that the field is moving to 3D imaging modalities, we agree with Reviewer #2, but for the reasons we exposed above in our previous answer, this imaging modality is currently unable to meet the time-resolution required for fast contracting cells such as cardiomyocytes and requires more expensive microscopes. Again, these aspects are now discussed in our manuscript in lines 474-488, with the discussion section that we added to answer an earlier comment above:

Potential limitations of the method

CONTRAX perform traction force microscopy measurement using 2D image videos. Recently, traction force microscopy has also been made possible using 3D images through advances in imaging and computing. 3D TFM has the advantage of being more comprehensive and accurate in that it encompasses traction stresses occurring out-of-plane, something that is not possible with 2D images. Nevertheless, 3D imaging possesses limits time-resolution due to the plane-by-plane imaging modality currently used, making this technique difficult to apply to fast contracting cell such as cardiomyocytes. Further, the sheer volume of data to process under such conditions makes the computing requirement another challenging aspect.

In this work, we made use of micropatterned hydrogel substrates to induce our hiPSC-CMs to adopt a physiological elongated shape. While micropatterning does contribute to reducing the heterogeneity found in hiPSC-CMs morphology, micropatterning is not a strict requirement for CONTRAX. To the contrary, the gating strategy enabled by the Cell Locator module enables to rapidly identify elongated cells on substrates with or without micropatterns, which potentially alleviates for the need to perform substrate micropatterning, a step that presents its own challenges despite the various methods now available.

- b. Line 980: “... by tracking the displacement of the fluorescent microspheres using digital image correlation (DIC). In this step, the user should verify that adequate DIC parameters are used; again, the TFM Benchmarking Model is helpful.” The authors provide reference [64] to cite their Benchmarking

Model. For reasons of methodological clarity the authors should elaborate as follows:

While the authors have written a parallel manuscript presenting their TFM validation tool in more detail, the authors should provide some basic information about how to tune the DIC parameters, and therefore, more details on how this Benchmarking Model works, without the need for having access to the other manuscript. This is even more important since the provided reference is a preprint that has not been peer reviewed.

The authors should provide some lines describing how the ground truth displacements and tractions were generated for TFM validation purposes, which cell geometry was used for it, how this ground truth is corrupted to simulate experimental measurements and how errors are addressed. The effect of simplifying a 3D to a 2D displacement field on the recovered tractions must be addressed here. Please provide some basic guidelines on the parameters to be tuned.

The paper would be stronger if some recommendations to the user would be provided. This information also seems to be missing in the preprint [64]. This information is crucial if the authors want to highlight the user friendliness of CONTRAX.

While I have not read the preprint [64] in detail, it seems that this is a relatively simplified simulation platform. The authors should also be aware of more recent and sophisticated ways of validating TFM methods and cite them in their manuscript. e.g.

- Holenstein et al Comput. Meth. Biomech. Biomed. Eng., 2019
- Barrasa-Fano et al Acta Biomaterialia 2021.

Detailed Answer

We thank Reviewer #2 for this comment and request for additional information regarding the benchmarking model. The model was generated to validate and test the effect of analysis parameters on users-acquired images, as detailed in our answer above. It has not been designed to perform for direct comparison between 2D and 3D TFM modalities. The model is also 2D, not 3D. While this would be of interest and there are examples in the literature for such approaches, it goes beyond the scope of this manuscript. We have decided to incorporate the preprint in the Supplementary file of this manuscript.

We also thank Reviewer #2 for this comment and request for additional information regarding the tuning of the various parameters. This is indeed an important aspect for which we now have provided more information. This information figures in the *User manual* on CONTRAX available with the code on our GitHub repository, submitted as Supplementary files, as well.

The effect of 2D versus 3D measurements are now discussed in lines 474-488, with the discussion section that we added to answer an earlier comment above:

Potential limitations of the method

CONTRAX perform traction force microscopy measurement using 2D image videos. Recently, traction force microscopy has also been made possible using 3D images through advances in imaging and computing. 3D TFM has the advantage of being more comprehensive and accurate in that it encompasses traction stresses occurring out-of-plane, something that is not possible with 2D images. Nevertheless, 3D imaging possesses limits time-resolution due to the plane-by-plane imaging modality currently used, making this technique difficult to apply to fast contracting cell such as cardiomyocytes. Further, the sheer volume of data to process under such conditions makes the computing requirement another challenging aspect.

In this work, we made use of micropatterned hydrogel substrates to induce our hiPSC-CMs to adopt a physiological elongated shape. While micropatterning does contribute to reducing the heterogeneity found in hiPSC-CMs morphology, micropatterning is not a strict requirement for CONTRAX. To the contrary, the gating strategy enabled by the Cell Locator module enables to rapidly identify elongated cells on substrates with or without micropatterns, which potentially alleviates for the need to perform substrate micropatterning, a step that presents its own challenges despite the various methods now available.

3. At various instances throughout the manuscript, information is missing to understand exactly what was done. Please provide more detailed information at all those instances (either within the main text, or as supplementary material):
 - a. A very important aspect of TFM is the fact that matrix (hydrogel) displacements must be calculated with respect to a stress-free hydrogel state in order to come up with the total force a cell is exerting. Typically, this state is obtained at the end of the experiment, by detaching (lysing) the cells, or alternatively at the start of the experiment before cell seeding. Did the authors do this? Or alternatively, were hydrogel displacements calculated with respect to the least deformed hydrogel state (i.e. for the cycle time at which cardiomyocytes exert the least force)? If the latter is the case, the authors cannot calculate the total force, but only a 'relative' force (force increment, or dynamic force component with respect to baseline value). This would be an important limitation that in this case must be clearly mentioned in the manuscript.

Detailed Answer

We thank Reviewer #2 for this comment. It is correct that in TFM, the traction stress is computed from the measured deformation of the hydrogel. This is typically done using DIC in 2D TFM, as in our manuscript, or using DVC in 3D TFM. It is also correct to say that DIC computes the deformation as the difference in the beads position under increased stress compared to a resting state. In the myocardium, the main function of cardiomyocytes is to produce powerful cyclic contractions to power the heart, and most cardiomyopathies will eventually affect this function. Hence, what is of prime interest in this context is the delta of traction stress between the most relaxed and the most contracted state of the cardiomyocytes over one or averaged over multiple contractile cycles, as we do with CONTRAX. To clarify this point, we have now stated this distinction in the manuscript on line 146:

[...] automate the quantification of multiple contractile parameters: the maximum delta between the most relaxed and contracted states of the cardiomyocytes over several cycles of contractions of the force F_{\max} , work W_{\max} , power P_{\max} , strain energy E_{\max} , as well as the frequency f_{contr} , the contraction and relaxation velocities v_{contr} and v_{rel} , the contraction and

relaxation impulse I_{contr} and I_{rel} ,^{32,33} the contraction duration at maximal velocity, t_{vmax} , and the overall duration of the contraction, t_{contr} . (see Supplementary information for details of calculations) [...]

As Reviewer #2 mentions, the resting tension could also be measured against a stress-free condition, that is, a condition where the cells are absent and any stress in the gel substrate is fully released. As Reviewer #2 mention, a typical procedure to obtain the later involves measuring a ground state, without cell attached. That can indeed be done by measuring the hydrogel state prior cell attachment prior the measurement, or after the measurement, by detaching the cells. Our method is fully compatible with the measurement of the resting tension from most relaxed state, i.e. hydrogel in the absence of cell, although we did not focus on this aspect for our study on cardiomyocytes.

- b. In order to compare cellular forces and tractions (and associated parameters) over such long (20 days) culture period, it is important that the mechanical (elastic) properties of the hydrogel (polyacrylamide) as well as the adhesiveness of the coating (microcontact printing with GFP-labelled gelatin + Matrigel) do not change substantially over the 20 day period.

Please provide experimental evidence that both the gel mechanical properties and coating remain sufficiently stable over the 20 day period (under the same culture conditions as for the reported study), so that their changes do not interfere with the reported force data and their evolution (maturation) in time.

Detailed Answer

We thank reviewer for this comment. It is correct that the mechanical properties of the hydrogel, and their stability overtime, are very important parameters in TFM, and that the stability overtime can be a source of concern. The same applies to the cell adhesion material used, in our case Matrigel. This is one of the reasons for which we have chosen to stick to well characterized and established materials, polyacrylamide hydrogel and Matrigel ECM micropatterns in our study. Our own lab has published a paper on the characterization of polyacrylamide hydrogels reference: Denisin, A. K., and Pruitt, B. L. (2016). Tuning the Range of Polyacrylamide Gel Stiffness for Mechanobiology Applications. ACS Appl Mater Interfaces, acsami.5b09344. This reference went missing in our manuscript, so we thank Reviewer #2 for his comment. The reference has now been added back.

In this reference, we showed using atomic force microscopy that some polyacrylamide hydrogel formulations do affect the stability over time of polyacrylamide hydrogels. We found that gel elastic modulus increases with increasing cross-link concentration until an inflection point, after which gel stiffness decreases with increasing cross-linking. This behavior arises because of the formation of highly cross-linked clusters, which add inhomogeneity and heterogeneity to the network structure, causing the global network to soften even under high cross-linking conditions. We identified these inflection points for three different total polymer formulations. When we alter gelation kinetics by using a low polymerization temperature, we find that gels are stiffer when polymerized at 4 °C compared to room temperature, indicating a complex relationship between gel structure, elasticity, and network formation. We also investigate how gel stiffness changes over time and find that specific gel formulations undergo significant stiffening (1.55 ± 0.13), which may be explained by differences in gel swelling resulting from initial polymerization parameters. Based on these data and our experience and expertise, we chose hydrogel formulations which demonstrated good performance for the duration of our experiments.

We also made use of appropriate controls for our experiments and only proceeded with a comparative analysis between conditions against a control condition. We also ensure that each cell batch was measured on hydrogel devices fabricated from the same polyacrylamide mix and on micropatterned substrate fabricated in parallel on the same days. Further, we use the same batch of acrylamide and bi-acrylamide and initiator for all experiments. We consistently use the same pipettes and consistently manufactured the device on the day prior to cell seeding. We believe that our rigorous and consistent approach with appropriate controls is the best way in obtaining controlled substrate properties and alleviate for any change over time. We further clarified this aspect by adding in the Supplementary information:

“These polyacrylamide formulations were chosen to maintain good mechanical properties for the duration of our experiments.⁴³”

As to the stability of the elastic properties of polyacrylamide, the senior author of the manuscript has previously reported on non-negligible changes of polyacrylamide gels stored in PBS for 10 days (changes of elastic modulus of more than 50%, as assessed by means of AFM, see Denisin and Pruitt, ACS Applied Materials & Interfaces 2016; 8: 21893–21902).

- i. How can the authors rule out that such changes (which may be even larger in the presence of cells) are not interfering with the reported changes in cellular force exertion over time?
- ii. The same is true for the ECM coating: if the coating is not stable over the reported 20 day period, cell-matrix adhesion may change, in turn affecting cellular force exertion and interfering with data interpretation concerning cell maturation.

Detailed Answer

We agree with the Reviewer #2 that the stability of the mechanical properties of the substrate, in our case polyacrylamide, and of the ECM coating, in our case Matrigel, is a very important aspect in longitudinal studies. To address this concern, we have added a comment in our manuscript to acknowledge this possibility in line 492 and the need for appropriate experimental controls:

[...] Another potential concern exists in the implementation of the workflow in extended longitudinal studies regarding the stability of the mechanical properties of the hydrogel substrates and of the ECM micropatterns, and appropriate controlled must be used. [...]

- c. Was the fluorescent labeling of the islands used to confirm that the variability in the cell shapes was not due to incomplete transfer of proteins during the patterning process? Please mention or describe this. See also previous question on coating stability, where the fluorescent labeling can also help in assessing the stability with time. In this respect, please clarify whether the fluorescent imaging was done only for the TFM beads or the ECM patterns as well. If both, did it increase imaging time or multiple band filters were available for simultaneous imaging? Provide sample movies used for data analysis in supplementary material.
- d.

Detailed Answer

We thank Reviewer #2 for this comment. It is true that it is possible to add a fluorescent label to some ECM proteins to visualize the micropatterns. Various methods exist to micropattern TFM hydrogels and each come with their own advantages and drawbacks. Micro-contact printing, which we used in our work, is not straightforward to master to obtain consistent results. This is why, as explained above, we have stuck to well-established and published protocols and have used extreme caution in staying consistent in our manufacturing, and, again, we made use of experimental control comparisons to alleviate for such eventualities.

In our work, we have used Matrigel as ECM matrix, as Matrigel remains the gold standard for culture of hiPSC- cardiomyocytes and many other cells. We have used fluorescent protein, or even to use a labeling kit to label the Matrigel proteins themselves. This allows to visualize and characterize the micropatterns. In our manuscript, we have used fluorescent protein spiking, using Alexa fluor-488 labelled gelatin, spiked into the Matrigel to validate our protocol for the generation of the micropatterns (this information was missing in the methods section, which is now corrected). In **Figure 1.2** and **Figure 2 A**, we show a fluorescent micrograph of such a fluorescent-labelled ECM micropattern overlaid with the fluorescent beads used for TFM measurement. To address the concern of the reviewer, we have added in our manuscript that the use of fluorescent ECM is compatible with our methods, in line 122:

[...]Additionally, fluorescently-labelled ECM proteins could be used to monitor the quality of each micropatterns, as shown in **Figure 2 C**.[...]

In the rest of our work and the bulk of our TFM measurements, we have not used such labels as we wanted to demonstrate the power of the approach to compare the effects on patient lines in a straightforward manner. We made use of appropriate comparison to a control condition in our analysis. We have provided a sample movie in our github repository for code testing.

- e. Line 816: “Using a sequence of image-processing steps including thresholding, edge detection, masking, and filtering, thousands of relevant single hiPSC-CMs are automatically identified and located within tens of seconds.” Please provide details on which edge detection algorithm was used and if there are any important parameters to be tuned. Please also mention which image filter was used.

Detailed Answer

We thank Reviewer #2 for this detailed question. Cell outline was performed as follow, which we also added to Detailed Methods in Supplementary Information:

Edge detection was performed using Matlab built in edge function using the “Canny” methods and sensitivity thresholds of [0.001,.4] to ignore edge that are not stronger than these threshold values. It is possible that image quality may vary depending on the imaging setup used, in which case the threshold values may need adjustment. However, in our hands, these values proved robust with images of varying qualities.

- f. Line 934: “To solve these problems, our algorithm identifies cell contours through a series of image-processing steps that detect both weak and strong edges despite the presence of image artifacts or fluorescent microspheres

visible as black dots in bright field images.” Please provide the mathematical/technical details of this algorithm.

Detailed Answer

We thank Reviewer #2 for this very detailed but relevant question related to the implementation of the cell contour detection algorithm. Our algorithm makes use of built-in functions in Matlab to detect the cell contour in brightfield images of the cardiomyocytes. Because the code is quite lengthy (305 lines), we outlined in the Supplementary information the most important steps of the workflow as follow:

The function `adapthisteq` was used to enhance the contrast of the grayscale image by transforming the values using contrast-limited adaptive histogram equalization (CLAHE)

The function `imbinatfilt` was applied as an edge-preserving Gaussian bilateral filter to the grayscale or RGB image, with a calculate degree of smoothing based on the image two-dimensional standard deviation, and with a standard deviation of the smoothing kernel that is also calculated from the image spatial resolution.

The function `edge` was applied to detect the cell edges using the “Canny” methods and sensitivity thresholds of [0.001 0.5] to ignore edge that are not stronger than these threshold values.

The resulting binary image was then dilated to connect the detected edges and filled. The results binary image is then eroded by a similar factor.

The resulting binary objects are then filtered to exclude small object resulting from noise or strongly showing beads and the convex hull was taken to obtain a single object from remaining individual objects.

The resulting image was then used to refined the edge detection using the function `activecontour` with 'Chan-Vese' method, a maximum 100 iteration and a 'SmoothFactor' of 3.

The contour was then smoothened once more with an erode and dilate steps.

- g. Line 950: “From this force-versus-time trace, the algorithm identifies the peak and baseline and computes metrics including peak amplitude, duration, mid-peak duration, maximum and minimum derivatives (for example contraction and relaxation velocities), frequency, and integral under the curve (impulse).” A user would like to know what he/she is calculating. For example, were the forces compared to the maximum forces? Were the velocities tracked from cell edges or from the beads? Again, was the maximum velocity locally on a cell used? Please specify how every parameter is calculated (if applicable, provide a formula). Moreover, Figure 1.4 is not referenced in the text. As this figure seems to be a good visual support to explain these equations and associated parameter, it should be referenced.

Detailed Answer

We thank the reviewer for this comment. For the sake of space, we had not detailed all the calculations in the main manuscript and proposed a graphical schematic providing a visual summary of the measured parameter in Figure 1.3.4 (to which Reviewer #2 referred as 1.4 in his comment), We now specifically refer to each subsection 1-4 of Figure 1.3 in the manuscript text, e.g. Figure 1.3.4. To further address this comment, and for the sake of

completeness, we have added the most important formulas for the calculation in the Supplementary Information and added Figure S10.

We thank Reviewer #2 for this comment. Figure 1 has three columns, each of whom is referred to in the text. Column 3 had four subparts, labelled 1), 2), 3) and 4), which we believe led to the confusion that Figure 1.4 was not referred to. It is now referred to 1.3.4.

- h. Cells are first located with the Cell-Locator. This module analyses an image acquired at low resolution, detects the positions of the cells and, after filtering them based on filtering criteria, stores them in a file to be used in the second module. The second module then images these positions at a higher resolution. Please provide more information on the following: Which positions are stored here? Is it the centroid of the cell? In this regard, since the cell segmentation is done via a Matlab script, the output will be pixel coordinates (x,y). How is that translated back into microscope coordinates? And more importantly, if those pixel coordinates have been calculated at 10x magnification, how do they translate into microscope coordinates of a higher magnification (at 40x)? How does the user ensure that the cell will be located in the center of the image during the TFM acquisition module? The authors mention that the Automated TFM Acquisition script has an option for doing autofocus at each cell position. Does this step only corrects for possible z shifts or also realigns the ROI to keep the cell at the center of the image?

Detailed Answer

We thank Reviewer #2 for this detailed question. We clarified these points in the Supplementary information as follow:

[...] The algorithm calculates the centroid position of the cells. The output is translated from pixels to microns using the micron/pixel conversion for the objective used. The position of a cell in an image is then translated into the absolute stage position using the stage position of the image. Once the stage position is known, it is possible to use any magnification to image a given cell. This requires well calibrated stage and objective turret alignment, or the application of an offset may need to be applied to all position to compensate. When starting a high-magnification acquisition, the user measure this offset and apply the necessary offset compensation to the rest of the positions in the list. The Automated TFM Acquisition allows to perform an autofocus at each cell position, which is recommended to correct for the focal plane offset often occurring with a change of objective and magnification on most microscopes. [...]

The autofocusing modalities often differ from microscope setups to setups. In our case, the autofocusing was software based and allowed to compensate for our setup's slight misalignment in terms of objective co-planarity. In most, if not all microscopes, the autofocus does generally not control the x-y stage position, but only the z-position, which would be required to recenter a cell.

- i. Based on Figure 1 and some parts of the text, the CONTRAX workflow proposes two imaging steps: a first one using a 10x magnification and a second one using 40x. While the low magnification is mentioned in the description of the Cell-Locator step, 40x is not mentioned in the description of

the Automated TFM acquisition step. Please include this here as well. Moreover, please indicate the equivalent pixel size that one obtains with each magnification.

Detailed Answer

We thank Reviewer #2 for this remark. These pieces of information were indeed missing and we have now corrected for this in the manuscript. The pixel size at 40x was 0.275 microns.

- j. Please provide a video of the usage pipeline of the CONTRAX software, as this will help the reader and potential future user of the software to better understand.

Detailed Answer

We thank Reviewer #2 for this comment. We have included a complete step-by-step user manual with screenshots and comments in our github repository and we do believe that it provides more information than we could possibly convey through a video.

4. Please also comment on the following (minor) questions, and adapt the manuscript where necessary:
 - a. Line 97-101: “The Cell-Locator module uses a low-magnification (10x) survey of the cells on the hydrogel and generates a position list of cells matching user-defined criteria: cell area, elongation, and orientation; and fluorescence intensity and contractility thresholds. This targeted and reproducible selection alleviates user-induced cell selection bias, provides population-wide cell-morphology metrics, and increases the robustness and throughput of the assay.” These two sentences are somehow contradictory. On the one hand, the authors say that the user can define parameters to select which cells to pick and immediately after, they claim that the results are free of user-induced bias. This is not entirely true. For example, the software will not provide results of cells with an area lower than the threshold that the user defines. This still biases the results. The authors should rephrase this second sentence.:

The main advantage of Cell-Locator is that one can have a quantitative assessment of certain parameters during imaging. This avoids selecting ROIs “by eye” and allows for selecting them with some extra quantitative assessment, but it is not completely bias free, as still dependent on user-defined thresholds.

Detailed Answer

We thank Reviewer #2 for this remark. We have modified our manuscript text as follow:

“The Cell-Locator module uses a low-magnification (10x) survey of the cells on the hydrogel and generates a position list of cells, with the option of using user-defined criteria: cell area, elongation, and orientation; and fluorescence intensity and contractility thresholds, to filter for cell matching these criteria. This automated, targeted and reproducible selection, which can be applied as gates in a similar way to what is done in Flow Cytometry and Fluorescence-Activated Cell Sorting (FACS) sorting, alleviates user-induced cell selection bias across experiments, provides population-wide cell-morphology metrics. Further, this module

increases the robustness and throughput of the assay by limited and targeted acquisition of TFM videos of strictly relevant cells. The generated position list also enables tracking the same single cells over time, thereby enabling longitudinal studies.”

- b. Which software were used to make the various plots? Please mention in the manuscript. If open source tools were used, then making the plotting codes available will also help others doing similar work, especially since the plots have a clear added value in visualizing complex results in an accessible way.

Detailed Answer

We thank Reviewer #2 for this question. The visualization software is Tableau Desktop™, which is not open-source but has educational licenses accessible to a large number of institutions. The name of the software is now mentioned in the in the Online Methods section of the manuscript. We note that this visualization software is not necessary to our approach.

- c. Utility of the software would be enhanced if the start of the imaging was synchronized with the electrical signal to check the phase of the contractile cycles of each cell, to comment on any mechanically aided synchronization effects, even if it was ensured that cells were sufficiently far apart to not interact mechanically through the gel.

Detailed Answer

We thank Reviewer #2 for this comment. It is true that additional features could follow from synchronization of electrical pacing with imaging. As this involves specialized pieces of hardware independent from conventional imaging setups, and as electrical pacing is not a requirement per say for using our method. We have added a comment in the manuscript that our approach could be combined with synchronized pacing with appropriate hardware design triggering in Supplementary information, as follow:

“If hardware allows, synchronizing electrical pacing and video acquisition could allow to further study the excitation-contraction coupling for example. “

- d. These factors should at least be acknowledged, and better, checked in the data.
 - i. Were there any spatial patterns between cells in proximity to each other or based on location on the gel?
 - ii. Was there an influence of the proximity to the electrodes?

Detailed Answer

We thank Reviewer #2 for this comment. We did not observe any spatial pattern across the gel, or because of cell proximity, but we did not specifically look for this. Nevertheless, we do not think that this should be a source of concern in our case, because our micropatterned we designed to be sufficiently spaced to avoid cells to influence one-another, based on the publication: Tang, X., Tofangchi, A., Anand, S. V. & Saif, T. A. A Novel Cell Traction Force Microscopy to Study Multi-Cellular System. PLoS Comput Biol 10, e1003631 (2014). We added the following sentence in Supplementary information to clarify:

"We designed sufficient distance ($>200\ \mu\text{m}$) between the micropatterns to ensure adequate decoupling of cell from other nearby contracting cells.⁷¹"

We did not either investigated the effect of electrodes proximity, which could be a subject of future studies, but in our experiments, we did not notice obvious differences apart from that that regardless of the positioning of the electrodes, some cardiomyocytes do not respond to pacing while some do, a likely consequence of heterogeneity in hiPSC-cardiomyocyte populations.

- e. Choice between standard error and standard deviations is not justified in Tables 1 and 2.

Detailed Answer

We thank Reviewer #2 for this comment. Table 1 provides a "*Summary of cell morphology measurements across experimental groups*". For this analysis, we find of most immediate interest to use the standard deviation, as it provides a measure of the dispersion or variance of the cell morphology in each group. Indeed, we want to show the mean value and its statistical comparison, but describe how large a variance is found in each group, something that the standard error would not do. We made this clear in the caption of Table 1 by adding:

"Mean values show a statistically significant decrease in aspect ratio on stiffer 35 kPa substrates and larger cell area in, M16_{glu}. The standard deviations show a large variance across each group. "

In Table 2, we provide a "Summary of contractile force measurements across experimental groups. Control conditions are 10 kPa substrate stiffness, M16_{lac} culture medium, and experimental day 20." For these data, we use a linear regression on the data and provide the standard error for the slope of the linear model, which estimates the standard deviation of the estimate, i.e., the linear regression, and is a most helpful in comparing the estimates in this case. We clarified this in the caption of Table 2.

- f. Interpretations based on the highest aspect ratios or areas might be limited by the lower number of cells exhibiting them. Was a threshold used to consider a phenotype useful for the analysis (i.e. say >10 cells exhibiting it)?

Detailed Answer

We thank Reviewer #2 for this detailed comment. We did not use any threshold to dismiss bins with low sample count, and, for this precise reason, we did not make any categorical conclusions between each of these (sometime small) groups, but rather only consider the trends across all groups for cell with aspect ratio $> 1:7$, stating:

"Nevertheless, as expected from the dimensional constraints imposed by the micropatterns, cells with an aspect ratio exceeding that of the micropattern ($>7:1$) had a smaller spread area than the maximum possible $2500\ \mu\text{m}^2$ on the micropattern (**Figure 2D**) due to a reduced cell width. Hence, the cells were adequately constrained on the micropatterned and the large variance is mainly a result of cell heterogeneity."

We note that the group size is found in Figure 2 C.

- g. How is statistical significance determined between the trend lines? Was it the sum of squares F test? Specify in image captions for clarity.

Detailed Answer

We thank Reviewer #2 for this comment. The statistical comparison between the trend line was calculated in GraphPad Prism using the built-in linear regression analysis that relies on the analysis of covariance ANCOVA. We have completed the existing information in the Statistic section in the Supplemental Information to avoid repeating this information in all figure captions. It now reads:

“Least-square regressions were performed on the data without constraints and regression models and slope differences between treatments were analyzed using a sum-of-square F test and the ANCOVA method with significance at $p < 0.05$.”

- h. Please specify what state of cell contractile cycle was the area calculated for? The average of the maximum and minimum during the cycle?

Detailed Answer

We thank the reviewer for this comment. The cell outline was calculated on a single brightfield image of the cardiomyocytes, without specific control of the state of contraction of the cell. Thus, the results may randomly include some cells in a contracted state. The relative change in cell area between the contracted and relaxed state remain very small compared to the observed changes across the measured groups, and in comparison, to the large variance observed in the population. On average the same number of cells may be contracted or relaxed state in each measured group, resulting in a statistically negligible deviation that is averaged across the large sample.

- i. In fig.1 time traces, when displacements are dropping to 0, why are the forces not also dropping to 0?

Detailed Answer

We thank Reviewer #2 for this detail comment. It is correct that, while the average beads displacement should fall to zero when a cell fully relaxes (except if an image drift or edge effect generate a net displacement in the DIC analysis that is not properly filtered by the algorithm), the total contractile force may sometime not fall to zero because it is computed as the scalar sum, and not the average, of the traction stress. Since traction stress computation is a noise sensitive analysis, the summation of the resulting noisy traction stress signal will result in non-zero total contractile force in the most relaxed state. This is why the algorithm computes the total contractile force as the difference between the most relaxed state and the most contracted state, instead of just the max amplitude. We made this clearer in the manuscript with the following sentence in the Supplementary information:

“The computation of the baseline for the most relaxed cell state is of high importance as the traction stress measurement is prone to noise that often results in non-zero values for the parameters calculated from scalar values of the displacement or stress field.”

- j. Please provide distance between micropatterned islands.

Detailed Answer

We thank Reviewer #2 for this comment. This data went missing in our method section, We now have added them back. It now reads:

“1-cm² stamps able to print >4000 patterns of 132.3 μm x 18.9 μm \cong 2500 μm^2 and aspect ratio 7:1, interspaced with 200 μm .”

k. Fig 5B: Contraction times are reported in ms. Probably this must be s.

Detailed Answer

We thank Reviewer #2 for this remarks. This was wrong in our manuscript and we have now corrected the axis label from ms to s.

reviewed by Hans Van Oosterwyck (with detailed input from Ravi Sinha and Jorge Barrasa Fano)

REVIEWER #3 (REMARKS TO THE AUTHOR):

It was a pleasure to read this manuscript, which was well written and contains clear and beautiful figures. The topic is highly interesting to the field because there is a lack of automated (and therefore unbiased) approaches to quantify hiPSC-CM contraction. The authors have developed an impressive tool that allows them to automate three important steps in the experimental procedure: cell selection, video acquisition and analysis. This will contribute to more reproducible and unbiased experiments for people interested in doing traction force microscopy in single cells. With this tool they have acquired an impressive amount of data.

However, there are still a number of concerns that need to be addressed.

POINT-BY POINT ANSWERS TO REVIEWER 3'S COMMENTS:

Major points

1. CONTRAX is presented as the solution for measuring contractility in hiPSC-CMs and as an alternative to for example high-speed microscopy and engineered heart tissues. In reality this is not a useful head-to-head comparison. While CONTRAX might be valuable to study single cells, the other approaches are able to study monolayers and/or 3D tissues. This nuance is extremely important, especially since in most recent publications it is shown that 3D tissues, often composed of multiple cell types, mature the hiPSC-CMs significantly and therefore the field is moving towards drug testing and disease modelling in 3D. The authors should acknowledge the differences and limitations of the different measurement tools and elaborate extensively on what type of measurement are only possible in single cells. This is where CONTRAX will provide an advantage.

Detailed Answer

We thank *Reviewer #3* for this helpful comment and agree that our head-to-head comparison was not necessary. We agree that these approaches are complementary and modified our introduction to better reflect these complementarities as follow:

“Current assays of cardiomyocyte contractile function are often mostly qualitative, lack field-wide standards, and generally fail to offer the scalability and throughput necessary for larger-scale studies. For example, high-speed video microscopy of CM monolayer contraction is relatively uncomplicated and scalable, but it only provides semi-quantitative measurements of contractile frequency and speed. Recent advances have enabled quantifying the total contractile force. Multicellular constructs, engineered heart tissue or heart-on-chip systems advantageously reveal the effect of cell-cell interactions, which was shown to promote hiPSC-CM maturation. although these data often remain difficult to standardize across platforms and studies. Complementary to these approaches, traction force microscopy (TFM) enable quantitative and standardizable measurements of contractile force at the single-cell level and has the potential to resolve the direct impact of subcellular defect, to reveal cell-to-cell variations, and to characterize population heterogeneity.”

2. This advantage should be demonstrated in the following sections of the manuscript. While the authors present an overwhelming amount of data new insights are lacking.

The authors show that contractility changes depending on the stiffness but this is already widely known.

Detailed Answer

As above, we agree with Reviewer #3 that single-cell level measurement have unique advantages compared to 3D tissues while acknowledging some existing disadvantages. In our manuscript, we not only validated previous findings gathered on until-now limited dataset sizes, i.e., that stiffness impact contractility, but we also show an intricate relationship between cell morphology and contractile maturity and its evolution over time, which is novel. We further demonstrate how different metabolic regimens impact contractile maturation over time in terms of contractility, and in terms of cell viability and morphology, which are all novel findings as well.

Nevertheless, following Reviewer #3 comment, we further point out how single cell resolution allows for further analysis of intracellular processes in conjunction to contractility measurements, such as measurements of sarcomere dynamics, mitochondrial activity, electrophysiology, and other live assays that can advantageously be performed on single cells, as well as potentially downstream omics on single cells, as follow:

“Single cell measurement also allows for performing other types of live assays at the single cell level, such as sarcomere dynamics or electrophysiology patch clamp measurements.”

The investigation of the power-law is interesting but limited: why is the power-law the best trend? Please explain in more detail than only the highest Rsquare.

- a. What is the adjusted Rsquare of all different fitted relationships in Figure S2?
- b. If the power-law trend is correct, how can this be used or applied?

Detailed Answer

We thank Reviewer #3 for his comment. The trend emerged from data analysis and statistical analysis revealed that the best trend was following a power-law. Interestingly, power-laws are often found in nature as underlying the relations between biological mass and metabolism across various length scale for example. In the context of hiPSC-cardiomyocyte, it is interesting to find such a power-law underlying contraction velocity and contraction force at the single cell level, because it does possibly reveal an analogy with the pressure-volume relation found in the heart at the single cardiomyocyte level. However, further studies would be needed to support such interpretation. Here, the take home message is that stiffness not only impacts the contractile force, but also the contractile velocity. The fact that it follows a power-law rather than a linear trend is of lesser direct importance.

3. The main applications of hiPSC-CMs are disease modelling and drug screening. It would be useful to demonstrate how this tool can be used for one of these applications. Specifically, it would useful to test different clinically known drugs to validate the response seen in the hiPSC-CMs.

Detailed Answer

We thank Reviewer #3 for his comment. We agree that demonstrating the usefulness for drug screening and disease modelling is important. Hence, we performed significant

additional work to add a drug study with the drug Mavacamten, as well as a study of the impact of disease mutation (Duchenne Muscular Dystrophy) on contractile function of hiPSC-CMs. This has added multiple specific paragraphs as well as 2 Figures: Figure 8 and Figure 9, as well as Figure S8 & S9, Table S4, S5 & S6, reproduced below.

We thank again Reviewer #3 for his comment, as we feel these additional data have significantly improved our manuscript.

Figure 8: Mavacamten treatment affects contractile function in hiPSC-CM. A) Bar plots showing a slight but consistent increase in single cell of the area and no significant change in cell aspect ratio upon treatment. **B)** Bar plots showing a consistent decrease in total force and contraction velocity in single cells but no change in contraction duration upon treatment. **C)** Spider plot providing a summary of the comparison of all measured parameters normalized to control, pre-Mavacamten treatment. The color shading groups parameters related to: green: cell morphology, blue: contractile stress, red: contractile dynamics, orange: temporality (see **Figure S10**) Statistics show: Mavacamten 1h post-treatment vs. control on 10 kPa substrates. For all panels, $*p < 0.05$, $**p < 0.005$, $***p < 0.001$, $****p < 0.0001$. ns, not significant.

Figure 9: Increased substrate stiffness exacerbated the effect of dilated cardiomyopathy causing DMD-mutation in hiPSC-CMs. A) Spider plots providing a summary of the comparison of all measured parameters normalized control hiPSCs on 10 kPa substrates. The color shading groups parameters related to: green: cell morphology, blue: contractile stress, red: contractile dynamics, orange: temporality (see **Figure S10**). Statistics show: DMD vs. control healthy hiPSC-CMs on 10 kPa and 35 kPa substrates. C) Bar plots showing a decrease in single cell of the area, but not in aspect ratio. D) Bar plots showing that increased substrate stiffness leads to a dramatic loss of total force and synchronicity on stiffer fibrotic-like 35 kPa substrates, but not on healthy 10 kPa substrates, and a loss in contraction velocity, contraction duration in DMD-hiPSCs compared to control hiPSCs. For all panels, * $p < 0.05$, ** $p < 0.005$, *** $p < 0.001$, **** $p < 0.0001$. ns, not significant.

Figure S8: Mavacamten treatment on 10 kPa substrate stiffness. Statistics with t-test: * $p < 0.05$, ** $p < 0.005$, *** $p < 0.001$, **** $p < 0.0001$. ns, not significant. (see Table S4 for the statistics)

Mean value Standard deviation Number of sample	CTRL vs MAVA	
	Control	Mavacamten
Area (um ²)	9.28e+02 2.39e+02 24	** 1.31e+03 4.12e+02 24
Aspect ratio (-)	6.09e+00 9.34e-01 24	ns 5.46e+00 1.38e+00 24
Beads displacement (m)	1.60e-07 8.59e-08 24	** 2.90e-08 2.95e-08 24
Contactile moment (Nm)	1.50e-12 8.56e-13 24	**** 4.38e-13 3.08e-13 24
Contraction duration (s)	5.84e-01 1.60e-01 24	ns 5.16e-01 2.52e-01 24
Contraction impulse (Ns)	3.92e-08 2.69e-08 24	** 1.51e-08 1.31e-08 24
Contraction power (W)	1.07e-10 1.01e-10 24	*** 1.08e-11 2.24e-11 24
Contraction velocity (m/s)	4.66e-07 2.66e-07 24	**** 9.45e-08 8.50e-08 24
Force x-axis (N)	1.25e-07 7.58e-08 24	*** 3.73e-08 4.17e-08 24
Force y-axis (N)	1.78e-08 1.36e-08 24	ns 1.37e-08 9.97e-09 24
Frequency (Hz)	6.39e-01 2.64e-01 24	ns 5.36e-01 2.52e-01 24
Relaxation impulse (Ns)	4.46e-08 3.49e-08 24	* 2.11e-08 2.29e-08 24
Relaxation power (W)	-8.56e-11 7.68e-11 24	*** -9.70e-12 1.61e-11 24
Relaxation velocity (m/s)	-4.21e-07 2.24e-07 24	**** -8.80e-08 8.04e-08 24
Strain energy (J)	2.87e-14 3.53e-14 24	** 2.09e-15 3.28e-15 24
Total force (N)	1.25e-07 7.80e-08 24	**** 3.91e-08 4.11e-08 24
Total impulse (Ns)	8.38e-08 6.09e-08 24	**** 3.62e-08 3.24e-08 24

Table S4: Summary statistics of Mavacamten treatment on 10 kPa substrate stiffness. Statistics with t-test: * $p < 0.05$, ** $p < 0.005$, *** $p < 0.001$, **** $p < 0.0001$. ns, not significant.

A

Figure S9: Multiple parameters measured on cardiomyocytes or without DMD mutations on 10 kPa and 35 kPa substrates. DMD hiPSC-CMs clearly demonstrate a general contractile deficiency that is exacerbated on stiffer fibrotic-like substrates. Statistics with t-test: * $p < 0.05$, ** $p < 0.005$, * $p < 0.001$, **** $p < 0.0001$. ns, not significant. (see Table S5 for the statistics)**

Mean value Standard deviation Number of sample	DMD vs CTRL / Stiffness			
	CTRL		DMD	
	10 kPa	35 kPa	10 kPa	35 kPa
Aspect ratio	7.56e+00	7.15e+00	6.87e+00	6.92e+00
	1.80e+00	2.14e+00	2.45e+00	2.22e+00
	35	177	24	282
Beads displacement	4.28e-08	3.13e-08	6.47e-09	1.63e-08
	5.03e-08	4.14e-08	8.09e-09	1.72e-08
	154	233	61	282
Brightfield contractility	4.87e-07	3.49e-07	3.50e-07	2.02e-07
	2.50e-07	7.55e-07	1.92e-07	1.33e-07
	55	251	19	244
Brightfield synchronicity	1.70e-01	2.42e-01	1.48e-01	3.59e-01
	1.55e-01	2.17e-01	1.16e-01	3.58e-01
	55	256	21	254
Contraction power	7.99e-14	3.50e-14	1.24e-14	1.88e-14
	1.17e-12	1.16e-13	4.76e-14	1.20e-13
	492	735	295	692
Contraction time	2.49e-01	3.04e-01	2.15e-01	2.76e-01
	1.43e-01	1.76e-01	1.19e-01	1.10e-01
	490	728	289	601
Contraction velocity	3.10e-07	1.88e-07	1.81e-07	1.04e-07
	9.19e-07	2.23e-07	2.65e-07	1.63e-07
	517	730	295	692
Force x-axis	3.69e-08	1.27e-07	2.69e-08	5.29e-08
	4.72e-08	2.40e-07	3.10e-08	4.82e-08
	142	194	50	238
Force y-axis	3.55e-08	9.69e-08	3.12e-08	5.03e-08
	4.11e-08	1.42e-07	4.12e-08	4.65e-08
	135	186	49	234
Frequency	1.03e+00	9.88e-01	1.19e+00	1.05e+00
	4.72e-01	5.80e-01	5.46e-01	4.92e-01
	518	727	290	601
Major Axis Length	1.05e+02	9.58e+01	8.29e+01	8.97e+01
	2.27e+01	2.65e+01	2.69e+01	2.45e+01
	35	177	24	282
Minor Axis Length	1.41e+01	1.37e+01	1.25e+01	1.33e+01
	2.42e+00	2.50e+00	2.58e+00	2.44e+00
	35	177	24	282
Relaxation power	-1.24e-14	-3.33e-14	-2.30e-15	-6.72e-15
	2.10e-14	1.22e-13	3.84e-15	1.07e-14
	154	234	61	282
Relaxation velocity	-1.59e-07	-1.19e-07	-1.12e-07	-6.69e-08
	2.67e-07	1.57e-07	1.83e-07	1.14e-07
	490	730	295	690
Spreading area	1.20e+03	1.16e+03	1.11e+03	9.81e+02
	4.68e+02	4.64e+02	4.52e+02	4.06e+02
	294	635	223	660
Time between main peaks	-1.24e-14	-3.33e-14	-2.30e-15	-6.72e-15
	2.10e-14	1.22e-13	3.84e-15	1.07e-14
	154	234	61	282
Total force	3.96e-08	1.36e-07	2.66e-08	6.07e-08
	8.23e-08	2.18e-07	4.26e-08	9.44e-08
	489	729	294	686

Table S5: Summary statistics of DMD vs control on 10 kPa and 35 kPa substrate stiffness. Statistics with t-test: * $p < 0.05$, ** $p < 0.005$, *** $p < 0.001$, **** $p < 0.0001$. ns, not significant.

Class/Health/ACC	lines	Gene	Age	Tissue	Gender	Source	Used in study
CCM	BMD1	BMD	(e1833_11(Ctrl))	8 PS/VC	male	Stanford	BMD study
CCM	BMD18	BMD	(e10171(Cr7))	10 dermal fibroblast	male	University of Washington	BMD study
CCM	BMD10	BMD	(e4018_01(Ctrl)(ClonTC))	8 dermal fibroblast	male	University of Washington	BMD study
CCM	BMD3	BMD	(e4800(CrC))	10 PS/VC	male	Stanford	BMD study
CCM	BMD3	BMD	(e3004_007(Del))	0 PS/VC	male	Stanford	BMD study
CCM	UC1016.8	BMD Isogenic	Ctrl 1-e1833 Ctrl of UC3.6	unrepaired Urine	male	University of Washington	BMD study
Healthy	BMD18Isa	Isogenic normal	normal of BMD #18	10 dermal fibroblast	male	University of Washington	BMD study
Healthy	BMD10Isa	Isogenic normal	normal of BMD #10	8 dermal fibroblast	male	University of Washington	BMD study
Healthy	10 normal	Healthy normal	normal	18 unrepaired	male	Stanford	BMD study
Healthy	Norm1	Healthy normal	normal	40 unrepaired	male	Harvard	BMD study
Healthy	Norm2	Healthy normal	normal	18 unrepaired	female	Harvard	BMD study
Healthy	UC3.6	Healthy normal	normal	unrepaired Urine	male	University of Washington	BMD study
Healthy	UTC	Healthy normal	normal	10 Ctrl fibroblast	male	Japan (NIGMS&IPSC) Allen Institute, Coriell	Time-course study & Maturation study

Table S6: Summary of cell lines used in the various studies.

4. Cell survival & longitudinal studies: it is clear (Fig 5B, panel cell survival) that the cell population in the experiment is (slowly) dying which is unsurprisingly since cardiomyocytes favour cell-to-cell contacts.
 - a. If the authors are not measuring in a stable population, how can it be clear that the changes in function over time are due to the experimental conditions instead of due to their inherent instability?

Detailed Answer

We thank Reviewer #3 for this comment. It is true that the cell population in the experiment are slowly dying or detaching from the patterns. This is either due to what Reviewer #3 mention of cardiomyocytes favoring cell-cell contact or to experimental condition. To alleviate for this inevitable fact, we made use of appropriate controls and perform a comparison over time between two populations. To clarify in the manuscript, we added the following sentence in the section concerning the cell survival:

“We note that appropriate controls and statistical repeats are important to make the experiments independent of experimental condition.”

- b. Authors claim in some conditions the cells provide a more mature phenotype, but it should be made very clear whether just the average changes (for example if the already ‘weaker’ cells die off earlier) or whether individual cells actually improve in function. For that reason it would be an absolute necessity to normalize each measurement point against its own baseline. This will also reduce the variability of the data due to the inherent heterogeneity of the hiPSC-CM differentiations. Since the system is able to track the cells over time in such studies, this should be relatively easy to do.
 - c. How do the different batches of differentiations map across the different parameters? It would be interesting to get insight in the reproducibility.

Detailed Answer

We thank Reviewer #3 for this comment. Indeed, it is of interest to understand how the changes occur, something that our single cell data is advantageously able to reveal. This observation can also be made on the clustering trajectory analysis in Figure 7. We also appreciated the idea to normalize the measurement against its own baseline.

To further clarify this, we added a figure in Supplementary material showing the underlying maturation trajectory for single cells. The batched are all combined and visible in this figure, and while it revealed some variation between batches, as expected, but which are forming a consistent trend overall. Our data strengthened the fact that higher

analysis throughput is necessary to study multiple batches to alleviate batch to-batch variability found in hiPSC-CMs.

Figure S6 Evolution of the measured contractile force in individual single cells over the time course experiment for both substrate stiffness and medium compositions, showing how higher stiffness and M16gluc medium result in reduced maturation and accelerated cell death.

- d. What was the efficiency of the differentiations in terms FACS analysis of cardiac specific markers?

Detailed Answer

We did not perform FACS analysis of cardiac differentiation makers in this study, but we made use of well described differentiation protocols resulting in consistent differentiation. Further, our single cell approach only analyzed cells with detectable contractions, thereby focusing only on cardiomyocytes. We have clarified this in the manuscript in the Methods.:

“We only measured beating cardiomyocyte to alleviate for potential low efficiency of some differentiation batches.”

5. Seahorse assay (Figure S4):

- a. the y-axis should report the normalization.

Detailed Answer

We thank Reviewer #3 for this remark and changed the axis label.

- a. Are the error bars std error or deviation? It seems very odd that the last two points are not significant compared to the other points: the error bars are smaller compared to other significant points. Moreover, proper analysis of this

assay is lacking. It is important to calculate the different activities of the respiration: basal respiration (=first 3 points – last three points), ATP Production (=first 3 points – point 4,5,6), etc. (see https://www.agilent.com/cs/publishingimages/Cell_Mito_Profile.jpg). If there is any difference between the two groups of the calculated parameters, that is relevant. The difference between two points in the graph (as tested now) is not relevant.

Detailed Answer

We thank Reviewer #3 for this comment, The error bars are standard error. We complemented the image caption to include this information that was missing. We performed our statistical analysis using Prism 8.4 and verified our calculations of statistics. We further performed grouped analysis of data points under similar conditions and updated our statistics.

We have now complemented our analysis by comparing the relevant points in a grouped manner, and this does not change the conclusion of our analysis.

- b. Pay special attention to the non-mitochondrial activity: why is the non-mitochondrial activity so much higher in Lact versus Gluc?

Detailed Answer

We thank Reviewer #3 for this comment. It is indeed something we observed and which correlates with previously reported measurements (Chang et al, Stem Cell Reports, 2021). We believe this could be due to the response of hiPSC-CMs to change of media when prepared for the Seahorse assay, which requires an unbuffered proprietary media. This warrants further dedicated investigations. We did normalize our results to the last data point, and again, this does not change the conclusion of our analysis.

The new normalized data are now shown in Figure S2

2. Figure 5B: calculating significant differences based on a 3-point regression slope is not accurate. The linear fit needs more points to be used for this. In addition, the opaque measurement points based on interpolation should be removed because they are not real values. It should just be a group comparison. If done, do the claims 'lactate-rich medium promotes contractile function' and 'stiffer substrate attenuate contractile function' still hold? Also, it seems that the stiffer substrates still have a higher level of contractile force if they can be measured (before they die off): it is not clear to the reader if the contractile function is actually attenuated. Again, please use a normalisation of each measurement point to its own baseline.

Detailed Answer

We thank Reviewer #3 for his comments. Due to the readily large experimental load of running time-course longitudinal measurements, and despite the gain in throughput offered by CONTRAX, it was not possible to include more time point in our analysis. Nevertheless, it was indeed possible to normalize the data to d20 for more direct comparisons, something that we have done and have included in addition to the presented data, in an extra panel in Figure 5B. This confirms that the trends observe with the linear regressions is correct and the the M16lac lactate-rich medium promotes contractile function and that stiffer substrate tend to affect function over time.

Further, we normalized all our data to the appropriate control in Figure 3, 4 & 5. We plotted using a spider chart all the normalized measured parameters for easy comparison across the multiparameter space, and provided all the necessary statistics for these measurements in Figure 3, 4 & 5, Figure S4 & S5 and Table S3. The full statistical analysis will be made available as Supplementary Information.

Figure 3: Contraction force, velocity, and power are impacted by substrate stiffness at day 20 in M16_{lac} medium. A) Balloon plot of the contractile force as a function of cell aspect ratio and area (grouped by range) shows a generally higher force on 35 kPa substrate and fewer cells of large spread area **B)** Force production increases with aspect ratio until reaching a maximum around aspect ratio 7:1, beyond which force seems to taper off. **C)** Contraction force and contraction power follow a, potentially allometric, power law (linear in this log-log plot) with contraction velocity that significantly depends on substrate stiffness. **D)** Spider plot providing a summary of the comparison of all measured parameters at day 20, normalized to 10 kPa in M16_{lac} medium. The color shading groups parameters related to: green: cell morphology, blue: contractile stress, red: contractile dynamics, orange: temporality (see **Figure S10**). Statistics show: 35 kPa vs. 10 kPa in M16_{lac} medium at day 20. In all panels, * $p < 0.05$, ** $p < 0.005$, *** $p < 0.001$, **** $p < 0.0001$. ns, not significant.

Figure 4: Contraction force, velocity, and power are equally dependent on area and aspect ratio for both medium compositions at day 20 on 10 kPa substrates. **A)** Balloon plot of the contractile force as a function of cell aspect ratio and area (grouped by range) shows more elongated and larger cell in M16_{glu} than in M16_{lac}. **B)** Force is equally dependent on aspect ratio in both media, except in for aspect ratios of 3-6. **C)** Contraction force and power follow a power law (linear in log-log plot) with contraction velocity. The linear regression for the force shows a weak dependence on medium composition. **D)** Spider plot providing a summary of the comparison of all measured parameters at day 20, normalized to 10 kPa in M16_{lac} medium. The color shading groups parameters related to: green: cell morphology, blue: contractile stress, red: contractile dynamics, orange: temporality (see **Figure S10**). Statistics show: M16_{glu} vs. M16_{lac} on day 20 and 10 kPa substrates. For all panels, * $p < 0.05$, ** $p < 0.005$, *** $p < 0.001$, **** $p < 0.0001$. ns, not significant.

Figure 5: The contractile function in micropatterned hiPSC-CMs better develop over time in M16_{lac} versus M16_{glu} medium. **A)** Ball plot showing that cells die sooner in M16_{glu} than in M16_{lac} medium, notably on stiffer 35 kPa substrate. Cells in M16_{lac} tend to increase in **B)** CONTRAX measured changes in multiple parameters over 20 days, for both medium conditions and substrate stiffnesses. *Left:* Absolute value of the parameters at each time point with circle size proportional to the number of cells measured; the vertical bars showing the standard error; and the lines showing linear regressions. **** $p < 0.0001$ for the comparison of the slope of linear regressions; ns, not significant. *Right:* Value of the parameters normalized to d20 showing divergence over time as function of media composition. See Supplementary Information **Figure S4** for more parameters. **C)** Spider plots providing a summary of the comparison of all measured parameters over time, normalized to control 10 kPa in M16_{lac} medium. The color shading groups parameters related to: green: cell morphology, blue: contractile stress, red: contractile dynamics, orange: temporality (see **Figure S10**). Statistics show: turquoise: M16_{lac} vs. control M16_{lac} - d20 - 10kPa; purple: M16_{glu} vs. control M16_{lac} - d20 - 10kPa; black: M16_{lac} vs. M16_{glu}. For all panels, * $p < 0.05$, ** $p < 0.005$, *** $p < 0.001$, **** $p < 0.0001$. ns, not significant.

Figure S4 Evolution of all the measured contractile parameters over the time course experiment for both substrate stiffness and medium compositions. (see **Table S3** for data)

Figure S5 Day 20-normalized evolution of all the measured contractile parameters over the time course experiment for both substrate stiffness and medium compositions.

mean value standard deviation n	Media / Stiffness / Exp. day											
	M16-1						M16-2					
	10kPa			35kPa			10kPa			35kPa		
	d20	d30	d40	d20	d30	d40	d20	d30	d40	d20	d30	d40
Avg. Area (um ²)	7.64e+02 3.50e+02 591	8.27e+02 4.23e+02 377	9.11e+02 4.60e+02 220	7.16e+02 2.91e+02 282	7.02e+02 3.74e+02 138	8.23e+02 4.36e+02 45	8.72e+02 3.70e+02 498	6.73e+02 3.15e+02 259	8.35e+02 4.88e+02 29	9.37e+02 3.82e+02 224	5.69e+02 2.60e+02 59	4.64e+02 2.19e+02 7
Avg. Aspect ratio (-)	4.27e+00 1.95e+00 591	4.38e+00 2.05e+00 377	4.23e+00 2.22e+00 220	3.02e+00 1.83e+00 282	3.49e+00 1.69e+00 138	3.29e+00 1.51e+00 45	4.09e+00 2.00e+00 498	4.03e+00 1.84e+00 259	4.41e+00 2.63e+00 29	4.03e+00 1.84e+00 224	2.47e+00 1.23e+00 59	2.58e+00 9.10e-01 7
Avg. Average contraction displacement (m)	2.09e-08 3.51e-08 566	1.49e-08 2.69e-08 375	2.89e-08 3.94e-08 218	7.51e-09 1.77e-08 282	8.61e-09 1.08e-08 138	7.77e-09 7.07e-09 40	1.94e-08 3.36e-08 494	1.73e-08 2.86e-08 257	1.27e-08 1.36e-08 28	9.33e-09 8.41e-09 223	9.27e-09 8.74e-09 59	5.48e-09 4.24e-09 6
Avg. Contraction Force [nN]	3.02e+01 4.71e+01 556	2.33e+01 3.55e+01 372	4.33e+01 5.64e+01 214	5.21e+01 6.26e+01 282	5.21e+01 6.24e+01 137	6.42e+01 7.42e+01 37	4.30e+01 6.09e+01 496	2.64e+01 4.09e+01 257	2.32e+01 4.57e+01 28	8.75e+01 7.57e+01 223	4.73e+01 5.77e+01 59	3.21e+01 3.39e+01 6
Avg. Contraction impulse (Ns)	8.01e-09 1.90e-08 555	5.52e-09 1.17e-08 370	7.76e-09 1.83e-08 214	1.56e-08 2.35e-08 282	1.09e-08 1.82e-08 137	8.23e-09 9.49e-09 36	1.47e-08 3.11e-08 487	7.38e-09 1.31e-08 255	4.58e-09 1.09e-08 28	2.62e-08 3.63e-08 223	1.03e-08 1.76e-08 59	4.92e-09 6.25e-09 6
Avg. Contraction power (W)	1.99e-11 1.03e-10 588	2.33e-11 1.68e-10 374	3.37e-11 1.05e-10 218	1.47e-11 5.30e-11 281	1.66e-11 2.85e-11 137	1.74e-11 2.52e-11 45	5.48e-11 3.31e-10 493	1.88e-11 9.31e-11 259	8.85e-12 2.66e-11 29	2.12e-11 3.73e-11 221	1.44e-11 3.55e-11 58	4.63e-12 6.18e-12 7
Avg. Contraction time (s)	2.03e-01 1.90e-01 530	2.11e-01 2.18e-01 361	2.11e-01 1.37e-01 210	2.86e-01 2.15e-01 282	2.31e-01 1.58e-01 134	1.84e-01 5.44e-02 40	2.40e-01 2.05e-01 445	2.44e-01 1.37e-01 235	1.06e-01 1.62e-01 27	2.91e-01 1.97e-01 206	2.94e-01 2.42e-01 57	2.20e-01 1.01e-01 6
Avg. Contraction velocity (m/s)	1.75e-07 2.64e-07 566	1.32e-07 2.24e-07 375	2.30e-07 2.93e-07 218	6.16e-08 1.50e-07 282	7.89e-08 9.01e-08 138	7.78e-08 6.02e-08 40	1.51e-07 2.42e-07 494	1.39e-07 2.14e-07 257	1.15e-07 1.23e-07 28	7.11e-08 6.76e-08 223	7.09e-08 7.16e-08 59	5.28e-08 4.24e-08 6
Avg. Frequency (autocorr. of peaks) (Hz)	9.96e-01 5.57e-01 591	9.42e-01 6.13e-01 376	1.09e+00 4.22e-01 220	7.48e-01 4.69e-01 282	1.09e+00 6.03e-01 138	1.34e+00 5.20e-01 45	8.45e-01 5.68e-01 498	9.28e-01 6.00e-01 259	1.04e+00 6.44e-01 29	9.77e-01 5.79e-01 224	1.15e+00 7.02e-01 59	1.07e+00 2.64e-01 7
Avg. Lambda	4.71e-10 1.32e-09 591	5.40e-10 1.43e-09 377	4.78e-10 1.22e-09 220	1.89e-10 4.77e-10 282	9.18e-11 3.24e-10 138	2.58e-10 5.56e-10 45	4.93e-10 1.37e-09 498	3.20e-10 1.05e-09 259	2.56e-10 1.01e-09 29	7.58e-11 2.97e-10 224	1.52e-10 4.32e-10 59	1.28e-16 0.00e+00 7
Avg. Length (um)	6.40e+01 2.55e+01 591	6.69e+01 2.70e+01 377	6.92e+01 3.15e+01 220	5.14e+01 2.23e+01 282	5.51e+01 2.42e+01 138	5.78e+01 2.44e+01 45	6.65e+01 2.56e+01 498	5.82e+01 2.26e+01 259	6.73e+01 3.00e+01 29	6.92e+01 2.71e+01 224	4.20e+01 1.79e+01 59	3.93e+01 1.58e+01 7
Avg. Relaxation impulse (Ns)	1.20e-08 3.71e-08 556	7.33e-09 1.09e-08 372	9.78e-09 1.27e-08 214	2.21e-08 3.12e-08 282	1.46e-08 2.05e-08 137	1.23e-08 1.46e-08 37	1.74e-08 3.37e-08 496	9.40e-09 1.58e-08 257	9.28e-09 2.89e-08 28	3.44e-08 4.45e-08 223	1.20e-08 1.67e-08 59	8.26e-09 8.43e-09 6
Avg. Relaxation power (W)	-1.34e-11 4.48e-11 588	-2.35e-11 2.13e-10 374	-3.43e-11 1.48e-10 218	-1.16e-11 5.77e-11 281	-1.14e-11 2.05e-11 137	-1.33e-11 2.04e-11 45	-5.85e-11 3.39e-10 495	-1.44e-11 6.91e-11 259	-6.96e-12 2.24e-11 29	-1.59e-11 3.71e-11 221	-9.78e-12 2.42e-11 58	-3.24e-12 4.34e-12 7
Avg. Relaxation velocity (m/s)	-1.05e-07 1.58e-07 566	-8.00e-08 1.38e-07 375	-1.42e-07 2.06e-07 218	-3.66e-08 8.67e-08 282	-4.93e-08 5.87e-08 138	-4.39e-08 3.95e-08 40	-1.04e-07 1.85e-07 494	-8.51e-08 1.44e-07 257	-6.92e-08 6.69e-08 28	-4.33e-08 4.37e-08 223	-4.45e-08 3.94e-08 59	-3.22e-08 2.44e-08 6
Avg. Strain (J)	2.03e-15 1.10e-14 577	1.85e-15 1.44e-14 376	3.37e-15 9.82e-15 219	1.76e-15 6.85e-15 281	1.85e-15 3.46e-15 138	1.49e-15 2.10e-15 40	4.15e-15 1.71e-14 495	1.77e-15 8.53e-15 258	5.38e-16 1.32e-15 28	2.45e-15 5.05e-15 222	1.46e-15 3.28e-15 59	5.62e-16 7.23e-16 5
Avg. Total force tot (N)	3.02e-08 4.71e-08 556	2.33e-08 3.55e-08 372	4.33e-08 5.64e-08 214	5.21e-08 6.26e-08 282	5.21e-08 6.24e-08 137	6.42e-08 7.42e-08 37	4.30e-08 6.09e-08 496	2.64e-08 4.09e-08 257	2.32e-08 4.57e-08 28	8.75e-08 7.57e-08 223	4.73e-08 5.77e-08 59	3.21e-08 3.39e-08 6
Avg. Total force x (N)	2.58e-08 4.02e-08 562	2.04e-08 3.24e-08 373	3.77e-08 4.94e-08 218	4.22e-08 5.01e-08 282	4.53e-08 5.49e-08 137	5.31e-08 6.20e-08 38	3.42e-08 4.96e-08 495	2.18e-08 3.66e-08 257	1.87e-08 3.23e-08 28	7.34e-08 6.54e-08 224	4.21e-08 5.18e-08 59	2.84e-08 2.94e-08 6
Avg. Total force y (N)	1.16e-08 2.22e-08 573	8.99e-09 1.12e-08 376	1.46e-08 2.23e-08 214	2.61e-08 3.64e-08 282	2.24e-08 2.81e-08 138	2.61e-08 3.06e-08 40	1.99e-08 2.99e-08 495	1.13e-08 1.37e-08 258	1.13e-08 2.67e-08 28	3.97e-08 3.32e-08 222	1.87e-08 1.97e-08 59	1.38e-08 1.17e-08 6
Avg. Total impulse (Ns)	2.00e-08 5.42e-08 555	1.29e-08 2.15e-08 370	1.75e-08 2.97e-08 214	3.77e-08 5.23e-08 282	2.56e-08 3.63e-08 137	2.08e-08 2.35e-08 36	3.19e-08 6.15e-08 487	1.67e-08 2.75e-08 255	1.39e-08 3.96e-08 28	6.06e-08 7.74e-08 223	2.23e-08 3.25e-08 59	1.32e-08 1.34e-08 6
Avg. Width (um)	1.56e+01 3.48e+00 591	1.62e+01 4.25e+00 377	1.74e+01 4.20e+00 220	1.88e+01 4.89e+00 282	1.65e+01 4.03e+00 138	1.83e+01 4.44e+00 45	1.73e+01 4.17e+00 498	1.53e+01 3.88e+00 259	1.69e+01 5.73e+00 29	1.80e+01 3.55e+00 224	1.77e+01 3.37e+00 59	1.52e+01 1.98e+00 7

Table S3: Summary statistics of longitudinal study on 10 and 35 kPa substrate stiffnesses and in M16_{lac} and M16_{gluc} media.

1. “(...) maximum around aspect ratio 7:1, beyond which force decreases.” This does not seem to be significant so this claim should not be made.

Detailed Answer

We thank Reviewer #3 for this comment. We changed our wording for: “which force seems to taper off”, instead of “decreases”.

2. “(...) preselects for the cells that are the most able to further mature over time (Figure 7).”
 - a. This is not clear from Fig 7. In fact, if we look only at 10kPa there is a similar trend in quadrant I and II.

Detailed Answer

We thank Reviewer #3 for this remark. We changed our wording to better reflect our intended message with:

“enable preselecting most-consistently for the cell that are the most able to further mature over time”

- b. It would also be interesting to know whether cells are moving from one quadrant to another.

Detailed Answer

We thank Reviewer #3 for this remark. Most naturally, cells do cross the arbitrary boundaries. Our analysis enables such analysis, but we did not think that it was the most interesting aspect to present in our manuscript.

Minor comments:

1. In-vitro in the abstract should be in vitro.

Detailed Answer

We corrected for this.

2. Page 5: .. criteria .. contractility thresholds. Is it correct to assume this is not defined by Cell-Locator but checked after full analysis?

Detailed Answer

It is in fact not correct to assume this. Criteria and thresholds are parameters that can be defined upstream in the Cell-Locator, but these parameters can be adapted or reused for further analysis down-stream of data acquisition.

3. The beginning of the discussion just describes all the panels of the figures (and even mentions the legends of asterisks) but this should be a more integral discussion of the results.

Detailed Answer

We fail to find what Reviewer #3 refers to, because the first paragraph of the discussion starts with the shortened duration of analysis and discuss this quite generally.

4. “These parameters yielded an acquisition frame rate of 30 frames/s on our setup, which is above the minimum necessary rate (>15 frames/s) to avoid undersampling and adequately quantify of the contractile dynamics of CMs.” This claim is not backed up with data while others report (and calculate) minimum required sample rates of 70 or even 100 fps (10.3791/55461, 10.1002/cphg.67). Especially when parameters change a frame rate of 30 FPS is low and in other optical approaches ~100 fps is the standard. It is advised to try and increase this. While perhaps not possible in this work, please adjust this statement and it is advised strongly to increase the framerate in future work.

Detailed Answerzs

We agree that there is no strict upper limit on sampling rate and that some methods have proposed and higher frame rate, especially important when considering calcium transient during contractions. But we agree that our sentence should be revise to reflect this and we reformulated as follow:

“These parameters yielded an acquisition frame rate of 30 frames/s on our setup, which is above a minimum necessary rate (>15 frames/s) to avoid undersampling and adequately quantify of the contractile dynamics of CMs, although a higher frame rate may be possible and sometime advantageous.”

Reviewers' Comments:

Reviewer #2:

Remarks to the Author:

I am very sorry to read that some of the authors have gone through very difficult times and I can only hope that they have been able to cope with such difficulties.

I have gone through the authors' reply and revised manuscript and appreciate the level of detail with which answers are provided. Most of our comments were related to methods, in particular missing information and limitations of the approach. The authors have included much more details on methods in the main text and especially the supplementary information, including a well elaborated user guide and a preprint. They are acknowledging the limitations of their 2D TFM approach and the use of brightfield images in their study, and provide a rationale for this choice (namely data throughput). While the consequences of this choice for traction accuracy are not really tackled in the manuscript, I also agree that this goes beyond the scope of the study (and at the same time, traction recovery is performed by means of well established algorithms and the algorithms are sufficiently detailed in the supplementary information).

I only have 3 minor comments left that require feedback from the authors:

While the authors mention in the reply that on average 10-50% of patterns are occupied by cells, this information is not included in the manuscript. I find it relevant to include this in the text, as it provides an additional and clear rationale for the Cell-Locator.

The new line 122 says: "[...]Additionally, fluorescently-labelled ECM proteins could be used to monitor the quality of each micropatterns, as shown in Figure 2 C.[...]" I believe this should be Figure 2A.

I only noticed now that in Figure 1.3.3 displacement magnitudes are of the order of 40 nm. I am confused about such low values, as they are below the resolving power of a diffraction-limited microscope. The same seems to be the case for the displacement magnitudes mentioned in Table S3. Perhaps there is something wrong with the units, or I am misinterpreting these values.

Reviewer #3:

Remarks to the Author:

The authors have improved the manuscript significantly by i) adding valuable new data (mavacamten and DMD response), ii) enhancing the representation of the vast amount of data further (spider plots) and iii) addressed almost all comments appropriately. For those reviewer comments where no changes were made to the manuscript, the authors explain well why this is the case (e.g. explanation of the power law).

One small point however, remains a thorn in the side: there is no reference or mathematical evidence that justifies stating 15 frames/s as a minimum necessary rate to avoid undersampling. If no justification is given, this reviewer would like to see this part of the sentence removed because it might give researchers a false sense of security. As mentioned in the previous comments, others report a minimum of 70 or even 100 fps, and the authors should provide evidence if they are challenging these numbers, or leave it out.

In summary, the manuscript has been improved significantly by the authors and it will be a valuable addition to the field.

Kind regards,
Berend van Meer

Reviewer #4:
None

Reviewer #5:
None

REVIEWERS' COMMENTS AND ANSWER FROM THE AUTHORS

Reviewer #2 (Remarks to the Author):

1. While the authors mention in the reply that on average 10-50% of patterns are occupied by cells, this information is not included in the manuscript. I find it relevant to include this in the text, as it provides an additional and clear rationale for the Cell-Locator.

Reply and modifications made by the authors

Thank you, we have now added the mention of the average pattern occupancy in the manuscript, which was indeed missing.

“On average, 10-50% of patterns are occupied by cells, and the Cell-Locator module identified >200 relevant cells per patterned gel substrates out of a total of 1600 patterns regions per gel substrates, increasing both the speed and throughput of acquisition compared to an undirected acquisition across every possible patterns.”

2. The new line 122 says: “[...]Additionally, fluorescently-labelled ECM proteins could be used to monitor the quality of each micropatterns, as shown in Figure 2 C.[...]” I believe this should be Figure 2A.

Reply and modifications made by the authors

Thank you, we have indeed modified the reference to the Figure 2a instead of 2c.

3. I only noticed now that in Figure 1.3.3 displacement magnitudes are of the order of 40 nm. I am confused about such low values, as they are below the resolving power of a diffraction-limited microscope. The same seems to be the case for the displacement magnitudes mentioned in Table S3. Perhaps there is something wrong with the units, or I am misinterpreting these values

Reply and modifications made by the authors

Thank you for this specific question. The reason of the small order of displacement magnitude lies in that this value is the average displacement magnitude of pixels within the cell area. Hence, while pixels at the extremities of a cardiomyocyte show a much larger displacement dipole, the average value include the close to null displacement of pixels at the center of the cell contraction dipole, resulting in overall much smaller value of the average displacement, as explained in Supplementary Information file:

“The average displacement in the ROI is defined as:

$$d(t_k) = \frac{1}{N} \sum_{k=1}^N \sqrt{u_{k,x}^2 + u_{k,y}^2}, \quad (1)$$

where $k = 1, \dots, N$ corresponds to the frame number in the video and (u_x, u_y) are the displacement vector at each pixel.”

To avoid confusion, we modified the text in the figure 1c from “Displacement” to “Average displacement”.

Reviewer #2 (Remarks on code availability):

Reviewer #3 (Remarks to the Author):

1. One small point however, remains a thorn in the side: there is no reference or mathematical evidence that justifies stating 15 frames/s as a minimum necessary rate to avoid undersampling. If no justification is given, this reviewer would like to see this part of the sentence removed because it might give researchers a false sense of security. As mentioned

in the previous comments, others report a minimum of 70 or even 100 fps, and the authors should provide evidence if they are challenging these numbers, or leave it out.

Reply and modifications made by the authors

Thank you. We have removed this part of the sentence as we agree higher frame rate are beneficial and that one should avoid providing a false sense of security.

Reviewer #3 (Remarks on code availability):

2. The code does contain a clear README file, however, I had no MATLAB at this point to test the code in detail.

Reply and modifications made by the authors

Thank you. We are working on the release of an executable that can be ran independently of Matlab and we will add it to our Github repository in the future.